# True Zero-Shot Inference of Dynamical Systems Preserving Long-Term Statistics

**Christoph Jürgen Hemmer[1,3]** , **Daniel Durstewitz[1,2,3]**

[1]Dept. of Theoretical Neuroscience, Central Institute of Mental Health,
Medical Faculty Mannheim, Heidelberg University, Mannheim, Germany
[2]Interdisciplinary Center for Scientific Computing (IWR), Heidelberg University, Germany
[3]Faculty of Physics and Astronomy, Heidelberg University, Heidelberg, Germany
{christoph.hemmer, daniel.durstewitz}@zi-mannheim.de

## Abstract

Complex, temporally evolving phenomena, from climate to brain activity, are governed by dynamical systems (DS). DS reconstruction (DSR) seeks to infer generative surrogate models of these from observed data, reproducing their long-term behavior. Existing DSR approaches require purpose-training for any new system observed, lacking the zero-shot and in-context inference capabilities known from LLMs. Here we introduce *DynaMix*, a novel multivariate ALRNN-based mixture-of-experts architecture pre-trained for DSR, the first DSR model able to generalize zero-shot to out-of-domain DS. Just from a provided context signal, without any re-training, DynaMix faithfully forecasts the long-term evolution of novel DS where existing time series (TS) foundation models, like Chronos, fail – at a fraction of the number of parameters (0.1%) and orders of magnitude faster inference times. DynaMix outperforms TS foundation models in terms of long-term statistics, and often also short-term forecasts, even on real-world time series, like traffic or weather data, typically used for training and evaluating TS models, *but not at all part of DynaMix' training corpus*. We illustrate some of the failure modes of TS models for DSR problems, and conclude that models built on DS principles may bear a huge potential also for advancing the TS prediction field.

## 1 Introduction

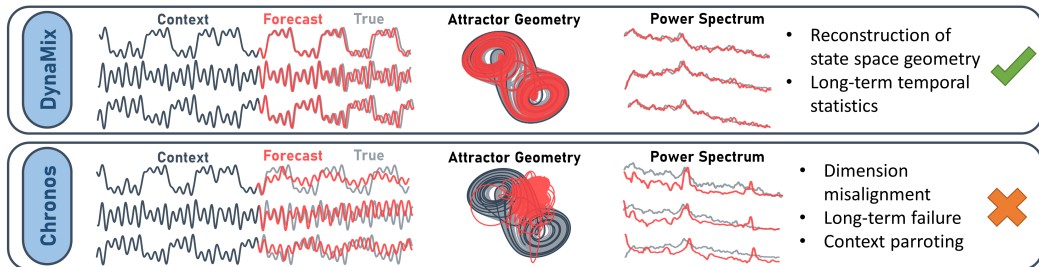

Figure 1: DynaMix achieves zero-shot DSR of attractor geometry and long-term temporal properties (power spectrum) from a short context signal while Chronos [2] fails.

Most real-world processes, from atmospheric phenomena and stock markets to brain activity or ecological networks, can be described as dynamical systems (DS) [58, 16, 88, 61]. Reconstructing

39th Conference on Neural Information Processing Systems (NeurIPS 2025).

these from observational data, called *dynamical systems reconstruction* (DSR), has been a long-standing challenge in scientific modeling [101, 24, 32]. DSR goes beyond conventional time series (TS) modeling, as we wish to have a *generative* model of the underlying process which exhibits long-term behavior with the same temporal and geometrical signatures (Fig. 1), i.e. the same invariant or 'climate' statistics [68, 72], as important in scientific applications. Achieving this usually requires special control-theoretic training techniques [60, 40] or loss objectives [71, 72, 43, 78] which accentuate the system's long-term dynamics. Numerous deep learning approaches for DSR, based on recurrent neural networks (RNNs; [87, 23, 90, 17, 7, 75, 40, 8, 72]), neural ODEs [18, 46, 1, 48], Koopman operators [15, 57, 65, 4, 62, 29, 93], or library-based methods [13, 55, 44, 20, 59], have been advanced over the years. However, all of these require purpose-training on the specific system observed and struggle to generalize beyond their training distribution [34].

Inspired by the strong in-context and zero-shot generalization abilities of LLMs [11, 28, 22, 19], there has been a push recently to develop models with likewise properties for the time series domain. Time series foundation models like Chronos [2, 3] or Mamba4Cast [6] are pre-trained on a huge database of real-world and artificial time series, and then tasked to forecast novel time series from which snippets are presented in-context, without any parameter fine-tuning. They are not built for DSR, however, and – as we show here – typically fail to properly capture a system's long-term behavior and the structure of its attractors (Fig. 1; [102]).

To address this gap, we introduce *DynaMix*, a novel mixture-of-experts model designed for DSR and zero-shot forecasting. By training across a diverse range of DS, DynaMix learns transferable representations, enabling it to forecast previously unseen systems without retraining, including their underlying attractor geometries and invariant statistics (Fig. 1). The core features of DynaMix are:

- **Accurate zero-shot DSR:** DynaMix achieves strong generalization across diverse DS, eliminating the need for fine-tuning while maintaining accuracy in reproducing attractor geometry and long-term statistics. *No other model tested achieved this.*

- **Multivariate information transfer:** Due to its multivariate architecture, the model efficiently captures dependencies among multiple system dimensions, enabling accurate reconstruction of their coupled dynamics. Further, it is neither bound to a specific dimensionality nor to a specific context length, but can flexibly adapt to other dimensions through specific embeddings.

- **Computational and parameter efficiency:** The model reaches high performance with a very lightweight architecture ($\approx$ 10k parameters) and small training corpus (34 systems), enabling orders of magnitude faster inference than other foundation models.

- **Interpretable dynamics composition:** The model provides insight into the dynamical composition of reconstructed systems, elucidating similarities between different DS.

- **Time series forecasting:** Beyond DSR, our model is applicable to general time series forecasting where we employ different embedding techniques to obtain data representations accessible to DSR.

We evaluate our model on multiple benchmark DS and real-world time series, demonstrating superior zero-shot generalization on DSR problems compared to current TS foundation models.

## 2   Related work

**Dynamical systems reconstruction (DSR)**   In DSR we seek to learn generative models from time series data that represent the underlying system dynamics with all its topological, geometrical, and temporal properties [24, 34, 72, 31, 14, 12], in that sense providing an approximation to the underlying system's governing equations. By definition, such a model should not only render viable short-term forecasts, but also reproduce the long-term evolution of the DS it has been trained on, both in state space and in the time domain. Methods approaching this goal have been founded on predefined function libraries, such as Sparse Identification of Nonlinear Dynamics (SINDy; [13, 55, 20, 44, 59]), on reservoir computers [69, 71, 72], neural ODEs [18, 46, 1, 48], Koopman operators [15, 57, 65, 4, 62, 29, 93], or different types of RNNs [87, 23, 90, 17, 7, 75, 40, 8]. More important than the architecture itself seems to be their proper training in order to ensure long-term (invariant) statistics of the underlying system are met: Methods like sparse [60, 7] or generalized [40]

teacher forcing allow RNN-generated trajectories to 'explore the future' whilst training, yet keep loss gradients in check. Alternatively, long-term statistics based on the observed system's Lyapunov spectrum, fractal geometry, or invariant measures may be added to regularize the loss [71, 72, 43, 78], but require to compute these properties from the data first. Despite these advances, out-of-domain generalization remains a key challenge in DSR [34]. Meta- and hierarchical learning models, trained across many DS simultaneously, have recently been designed as steps toward foundation models for DSR [10, 64, 47, 100], but still require parameter fine-tuning and lack in-context inference capabilities.

**Time series foundation models** Large language models (LLMs) exhibit an impressive ability to infer patterns from prompts (context) and generalize to novel situations without retraining [11, 28, 22, 19], although the underlying reasons for this are still a matter of debate [53, 92, 99]. This inspired the development of general-purpose time series (TS) foundation models which could accurately forecast time series from a short segment provided 'in-context' [2, 27], without the need of task-specific fine-tuning [21]. One idea is to simply use pretrained LLMs directly as TS forecasters [35, 73, 81, 95, 85], but the inherent differences between textual and many temporal data, such as continuity, present significant challenges [84]. To overcome these, transformer-based architectures, similar in design to LLMs, such as Chronos [2, 3] or TimesFM [21], were specifically pretrained on a large corpus of time series data. Promising zero-shot forecasting capabilities have also been achieved with alternative architectural designs, such as Tiny Time Mixers [27] or state-space models like Mamba [6].

The success of TS foundation models raised hope these could also be utilized for zero-shot DSR, but the – to our knowledge – so far only previous study on this, based on Chronos, had mixed outcomes and fell short of a full DSR evaluation [102]: Successful DSR, if present at all, heavily depended on the initial conditions, and mechanisms such as context parroting visually gave the wrong illusion that features of the dynamics had been captured. In fact, here we show that existing TS foundation models are generally *not* capable of producing valid DSRs (Fig. 1), and highlight some of their failure modes. This is in contrast to the zero-shot DSR foundation architecture we develop here, based on models [8] and training algorithms [60] successful in DSR.

# 3 Methods

## 3.1 Model architecture

To enable zero-shot reconstruction of novel DS, we develop a specific mixture-of-experts (MoE) architecture that can be pretrained across many diverse DS (Fig. 10), with different *experts* possibly specializing in different dynamical regimes. As a SOTA DSR base model for the experts, we leverage a recent parameter-friendly RNN, which allows for highly efficient DSR training and is designed to yield topologically parsimonious and interpretable representations of DS it is being trained on, the *Almost-Linear RNN* (AL-RNN; [8]):

$$\boldsymbol{z}_t = \boldsymbol{A}\boldsymbol{z}_{t-1} + \boldsymbol{W}\Phi^*(\boldsymbol{z}_{t-1}) + \boldsymbol{h} \ . \tag{1}$$

The model describes the evolution of an $M$-dimensional latent process $\boldsymbol{z}_t \in \mathbb{R}^M$, with linear self-connections $\boldsymbol{A} \in \text{diag}(\mathbb{R}^M)$, weight matrix $\boldsymbol{W} \in \mathbb{R}^{M \times M}$, bias term $\boldsymbol{h} \in \mathbb{R}^M$, and $\Phi^*(\boldsymbol{z}_t)$ defined as

$$\Phi^*(\boldsymbol{z}_t) := [z_{1,t}, \cdots, z_{M-P,t}, \max(0, z_{M-P+1,t}), \cdots, \max(0, z_{M,t})]^T \ , \tag{2}$$

i.e. with a ReLU nonlinearity on only $P << M$ out of the $M$ AL-RNN units. The first $N$ units are interpreted as the network's readouts and provide the predicted observations, $\hat{\boldsymbol{X}} = \{\hat{\boldsymbol{x}}_t = \boldsymbol{z}_{1:N,t}\} \in \mathbb{R}^{N \times T}$, where $N$ is the observation dimension.

The selection of AL-RNN experts is achieved through a gating network (Fig. 2), which receives as inputs the (generally multivariate) context time series $\boldsymbol{C} = \{\boldsymbol{c}_t\} \in \mathbb{R}^{N \times T_C}$ as well as the current latent state $\boldsymbol{z}_t$. Both are passed into a state attention mechanism defined as

$$\boldsymbol{w}_t^{att} = \sigma \left( \frac{\left| \boldsymbol{C} - (\boldsymbol{D}\boldsymbol{z}_t + \boldsymbol{\epsilon}) \, \boldsymbol{1}_{T_C}^\top \right|^\top \boldsymbol{1}_N}{\tau_{\text{att}}} \right) \in \mathbb{R}^{T_C} \ , \tag{3}$$

where $\boldsymbol{D} \in \mathbb{R}^{N \times M}$ is a learnable matrix which maps the latent state into observation space, $\boldsymbol{\epsilon} \sim \mathcal{N}(0, \boldsymbol{\Sigma})$ exploration noise with a learnable covariance $\boldsymbol{\Sigma}$, $\tau_{\text{att}}$ a learnable temperature parameter,

$\mathbf{1}_{\{T_C, N\}}$ column vectors of ones of length $T_C, N$, respectively, and $\sigma(\cdot)$ the softmax returning normalized weights. Hence, this mechanism computes attention weights based on some distance between projected latent states and actual context observations. At the same time, the context signal $\boldsymbol{C}$ is processed by a CNN, yielding temporal features $\tilde{\boldsymbol{C}} \in \mathbb{R}^{N \times T_C}$. These features are then weighted with the attention weights $\boldsymbol{w}_t^{att}$, and together with the current state $\boldsymbol{z}_t$ further processed by a multi-layer perceptron (MLP) followed by a softmax, yielding a set of expert weights at any time $t$:

$$\boldsymbol{w}_t^{exp} = \sigma \left( \frac{\mathrm{MLP}(\tilde{\boldsymbol{C}}\boldsymbol{w}_t^{att}, \boldsymbol{z}_t)}{\tau_{\exp}} \right) \in \mathbb{R}^J \ , \tag{4}$$

where $\tau_{\exp}$ is another learnable temperature parameter. The forward-iterated latent states $\boldsymbol{z}_{t+1}^j$ of the individual experts $j \in \{1, ..., J\}$ are then weighted by $\boldsymbol{w}_t^{exp}$ to yield the next time step prediction $\boldsymbol{z}_{t+1} = \sum_{j=1}^{J} w_{j,t}^{exp} \cdot \boldsymbol{z}_{t+1}^j$. Fig. 2 illustrates the whole approach. Note that one key advantage of this compared to other TS foundation architectures is that the *context length is flexible by design*, i.e. the CNN and the attention mechanism may take as input a context signal of arbitrary length.

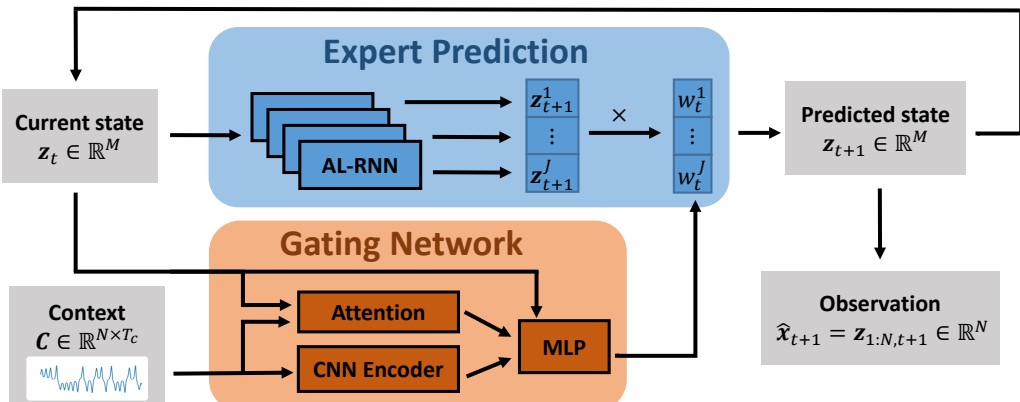

Figure 2: Illustration of the DynaMix architecture. At test time, only a context signal $\boldsymbol{C}$ is provided and guides the selection of experts to yield arbitrarily long forward predictions of the dynamics.

### 3.2 Model training

For training, a collection of just 34 different chaotic and cyclic DS is used [30], see Appx. A.2 for full specification and Fig. 10 for illustration. From each, multivariate time series $\boldsymbol{X} \in \mathbb{R}^{N \times T}$ are simulated ($\approx 6 \times 10^5$ in total) and then standardized dimension-wise (likewise for the test set), of which the first $T_C < T$ column entries are defined as the context signal $\boldsymbol{C} = \boldsymbol{X}_{1:T_C}$. The experts are initialized at $t_0 = T_C - \Delta t + 1$ and forward-iterated until time $T$, such that they overlap for a period $\Delta t$ with the context. This is to ensure that the model learns to optimally utilize the context for generalization. A MSE loss is then computed across ground truth, $\boldsymbol{X}_{T_C - \Delta t + 1 : T}$, and respective model-generated time series $\hat{\boldsymbol{X}}$. The model is trained by *sparse teacher forcing* (STF; [7, 60]), a control-theoretic technique specifically designed for DSR. STF replaces part of the forward-iterated latent states $\boldsymbol{z}_t$ by data-inferred states $\hat{\boldsymbol{z}}_t$ at optimally chosen intervals $\tau$, to avoid exploding gradients even for chaotic systems while enabling the DSR model "to explore the future" (see [60] for detailed theoretical motivation). As shown in Fig. 26, STF with optimal $\tau$ is essential for achieving good zero-shot DSR results. Note that STF is *only used for training* and turned off at test time.

### 3.3 DSR evaluation & time series forecasting

We used two established measures to assess DSR quality [49, 7, 40, 97, 102, 72, 60, 66, 31]: First, to quantify agreement in *state space (attractor) geometry*, we employed a Kullback-Leibler divergence defined across *space*, $D_{stsp}$ (see Appx. A.1 for details). For assessing agreement in *long-term temporal properties*, we used the Hellinger distance $D_H$ defined across power spectra of the true and model-generated trajectories. Crucially, to properly assess the *long-term* dynamics, both these measures were evaluated in the limit of large $T = 10,000$. In addition, a *short-term*, $n$-step-ahead

mean absolute prediction error (MAE) was evaluated. As has been repeatedly emphasized in the statistical and DSR literature [97, 49, 7, 60, 70, 31], prediction errors defined on time series are often only sensible on *short* time scales, because of the well known exponentially fast divergence of trajectories in chaotic DS (see Fig. 24).

Empirically observed time series are often just one-dimensional ($N^* = 1$), but come from an inherently much higher-dimensional underlying DS. To work efficiently in a DSR setting, the context signal should represent the underlying DS as good as possible, for which it commonly needs to be lifted into a higher-dimensional space where sets explored by the dynamics ideally become diffeomorphic to those in the system's true state space. Temporal delay embedding is the most popular technique to achieve this [82, 76]. For a $1d$ time series $\{x_t\}$, a $d$-dimensional embedding can be defined as [50]

$$\boldsymbol{x}_t^{emb} = (x_t, x_{t-\tau_1}, ..., x_{t-\tau_{d-1}}) \, ,\tag{5}$$

where the $\tau_i$ are time lags, commonly estimated from the autocorrelation of the time series. Alternatively, if the delay embedding would become too large ($d > N$), we augment the empirically observed time series by a variant of the positional encoding common in transformers, defined here as

$$\boldsymbol{x}_t^{emb} = \left( x_t, \sin\left(\frac{2\pi t}{\tau} + \phi_1\right), ..., \sin\left(\frac{2\pi t}{\tau} + \phi_{N-1}\right) \right)\tag{6}$$

where $\tau := \arg\max_{\tau > \tau_{min}} \mathbb{E}[x_t x_{t+\tau}]$ is given – provided it exceeds a threshold $\tau_{min}$ – by the maximal autocorrelation, and a random phase $\phi_i \in \left[0, \frac{\pi}{2}\right]$ is assigned to each dimension.

## 4 Results

### 4.1 Zero-shot DSR

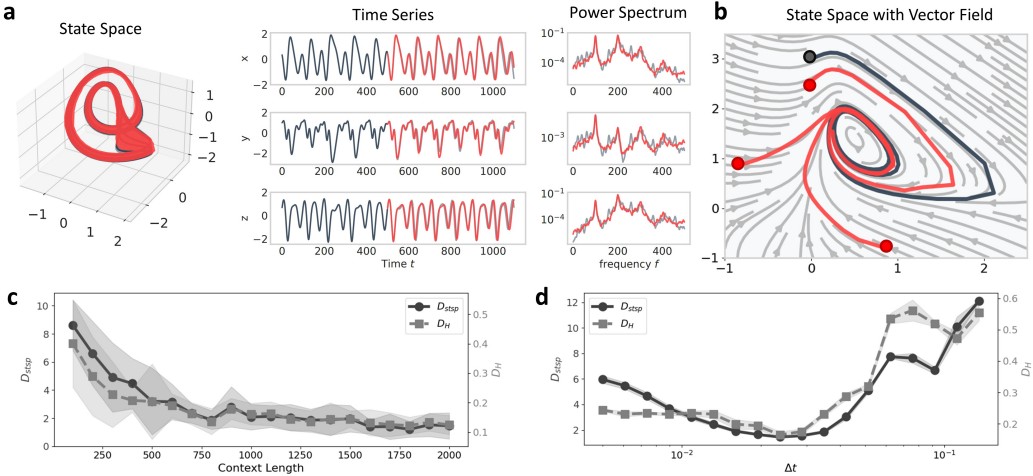

Figure 3: **a**) DynaMix zero-shot DSR (red) compared to ground truth (lightgray) using a 500-step context (darkgray) for the Sprott M system. **b**) Zero-shot forecasts for the Selkov DS (true vector field in lightgray) from different initial conditions (red) outside the context range (darkgray). **c**) DSR quality as a function of context length for Lorenz-63. **d**) DSR quality as a function of the temporal resolution $\Delta t$ of the context signal. Error bands = STD

Figs. 1 (top) & 3**a** present example zero-shot DSRs of new chaotic DS, which were not part of DynaMix' training domain, from just a provided context signal $C$ (see Figs. 11-15 for further examples). Despite the inherent challenge of predicting chaotic systems, where small differences in initial conditions lead to rapidly diverging trajectories (Fig. 24), DynaMix successfully captures the system's temporal & geometrical behavior, i.e. accurately reconstructs the underlying state space and faithfully recovers the true system's power spectrum. Fig. 3**b** highlights further aspects of DynaMix' reconstructions: Even though not trained on any $2d$ systems, the model successfully recreates the limit cycle from the $2d$ Selkov system [79]. Notably, the model even *generalizes to new initial*

*conditions outside the scope of the context data*, i.e. correctly infers properties of the state space beyond the short context trajectory (see Fig. 22 for further examples). Likewise, although trained only on $3d$ DS, DynaMix also successfully generalizes to *higher-dimensional* DS, as illustrated for the $6d$ Lorenz-96 system [56] in Fig. 34 (see Appx. A.6 for details on the setup and comparisons to other models). As further shown in Fig. 3c, the context length necessary to achieve good DSR quickly converges, with as few as 500 time steps often sufficient. Fig. 3d illustrates that our approach also works across a surprisingly wide range (about an order of magnitude) of temporal resolutions at which the underlying DS was sampled, confirming its robustness w.r.t. sampling frequency.

We next compared the zero-shot, out-of-domain DSR performance of DynaMix to that of a number of recent TS foundation models, including various versions of Chronos [2], Chronos-2 [3], Panda [52], Mamba4Cast [6], TimesFM [21], and Tiny Time Mixers [27], using the long-term statistics $D_{stsp}$ and $D_H$ for evaluation (see Sect. A.1). As seen in Fig. 4a-b, DynaMix clearly and significantly outperforms all of these. In terms of *short-term* (10-step ahead) forecasts, for which TS foundation models (unlike DynaMix) are optimized, DynaMix performs on par with the strongest competitors in our batch, as shown in Fig. 4c. This is especially surprising when compared to Panda, which included variations of our test set DS in its training repertoire, biasing the evaluation in its favor. Table 9 assembles further results on performance comparisons, while Figs. 16-21 provide specific examples. Moreover, a significant limitation of particularly the large Chronos models with millions of parameters is their extreme inefficiency in both inference time and computational costs, as compared in Fig. 5. In fact, their inference time can even exceed that of a custom-trained model. In contrast, DynaMix provides a highly efficient alternative, offering superior DSR performance while using orders of magnitude less parameters ($\approx 10k$ in total) and computation time.

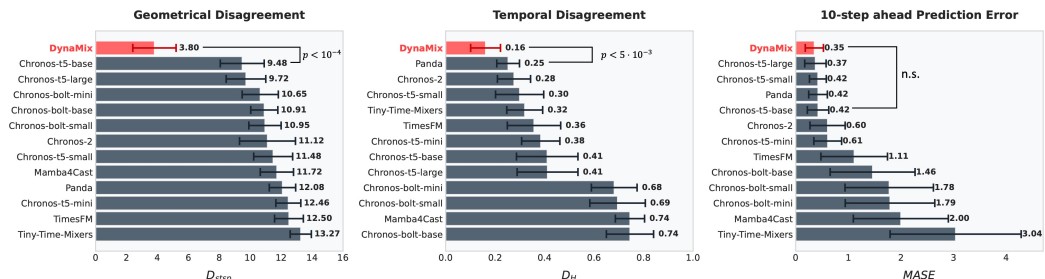

Figure 4: Zero-shot DSR performance across all 54 test set DS for DynaMix and various TS foundation models for context length $T_C = 2000$ (see Fig. 23 for results with $T_C = 512$). Median$\pm$MAD of $D_{stsp}$ (left, geometrical disagreement), $D_H$ (middle, temporal disagreement), and MASE (right, short-term prediction error). Statistical testing based on Wilcoxon signed-rank tests.

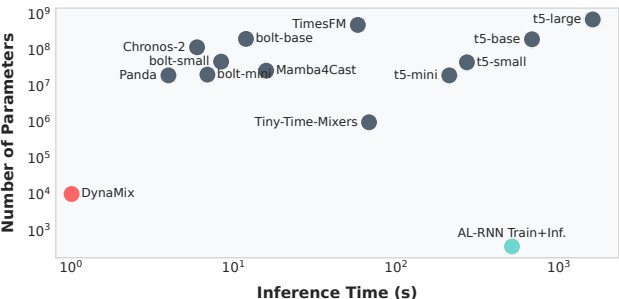

Figure 5: Number of model parameters vs. inference times for zero-shot generation of Lorenz-63 DS with 10000 forecasting steps. All models were provided the exact same context and run on the same hardware. Note the log-scale on the $x/y$-axis. For comparison, also the time needed for training and inference of a custom-trained AL-RNN is shown (turquoise).

To put DynaMix' performance further in context, we also compared it to a couple of *custom-trained* DSR models, i.e. SOTA DSR models explicitly trained on the context data, incl. the plain AL-RNN

trained by STF [8], Neural ODEs [38, 18], and reservoir computers which are commonly used for DSR [72] (see Appx. A.7 for details). Despite the custom trained models' unduly advantage, performing – unlike DynaMix – only *in-domain* generalization [38], DynaMix keeps up with, and in some cases even outperforms, them (Appx. Tables 6–8).

## 4.2    Reasons for the failure of TS foundation models on DSR

Why do TS foundation models perform so poorly on DSR problems? Essentially, they face three key issues: First, most TS foundation models cannot efficiently deal with *multivariate* time series, but treat dimensions independently. However, in a nonlinear DS all variables are usually coupled, and their joint evolution is actually *defining* for the system's (long-term) dynamics and attractor states. Thus, ignoring this, as illustrated in Fig. 6**a**, results in poor long-term accuracy (see also Figs. 16-21). This highlights a fundamental limitation: Univariate approaches are *inherently* insufficient for reconstructing multivariate, interacting DS. Second, TS foundation models are *trained for short-term prediction, but not for DSR*, providing another reason why they fail to reconstruct the correct long-term behavior. As illustrated in Figs. 1 & 6**b**, in the longer run they often converge to fixed points or simple cycles where the true dynamics is chaotic, not reflecting the context anymore. Third, *context parroting*, a phenomenon originally identified in LLMs [5], has also been observed in TS foundation models [102], where they tend to repetitively reproduce the exact input pattern rather than dynamically adapting to the system's true evolution. Thus, TS foundation models simply produce oscillations, where the true dynamics is chaotic and therefore by definition *irregular* and *non-repeating* (Fig. 6**c**). This is illustrated in more detail in Fig. 6**d** for Chronos-t5-base, the best of the TS foundation models according to $D_{stsp}$. It is particularly evident in the power spectrum (Fig. 6**d** right), where Chronos' sharp, narrow peaks wrongly suggest periodicity, while the true spectrum is usually rather broad and smeared out for chaotic systems, as the one exhibited by DynaMix. We further quantified this by calculating from the forecast time series the maximum Lyapunov exponent [60], a measure for the

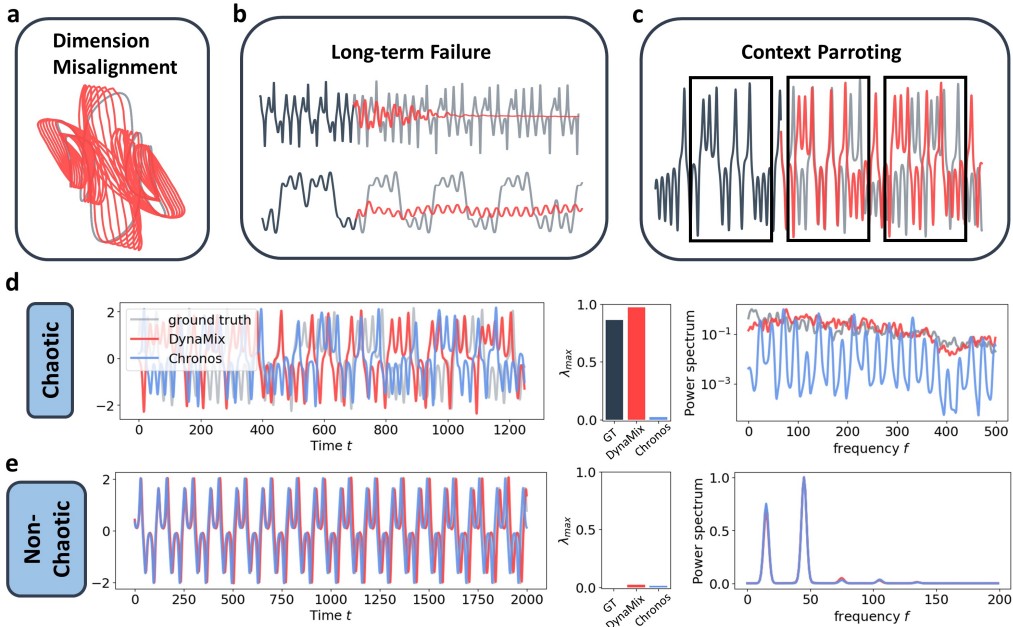

Figure 6: Common problems of TS foundation models (red: forecast, darkgray: context, lightgray: ground truth). **a**) Dimensions are dynamically decoupled. **b**) Long-term forecasts often converge to simple fixed points or cycles unrelated to the true dynamics. **c**) Context parroting covers up the true dynamics. **d**) Example forecasts for the chaotic Lorenz-63 system with DynaMix and Chronos-t5-base. Chronos' context parroting results in cyclic repetition of a fixed pattern, thus *inherently* missing the true aperiodic, chaotic dynamics. This is evident in the close-to-0 max. Lyapunov exponent (center) and peaked power spectrum (right). **e**) For comparison, if the underlying behavior is truly cyclic with prominent peaks in the power spectrum, both Chronos and DynaMix are able to capture it.

exponential divergence rate of trajectories (see Appx. A.1). For Chronos, the value $\lambda_{Chronos} \approx 0.02$ close to zero confirms the nearly periodic behavior, while DynaMix produces exponentially diverging trajectories with $\lambda_{DynaMix} \approx 0.96$ close to the ground truth value of $\lambda_{Lorenz63} \approx 0.87$. Hence, unlike DynaMix, Chronos does not reconstruct the underlying dynamics but just repeats the context. On the other hand, if the underlying dynamics truly *is cyclic*, DynaMix settles into a cyclic pattern as well, exhibiting no intrinsic bias, as illustrated in Fig. 6e.

## 4.3 Dynamical similarity

DynaMix' mixture-of-experts design produces a natural similarity measure for comparing the dynamics of different systems, by evaluating the relative contribution of each of the experts to the forecast dynamics. As shown in Fig. 7a (bottom), different expert mixtures specialize in specific dynamical patterns or regimes. Hence, we can utilize the expert usage $\boldsymbol{w}^{exp}$ across the generated time series to construct a similarity metric $S = \frac{1}{1+W_2^2} \in [0, 1]$, based on the squared 2-Wasserstein distance $W_2^2$ between the standardized expert weights $\boldsymbol{w}^{exp}$ of any two systems.

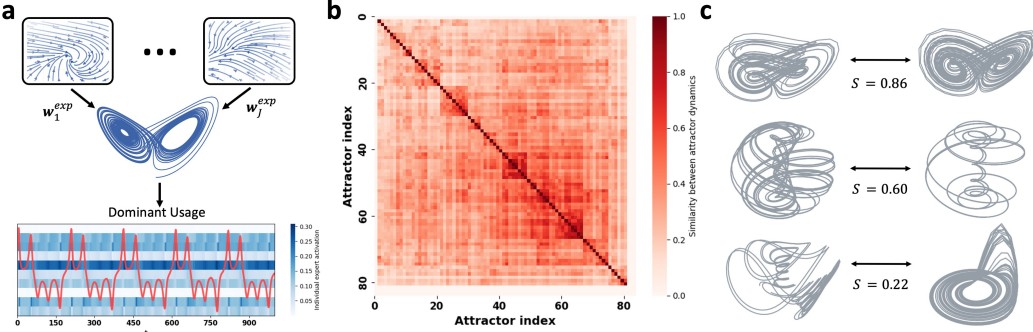

Figure 7: **a**) Top: Vector fields of different experts which in combination determine the attractor. Bottom plot illustrates the expert usage across time. **b**) Similarity matrix based on $S$ among all $84$ DS in the dataset, sorted via hierarchical clustering. Structure in the matrix indicates there are clusters of DS with high dynamical similarity. **c**) Examples of two pairs of attractors classified as similar (top), and one dissimilar pair (bottom).

## 4.4 Time series forecasting from a DS perspective

Table 1: Performance comparison on empirical time series in terms of geometrical divergence ($D_{\text{stsp}}$), long-term temporal distance ($D_H$), and forecast error (MAE). Best in red, second-best in blue.

| System | DynaMix | | | Chronos-t5-base | | | Mamba4Cast | | | TTM | | | TimesFM | | |
|---|---|---|---|---|---|---|---|---|---|---|---|---|---|---|---|
| | $D_{\text{stsp}}$ | $D_H$ | MAE | $D_{\text{stsp}}$ | $D_H$ | MAE | $D_{\text{stsp}}$ | $D_H$ | MAE | $D_{\text{stsp}}$ | $D_H$ | MAE | $D_{\text{stsp}}$ | $D_H$ | MAE |
| Partially obs. DS | 0.02 | 0.25 | 0.39 | 0.02 | 0.32 | 0.39 | 8.35 | 0.35 | 0.76 | 4.76 | 0.31 | 0.90 | 7.30 | 0.36 | 0.71 |
| ETTh1 | 0.22 | 0.08 | 4.92 | 0.23 | 0.09 | 3.22 | 0.80 | 0.10 | 3.63 | 6.36 | 0.09 | 4.64 | 0.24 | 0.10 | 3.04 |
| Traffic | 0.81 | 0.21 | 6.93 | 0.40 | 0.10 | 3.16 | 1.59 | 0.34 | 7.68 | 1.36 | 0.25 | 5.25 | 0.43 | 0.11 | 3.66 |
| Cloud Requests | 0.27 | 0.14 | 161.58 | 1.86 | 0.31 | 557.35 | 5.89 | 0.16 | 282.95 | 1.31 | 0.25 | 147.54 | 0.33 | 0.29 | 239.31 |
| Weather (Temp.) | 0.66 | 0.09 | 2.24 | 4.78 | 0.21 | 4.14 | 6.86 | 0.57 | 9.39 | 1.28 | 0.12 | 1.92 | 6.86 | 0.13 | 3.31 |
| Weather (Press.) | 0.39 | 0.19 | 9.03 | 6.68 | 0.23 | 9.05 | 7.07 | 0.52 | 8.60 | 7.31 | 0.26 | 10.03 | 9.59 | 0.60 | 8.41 |
| Human fMRI | 0.17 | 0.09 | 0.45 | 5.78 | 0.12 | 0.85 | 7.97 | 0.11 | 1.35 | 4.87 | 0.39 | 1.54 | 6.19 | 0.21 | 1.11 |
| Human EEG | 0.79 | 0.23 | 1.07 | 9.45 | 0.24 | 1.10 | 5.87 | 0.61 | 1.12 | 10.36 | 0.26 | 1.25 | 10.07 | 0.27 | 1.32 |

We next tested and compared DynaMix on a variety of real-world datasets often used to probe time series models. As discussed in Sect. 3.3, for DynaMix we used an embedding of the typically $1d$ empirical time series, to better reflect the underlying DS properties (positional embedding for all empirical time series, akin to what is done in TS foundation models, and delay embedding for the Lorenz-63). Fig. 8a confirms for a $1d$ time series from the $3d$ chaotic Lorenz-63 system that this works well and DynaMix, in contrast to all other models, is still able to reconstruct the chaotic behavior. When applied to real-world time series, such as traffic, weather, or cloud request data,

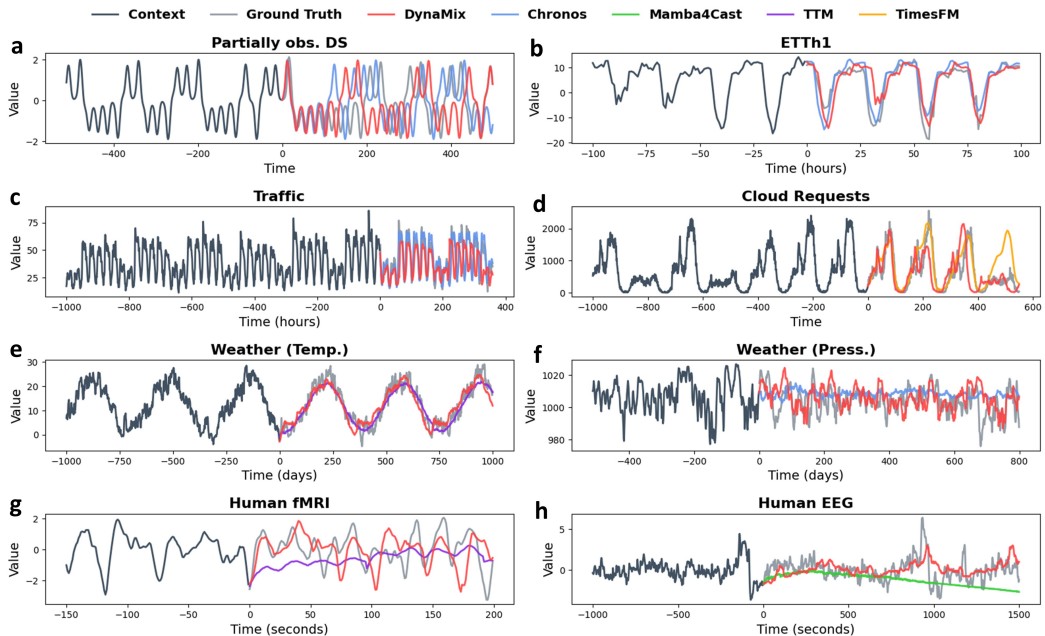

Figure 8: Comparison of DynaMix (red) to strongest competitor in terms of $D_{stsp}$ (other colors) on zero-shot forecasts of various empirical time series (see Fig. 27-30 for comparisons to all other TS foundation models): Forecasts of **a**) partially (1d) observed Lorenz-63 DS, **b**) electricity transformer temperature, **c**) hourly car traffic data with weekly cycle, **d**) Huawei cloud requests, **e**) soil temperature development, **f**) air pressure, **g**) human functional magnetic resonance imaging (fMRI), **h**) human electroencephalogram (EEG).

where previous TS foundation models have shown inconsistent results [86], our model produces accurate forecasts which reflect the dynamics well (Fig. 8**b**-**f**, see Appx. A.2 for further details on the datasets). Our model seems to excel particularly on human physiological signals, such as functional magnetic resonance imaging (fMRI) or electroencephalographic (EEG) recordings, where all other models essentially produce meaningless dynamics, at best capturing the mean trend (Fig. 8**g**,**h**).

Table 1 quantitatively confirms that DynaMix mostly outperforms the other models tested on the DSR measures. Surprisingly, it often even outperforms them in terms of the forecasting error MAE, *although its training corpus consisted purely of the simulated DS shown in Fig. 10*, i.e. did not include *any* empirical data at all, let alone traffic, cloud usage, electricity, or temperature data! This is in stark contrast to the other TS foundation models tested here, which mostly had examples of such empirical data in their training corpus and hence might be expected to have an edge over DynaMix. Many real-world data, like weather, human fMRI, or EEG data, are known to bear signatures of deterministic chaos [60, 51], potentially at least partly explaining DynaMix' strongly competitive forecasts, although it has never seen such type of data in training.

### 4.5 Which ingredients are most crucial for DynaMix' performance? Ablation studies

Which are the components of the DynaMix framework contributing most to its success in zero-shot out-of-domain generalization? Training by STF (Fig. 26), a sufficiently diverse training corpus, a sufficient number of experts (Fig. 25), the attention mechanism, and context preprocessing by the CNN, all appear vital to its strong performance (Fig. 9). Replacing the final gating step (the MLP) by a simple linear operation and the exact expert model *within the class of piecewise linear RNNs*, however, had less of an impact. Similar results were obtained when the AL-RNN was swapped for the clipped shallow PLRNN [40], and a probabilistic version of the AL-RNN (adding a noise term $z_t = F(z_{t-1}) + \eta, \eta \sim \mathcal{N}(0, \Gamma)$) even slightly improved long-term performance, albeit at the cost of some short-term prediction accuracy. Using as experts LSTMs [42], vanilla RNNs or reservoir computers [67] instead led to decay in performance, in line with previous results establishing PLRNNs as SOTA DSR models which allow for efficient training by STF [7, 40, 8]. Hence, it appears that the

whole 'DSR package' consisting of 1) architectural choices, 2) a training algorithm specialized for DSR, and 3) properties of the DS training corpus, is crucial to DynaMix' success.

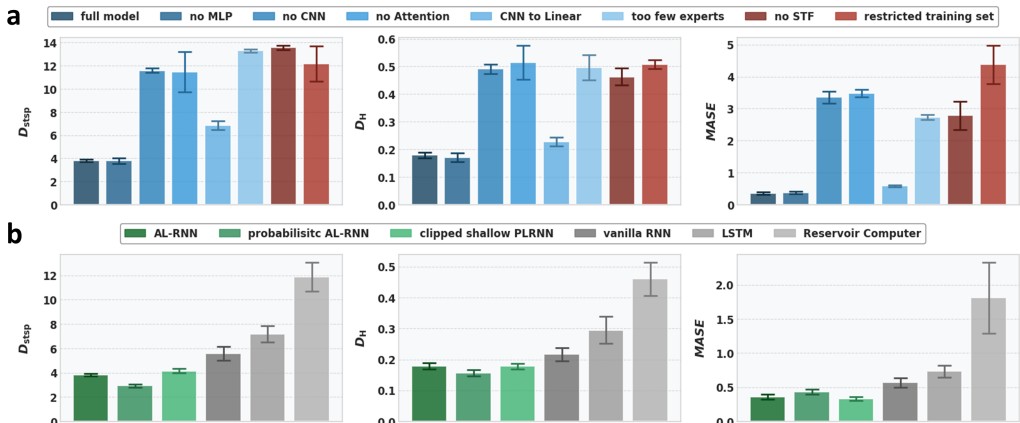

Figure 9: Ablation studies. **a**) Effect of removing or altering components of the gating network (no MLP, no CNN, no Attention, CNN to linear layer), including too few experts ($J < 5$), replacing STF by standard BPTT training, restricting the training corpus to Lorenz-type systems only. **b**) Effect of replacing the AL-RNN by other RNN models. Error bars = STD

## 5  Conclusions

Here we introduce the first DSR foundation model that achieves zero-shot reconstructions of novel DS, accurately reproducing their long-term statistics, just from a provided context signal without any retraining or fine-tuning. It hugely outperforms current TS foundation models like Chronos or Panda in terms of DSR quality, although much more lightweight with only a fraction (0.1%) of the number of parameters. Consequently, DynaMix features inference times about an order of magnitude faster than the closest competitor. Most surprisingly, it even outperforms current TS foundation models on empirical data of types it has – in contrast to competitors – never seen in training, often producing also better short-term forecasts. All of this is achieved with a fairly narrow training corpus consisting mainly of $3d$ synthetic data from chaotic systems, which suggests there may be considerable room for further improvement by extending DynaMix' training to include, e.g., various empirical time series.

One important take-home from this work therefore is that foundation models built based on principles of DS theory may be able to profoundly improve performance of current zero-shot TS forecasters. Most (if not all) empirically observed time series come from some underlying DS, and acknowledging this fact in model training and construction may help to advance the field. Besides DynaMix' specific DSR architecture, control-theoretically motivated training techniques, and embedding principles, also the fact that it has seen many *chaotic* systems in training may play a role. Most complex DS in nature [16, 88, 61], engineering [80], and society [58, 94] are likely chaotic. Moreover, many chaotic attractors, like the Lorenz-63, feature a skeleton of infinitely many (unstable) periodic orbits of all possible periods [37], thus *inherently expressing a wide spectrum of temporal patterns*.

**Limitations**   For now, we have mainly focused on rather stationary time series (but see Fig. 8**b**). Changes in statistical properties over time, tipping points, or widely differing time scales in the data, impose severe difficulties for zero-shot forecasting, as illustrated in Fig. 31**a**,**b**, which also plague other TS foundation models. This may partly be amended by explicitly including such types of non-stationary and multiscale DS in the training corpus. Another potential solution may be adding explicit filtering and decomposition modules, similar as in Auto- [98] or FEDformers [104]. A proof of concept of this idea is provided in Appx. A.5. Incorrect embeddings, which may not be immediately obvious, can also lead to forecast failures (Fig. 31**c**). Finally, continuous-time versions of the experts which can deal with sampling at irregular time points would be another fruitful direction.

Code available at https://github.com/DurstewitzLab/DynaMix-julia (Julia), https://github.com/DurstewitzLab/DynaMix-python (Python).

## Acknowledgements

This work was funded by the German Research Foundation (DFG) within Germany's Excellence Strategy EXC 2181/1 – 390900948 (STRUCTURES), and through DFG individual grant Du 354/15-1 to DD.

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

# A  Appendix

## A.1  Methodological details

**Training method**  For training our model, we used a variant of sparse teacher forcing (STF), a control-theoretic method for DSR, with the theoretical background and rationale developed in [60]. STF and related training methods [7, 9, 40, 60, 25] circumvent the exploding gradient problem when training on chaotic systems, while still enabling the DSR model to sufficiently forward-iterate trajectories into the future to capture long-term properties (see [60] for details), and produces state-of-the-art results for DSR [8, 10]. In STF, current latent states states $z_t$ are replaced with states inferred from the data at fixed intervals $\tau$ by (pseudo-)inversion of the decoder model, $\tilde{z}_t = g^{-1}(x_t)$, thus recalibrating the trajectory at times ideally chosen based on the system's Lyapunov spectrum [60] (or simply treating $\tau$ as a hyper-parameter). In our case of an identity mapping from a latent subspace to the observations, $\hat{x}_t = \mathcal{I}_{(N \times M)} z_t$, this inversion becomes trivial, where $N$ is the number of observations (and thus readout neurons) and $\mathcal{I}_{(N \times M)}$ an $N \times M$ matrix with $\mathcal{I}_{(N \times M), rr} = 1$ for $r \leq N$ and zeros else. Thus, during training, we force

$$z_{t+1} = \begin{cases} F_{\boldsymbol{\theta}}(\tilde{z}_t, \boldsymbol{C}) & \text{if } t \in \mathcal{T} = \{l\tau + 1\}_{l \in \mathbb{N}_0} \\ F_{\boldsymbol{\theta}}(z_t, \boldsymbol{C}) & \text{otherwise,} \end{cases} \tag{7}$$

where $\tilde{z}_t = (x_t, z_{N+1:M,t})^T$, $F_{\boldsymbol{\theta}}$ is the mixture of experts using the context $\boldsymbol{C}$, and $\tau = 10$ here (see Fig. 26). Importantly, STF is *only used for training* the model, applied after calculating the loss, and is turned off at test time.

For the loss, we simply use the standard mean squared error (MSE) between model predictions $\hat{\boldsymbol{X}}$ and ground truth observations $\boldsymbol{X}$,

$$\mathcal{L}_{MSE}(\hat{\boldsymbol{X}}, \boldsymbol{X}) = \frac{1}{N \cdot (T - T_C + \Delta t)} \sum_{t=T_C - \Delta t + 1}^{T} \|\hat{x}_t - x_t\|_2^2. \tag{8}$$

To this we add a regularization term meant to encourage the model to explore a wider region of state space based on the context, by enhancing the variance $\boldsymbol{\Sigma}$ of exploration noise in the state attention:

$$\mathcal{L}_{reg} = \lambda \frac{1}{N} \sum_i \exp\left(- |\Sigma_{ii}| / c\right), \tag{9}$$

where we chose $\lambda = 0.1$ and $c = 0.01$. Rectified adaptive moment estimation (RADAM) [54] was employed as the optimizer, with $L = 50$ batches of $S_B = 16$ sequences per epoch, each of length $T = 550$, and 2000 epochs in total. We used a learning rate exponentially decaying from $\eta_{\text{start}} = 5 \cdot 10^{-3}$ to $\eta_{\text{end}} = 10^{-5}$. The context length used in training was set to $T_C = 500$, and the window of overlap with the model-generated time series to $\Delta t = 50$, see Sect. 3.2. Training was performed on a single CPU (18-Core Xeon Gold 6254), and a single epoch took 30 seconds depending on sequence length and model size. At test time, no retraining or fine-tuning is performed, but utilizing the context $\boldsymbol{C}$ the model just forward-iterates from the last context time step. Hyperparameters for training were partly selected according to previous results in [8, 7], and partly a few different parameter settings were tested, namely $\lambda = \{0.0, 0.1, 0.2\}$, $\Delta t = \{0, 50, 80\}$, and $T_C = \{250, 500\}$, as extensive grid search was not necessary to obtain a well-performing model.

**Model details**  We use $J = 10$ AL-RNN experts for our model. Each expert has a latent dimension of $M = 30$, of which $P = 2$ are rectified-linear units (ReLUs). As suggested in [8], we only use the first $N$ of the $M - P$ linear units for the readout. We followed the initialization protocol in [7, 83], drawing $\boldsymbol{W}$ from a Gaussian with mean $\boldsymbol{0}$ and $\boldsymbol{\sigma} = 0.01\mathbb{1}$, setting $\boldsymbol{h} = \boldsymbol{0}$, and $\boldsymbol{A}$ to the diagonal of a normalized positive-definite random matrix. The initial latent state was estimated as

$$z_1 = \begin{bmatrix} x_1 \\ \boldsymbol{L} x_1 \end{bmatrix}, \tag{10}$$

where $\boldsymbol{L} \in \mathbb{R}^{(M-N) \times N}$ is jointly learned with other model parameters.

The gating network is implemented using a single-layer CNN with three channels and a kernel size of 2, stride of 1, and zero padding, with the identity as activation function. The MLP, which takes

as inputs the concatenated weighted CNN outputs and current latent state, consists of two layers with ReLU activation. We initialize the temperature weights $\tau_{\text{att}}$ and $\tau_{\text{exp}}$ to 0.1, the covariance matrix $\boldsymbol{\Sigma} = 0.05 \cdot \mathbb{1}$, matrix $\boldsymbol{D} = \mathcal{I}_{(N \times M)}$, and draw all other CNN and MLP matrices from a Gaussian with standard deviation 0.01. Model hyperparameters were selected according to previous results in [8] and by probing a few relevant parameter settings, namely $J = \{1, 3, 5, 10, 20, 30\}$ (Fig. 25), #CNN-layers $= \{1, 2\}$, CNN-kernel-size $= \{2, 3, 5\}$, #MLP-layers $= \{1, 2, 3\}$, and different activation functions $\{\mathbb{1}, \tanh, \text{ReLU}\}$ for the MLP/CNN. We found that an extensive grid search was not necessary to obtain a well-performing model.

**Performance measures**    As in [49, 60, 40, 9, 66, 31, 102], we assess the geometric (dis)similarity between true and model-generated attractors using a Kullback-Leibler divergence ($D_{\text{stsp}}$) defined across state space. Specifically, $D_{\text{stsp}}$ determines the overlap between the distributions of true trajectory points, $p_{true}(\boldsymbol{x})$, and of model-generated trajectories, $p_{gen}(\boldsymbol{x}|\boldsymbol{z})$, given by

$$D_{\text{stsp}}(p_{true}(\boldsymbol{x}) \| p_{gen}(\boldsymbol{x}|\boldsymbol{z})) = \int p_{true}(\boldsymbol{x}) \log \frac{p_{true}(\boldsymbol{x})}{p_{gen}(\boldsymbol{x}|\boldsymbol{z})} d\boldsymbol{x}. \tag{11}$$

While especially for higher-dimensional state spaces these distributions may be approximated by Gaussian mixture models [49], here a discretized version was sufficient, with the state space parcellated into $K = m^N$ bins, where $m$ is the number of bins per dimension and $N$ the dimensionality of the system. We estimate the occupation probabilities $p_i$ of each bin via the relative frequencies $\hat{p}_i$ of trajectory visits and approximate $D_{\text{stsp}}$ as

$$D_{\text{stsp}} \approx \sum_{i=1}^{K} \hat{p}_{true;i} \log \frac{\hat{p}_{true;i}}{\hat{p}_{gen;i}}. \tag{12}$$

We set $m = 30$ bins per dimension, following [39]. For the $6d$ Lorenz-96 we use $m = 5$ and for the empirical time series $m = 20$. To ensure a steady-state distribution is reached, long trajectories of $T = 10,000$ time steps are sampled from the DS.

To evaluate the agreement in long-term temporal dynamics, we compute the *Hellinger distance* $D_H$ between the power spectra of the true and model-generated time series [60, 40], defined as

$$H(F(\omega), G(\omega)) = \sqrt{1 - \int_{-\infty}^{\infty} \sqrt{F(\omega)G(\omega)} d\omega} \ \in [0, 1], \tag{13}$$

where $F(\omega)$ and $G(\omega)$ are the power spectra of the true and generated time series, respectively. Power spectra are obtained via the dimension-wise Fast Fourier Transform (FFT), smoothed using a Gaussian kernel (with $\sigma = 20$ for DS and $\sigma = 2$ for the short empirical time series), and normalized to ensure comparability. High-frequency tails dominated by noise are truncated as described in [40]. The aggregated Hellinger distance $D_H$ is then computed as the average across all dimension-wise spectral comparisons.

Note that both $D_{\text{stsp}}$ and $D_H$ are assumed to assess (dis-)agreement in *long-term properties* in the limit $T \to \infty$. Hence, they are only sensible on rather long-term horizons, with $T = 10,000$ used here.

For assessing short-term forecast quality aggregated across the whole test set of $54$ DS, we used a *normalized* $n$-step ahead prediction error (with $n = 10$ in our evaluation) as recommended in [41], given by

$$\text{MASE} = \frac{1}{N} \sum_{i=1}^{N} \frac{\frac{1}{n} \sum_{t=T_C+1}^{T_C+n} |x_{i,t} - \hat{x}_{i,t}|}{\frac{1}{T} \sum_{t=T_C+1}^{T_C+T} |x_{i,t} - x_{i,t-1}|} \tag{14}$$

For individual comparisons on single empirical times, the MAE for evaluating short term predictions [41] was used:

$$\text{MAE} = \frac{1}{n} \sum_{t=T_C+1}^{T_C+n} |x_t - \hat{x}_t| \ , \tag{15}$$

where we chose $n$ according to the data's temporal scale and resolution ({ETTh1,fMRI}: $n = 40$,{traffic, cloud requests, partially obs. DS}: $n = 80$, {human EEG, weather air pressure}: $n = 120$, {weather temperature}: $n = 200$).

**Lyapunov exponent**   For numerically estimating the *maximum Lyapunov exponent* $\lambda_{\max}$ from a one-dimensional empirical time series $\{x_t\}, t = 1 \ldots T$, one common approach is the *Rosenstein algorithm* [74]. First, the time series is embedded into a state space using time-delay embedding as in eq. 5 (e.g. using the PECUZAL algorithm [50]). For each embedded point $\boldsymbol{x}_i$, the nearest neighbor $\boldsymbol{x}_j$ is found under the constraints $|i - j| > l_t$, with $l_t$ a threshold to remove purely temporal neighbors (living on the same piece of trajectory), and a minimal initial state space separation $\|\boldsymbol{x}_i - \boldsymbol{x}_j\|_2 > l_s$, where we chose $l_t = 150$ (according to the mean periodicity [74]) and $l_s = 0.25$. The local divergence between initially nearby trajectories is then tracked over a range of time steps $k = 0 \ldots k_{max}$,

$$d_i(k) = \|\boldsymbol{x}_{i+k} - \boldsymbol{x}_{j(i)+k}\|_2 \ . \tag{16}$$

An estimate of the maximum Lyapunov exponent is given by

$$d_i(k) \approx d_i(0)e^{\lambda_{\max}k\Delta t} \ , \tag{17}$$

where $\Delta t$ is the temporal resolution of the series. Taking the logarithm and averaging over all pairs leads to

$$\langle \ln d(k) \rangle \approx \lambda_{\max}k\Delta t + C. \tag{18}$$

In chaotic systems, for a suitable choice of $k_{max}$ (here: 50), this quantity initially grows approximately linearly with time [45], with the slope given by $\lambda_{\max} > 0$. In contrast, periodic (cyclic) systems have $\lambda_{\max} = 0$. As for $D_{\text{stsp}}$ and $D_H$, Lyapunov exponents can only be reasonably assessed for sufficiently long time series, with practical guidelines of $T \approx 10^d - 30^d$, where $d$ is the attractor's fractal dimension [96, 74].

## A.2   Datasets

**Training data**   DynaMix is trained on about 0.6 million simulated time series of length $T = 550$ sampled from 34 different $3d$ DS with cyclic or chaotic attractors, collected in [30]. Time series were standardized to ensure comparable scaling of all data, and Gaussian noise of 5% of the data standard deviation was added to all dimensions for the ground truth. Attractor dynamics were chosen to reflect different types of behavior, see Fig. 10.

**Test data**   Our test set for DS consists of simulated time series of length $10^5$ sampled from 54 different $3d$ DS collected in [30], which are not part of the training set.

Furthermore, we evaluated several $2d$ systems: The Selkov system [79] describing a kinetic model of an open monosubstrate enzyme reaction given by the equations

$$\begin{aligned}
\frac{dx}{dt} &= -x + ay + x^2 y, \\
\frac{dy}{dt} &= b - ay - x^2 y,
\end{aligned} \tag{19}$$

where we chose $a = 0.1$ and $b = 0.5$. The Van-der-Pol system [89] describes self-sustaining oscillations in vacuum tubes by

$$\begin{aligned}
\frac{dx}{dt} &= y, \\
\frac{dy}{dt} &= \mu(1 - x^2)y - x,
\end{aligned} \tag{20}$$

where we chose $\mu = 0.5$. Similar to the Van-der-Pol system, the Rayleigh oscillator [63] describes a self-sustained nonlinear oscillator through

$$\begin{aligned}
\frac{dx}{dt} &= y, \\
\frac{dy}{dt} &= \mu(1 - \frac{y^2}{3})y - x,
\end{aligned} \tag{21}$$

where we chose $\mu = 1.0$. As a higher dimensional DS test case we use the Lorenz-96 system [56] defined by

$$\frac{\mathrm{d}x_i}{\mathrm{d}t} = (x_{i+1} - x_{i-2})x_{i-1} - x_i + F, \tag{22}$$

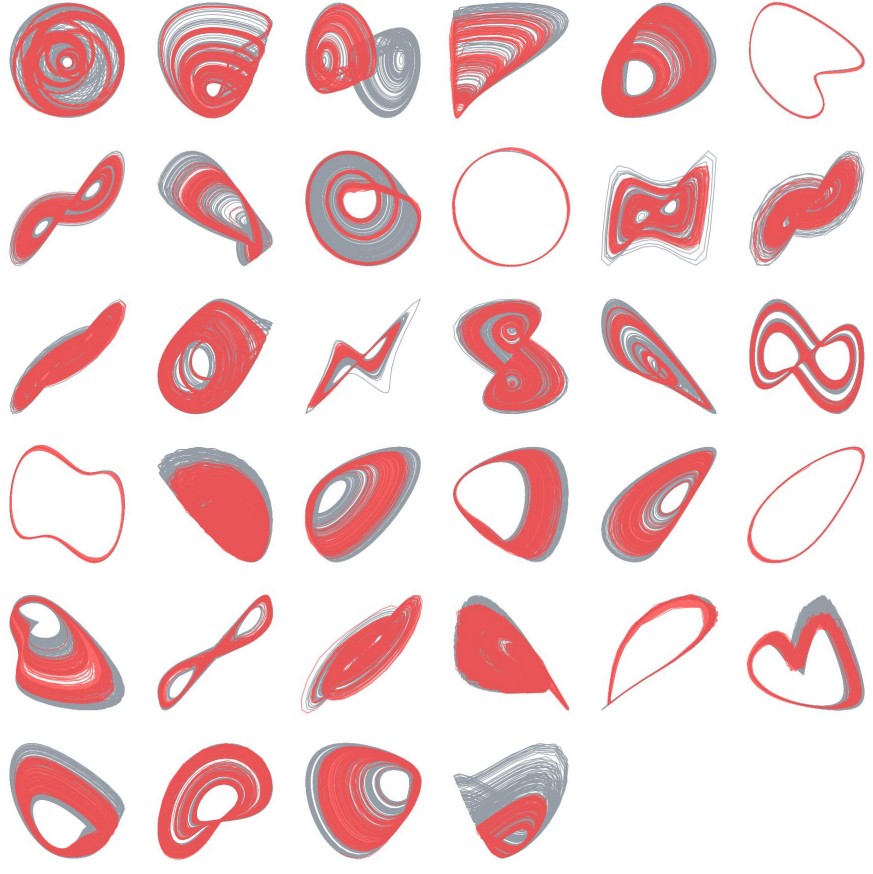

Figure 10: Training data (gray) and their reconstructions in red.

with system variables $x_i$, $i = 1, ..., N$, and forcing term $F$ (here, $F = 8$ and $N = 6$, in the chaotic regime). Furthermore, cyclic boundary conditions are assumed with $x_{-1} = x_{N-1}$, $x_0 = x_N$, $x_{N+1} = x_1$, and the system was solved with integration step $\Delta t = 0.08$.

We further probed forecasting and DSR on different types of real-world data: The *traffic data* are hourly recordings of the number of cars passing road junctions (https://www.kaggle.com/datasets/fedesoriano/traffic-prediction-dataset/data). The *cloud data*, also used in [86] to evaluate TS foundation models, are publicly available from Huawei Cloud (https://github.com/sir-lab/data-release). It consists of function requests from Huawei's serverless cloud platform. The *weather data* consists of daily sampled soil temperature and air pressure measured in the city of Mannheim, Germany (source: Deutscher Wetterdienst [German weather service]), and can be accessed via https://www.dwd.de/EN/ourservices/cdc/cdc_ueberblick-klimadaten_en.html. The *functional magnetic resonance imaging (fMRI) data* comes from human subjects performing cognitive tasks and is publicly available on GitHub [51]. We followed Kramer et al. [51] and selected the first principal component of BOLD activity in each of the 20 brain regions. Kramer et al. [51] report a positive maximum Lyapunov exponent for models reconstructed from these time series, indicating their chaotic nature (see also [91]). The *ETTh1 dataset* is part of the Electricity Transformer Temperature (ETT) benchmark, which is widely used for evaluating TS forecasting models. It contains hourly data collected from a power transformer station [103] and can be accessed at https://github.com/zhouhaoyi/ETDataset. *Electroencephalogram (EEG) data* were taken from a study by Schalk et al. [77], comprising 64-channel data collected from human subjects performing various motor and imagery tasks. Following the approach of Brenner et al. [7], the signals were smoothed using a Hann window of length 15. As for the training data, all test data were standardized for processing by the models.

### A.3  TS foundation models

**Chronos**  Chronos is a recent TS foundation framework that adapts a transformer-based LLM architecture for probabilistic time series forecasting [2]. Key to this approach is the tokenization of real-valued time series observations through scaling and quantization, transforming them into token sequences accessible to language models. The model is pretrained on extensive datasets, including both synthetic data generated via Gaussian processes, as well as an extensive batch of empirical time series from [2, 33], including traffic, weather/ climate, electricity and web data, enabling it to achieve strong zero-shot performance across diverse datasets without task-specific fine-tuning. For our evaluation we used the standard pipeline as described in `https://github.com/amazon-science/chronos-forecasting`.

**TimesFM**  TimesFM is another transformer based TS foundation model, with a decoder-only style architecture using input patching [21]. Its training corpus consists of synthetic as well as real-world time series data, and it exhibits generalization across different time series domains and context lengths. Evaluation is performed as in `https://github.com/google-research/timesfm`.

**Mamba4Cast**  Mamba4Cast is a zero-shot time series forecasting model based on the Mamba architecture [36], a type of linear ('state-space') RNN with nonlinear input & output gating, and inspired by Prior-data Fitted Networks (PFNs) [6]. Trained exclusively on synthetic data, Mamba4Cast can generate zero-shot forecasts when provided with time series context information. For evaluation we follow `https://github.com/automl/Mamba4Cast`.

**Tiny Time Mixers**  Like Mamba4Cast, Tiny Time Mixers is a time series foundation model *not* founded on transformers [27]. It builds upon the TSMixer backbone [26] to which it adds several key innovations such as adaptive patching, diverse resolution sampling, and multi-resolution prefix tuning to improve generalization across datasets with varying temporal resolutions. Evaluation here is performed as in `https://github.com/glehet/TTM1`.

**Panda**  Panda is a recently proposed foundation model for short-term forecasting of DS [52]. It is based on a transformer architecture, which relies on patching the DS-generated time series. The model is trained on $2 \cdot 10^4$ DS produced by combining base DS from the same database used to train and evaluate DynaMix in skew-product form [30]. Evaluation is performed as in `https://github.com/abao1999/panda`.

For comparability, all models were evaluated on the exact same CPU (18-Core Xeon Gold 6254) and GPU (Nvidia RTX 2080 Ti) using 512GB of RAM.

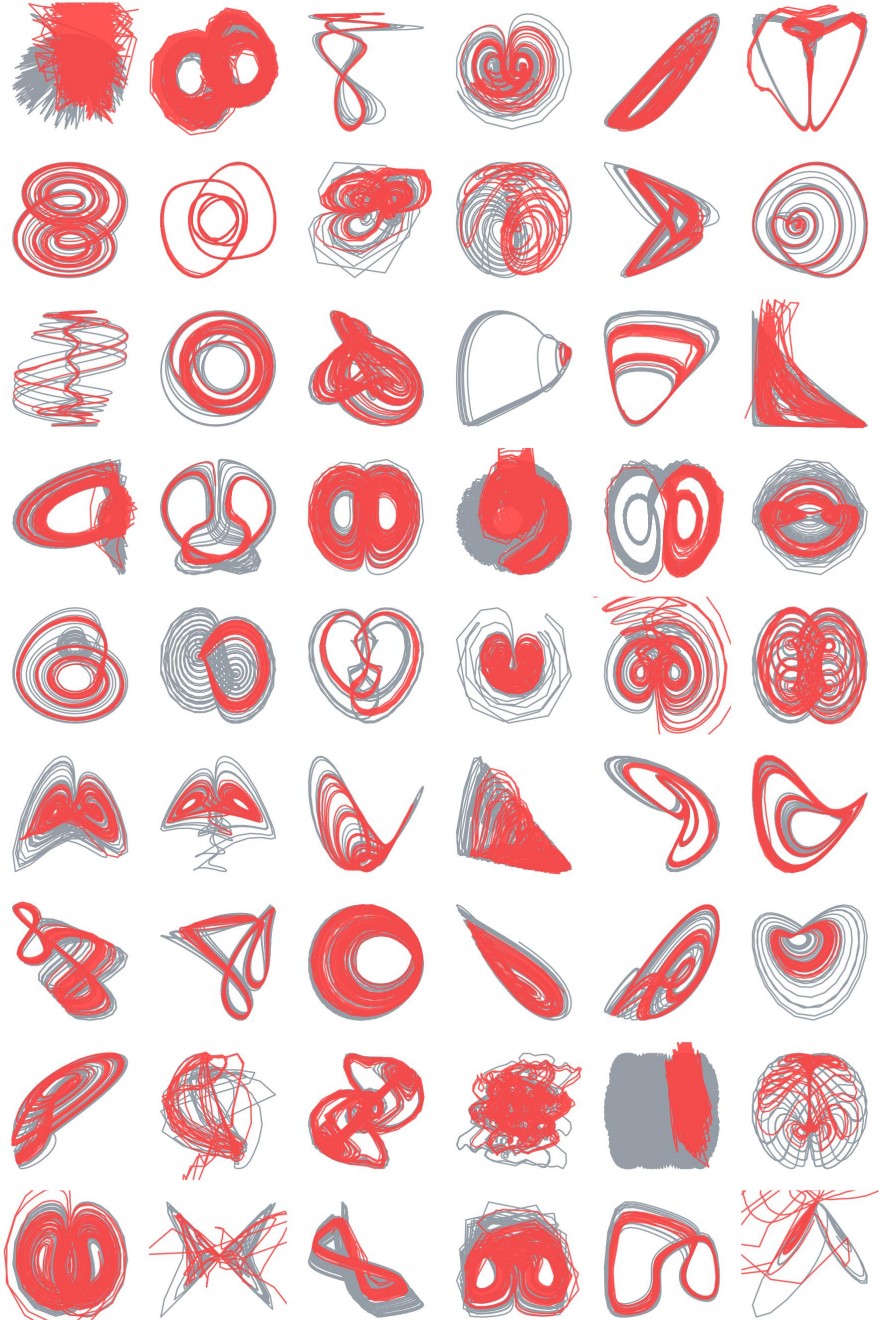

Figure 11: Zero-shot DSR (red) from a 2000-step context of unseen DS not contained in the training corpus (ground truth in gray).

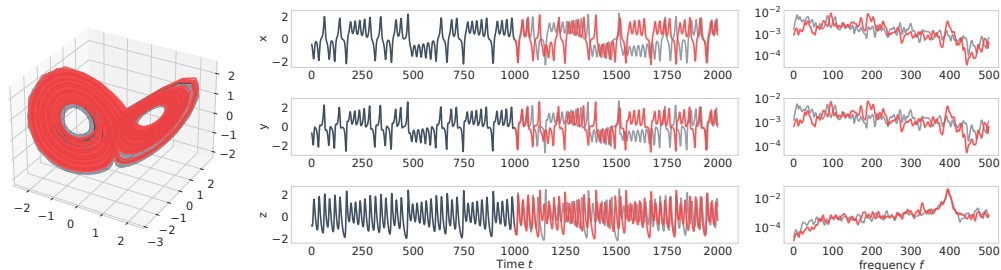

Figure 12: Zero-shot DSR of chaotic Lorenz-63 system (darkgray: context, lightgray: ground truth, red: model-generated). Left: State space, center: time graphs, right: power spectrum.

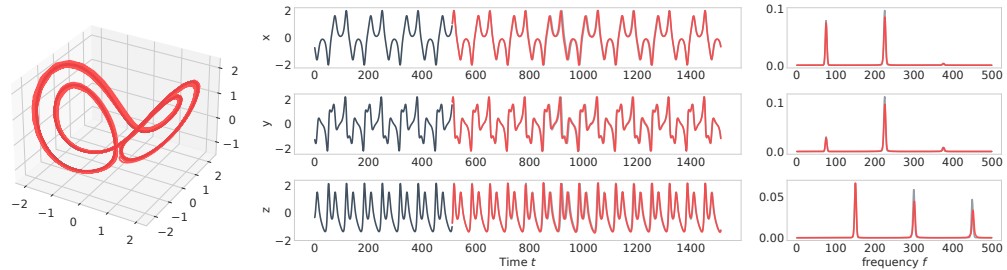

Figure 13: Zero-shot DSR of Lorenz-63 system in cyclic regime (darkgray: context, lightgray: ground truth, red: model-generated). Left: State space, center: time graphs, right: power spectrum.

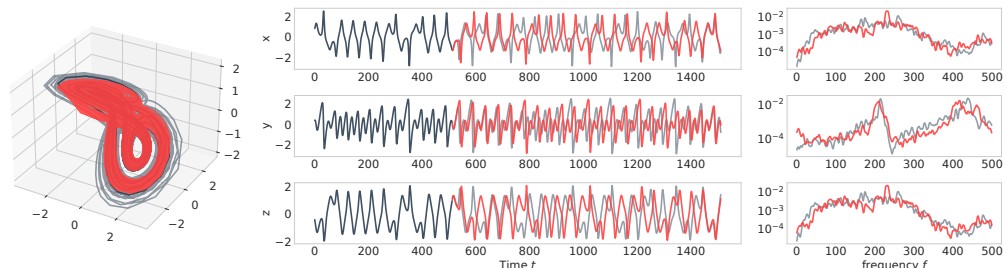

Figure 14: Zero-shot DSR of chaotic finance system (darkgray: context, lightgray: ground truth, red: model-generated). Left: State space, center: time graphs, right: power spectrum.

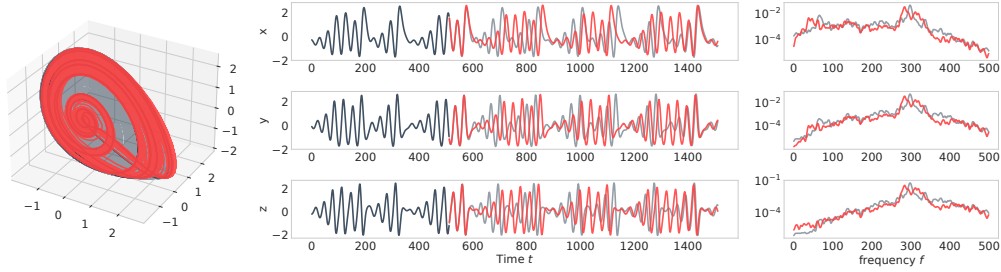

Figure 15: Zero-shot DSR of chaotic Genesio Tesi system (darkgray: context, lightgray: ground truth, red: model-generated). Left: State space, center: time graphs, right: power spectrum.

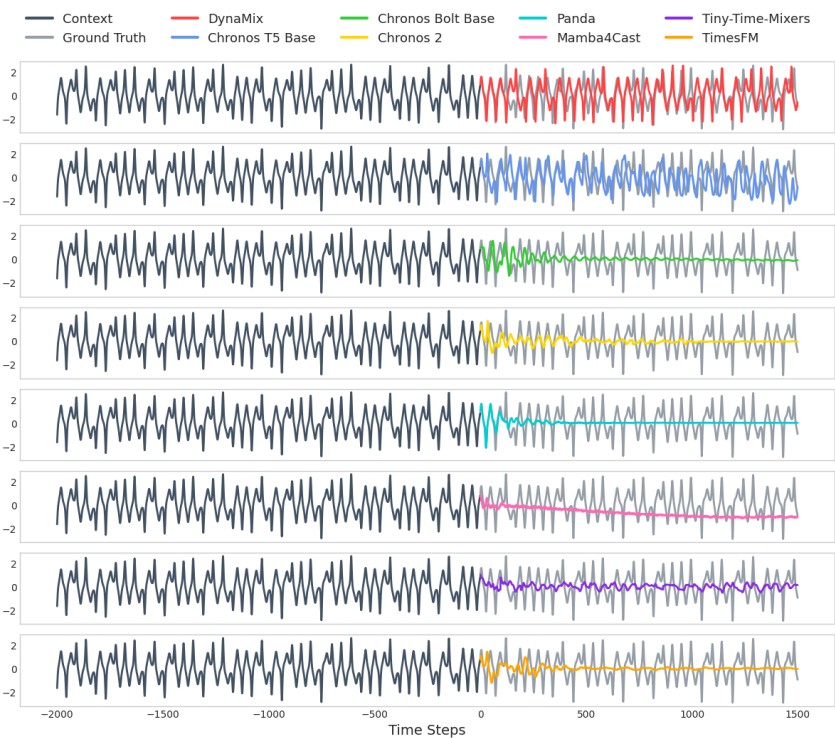

Figure 16: Comparison of zero-shot forecasting of finance system for DynaMix vs. different TS foundation models.

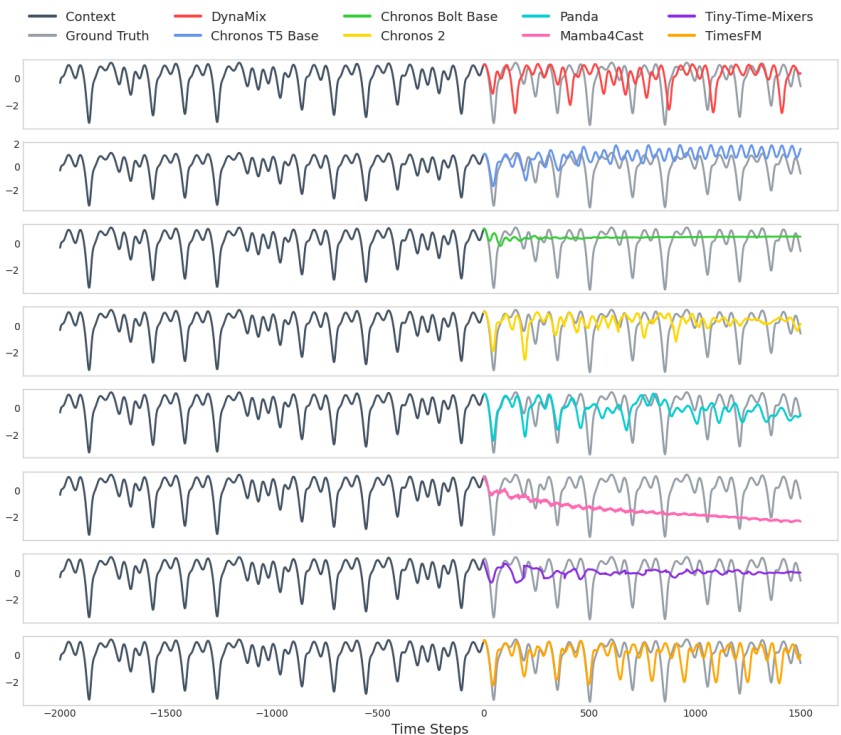

Figure 17: Comparison of zero-shot forecasting of Sprott D system for DynaMix vs. different TS foundation models.

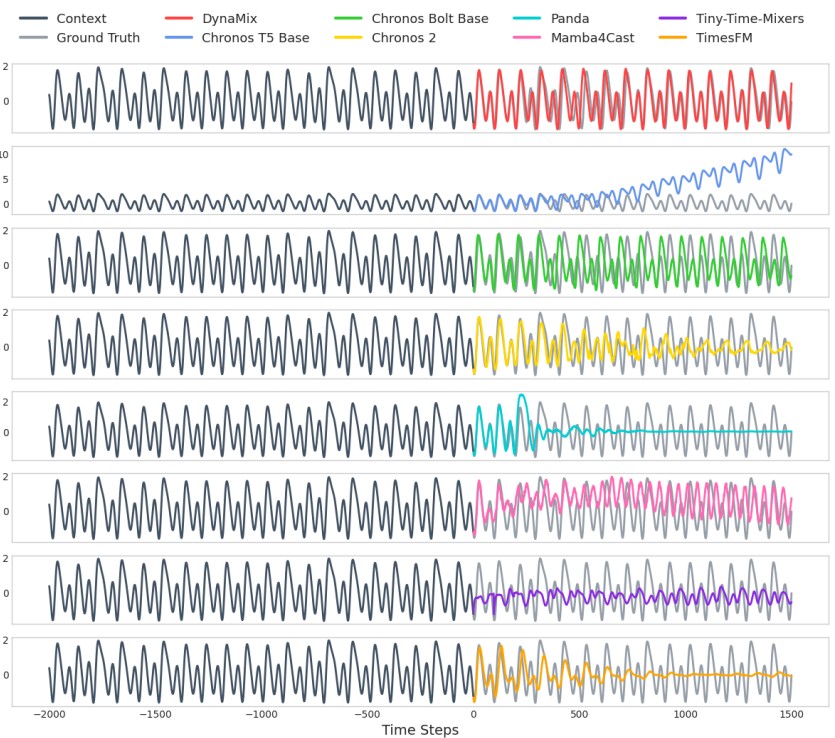

Figure 18: Comparison of zero-shot forecasting of Sprott M system for DynaMix vs. different TS foundation models.

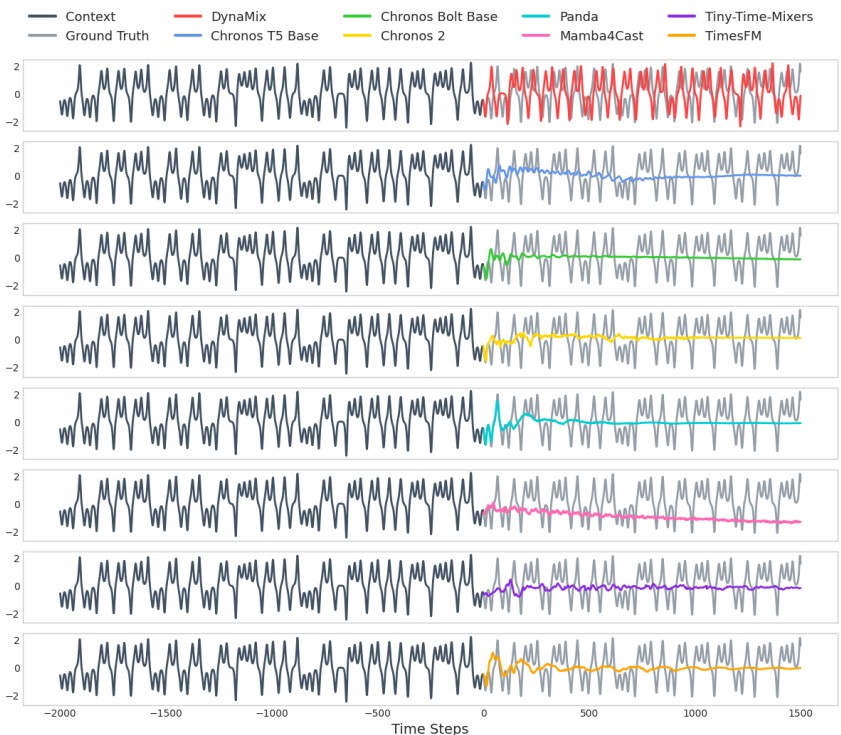

Figure 19: Comparison of zero-shot forecasting of Vallise El Nino system for DynaMix vs. different TS foundation models.

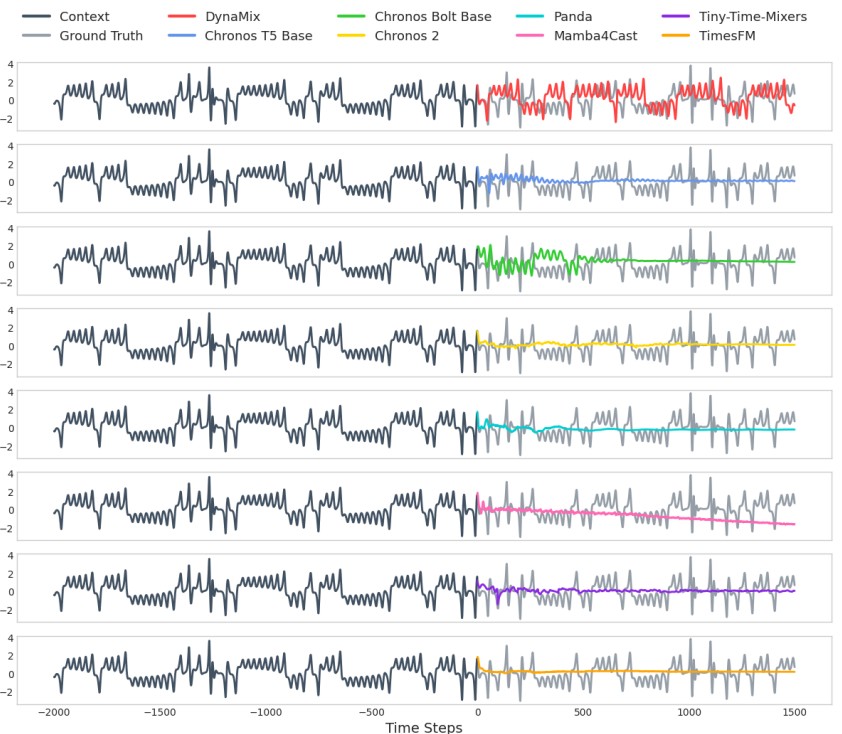

Figure 20: Comparison of zero-shot forecasting of Sprott C system for DynaMix vs. different TS foundation models.

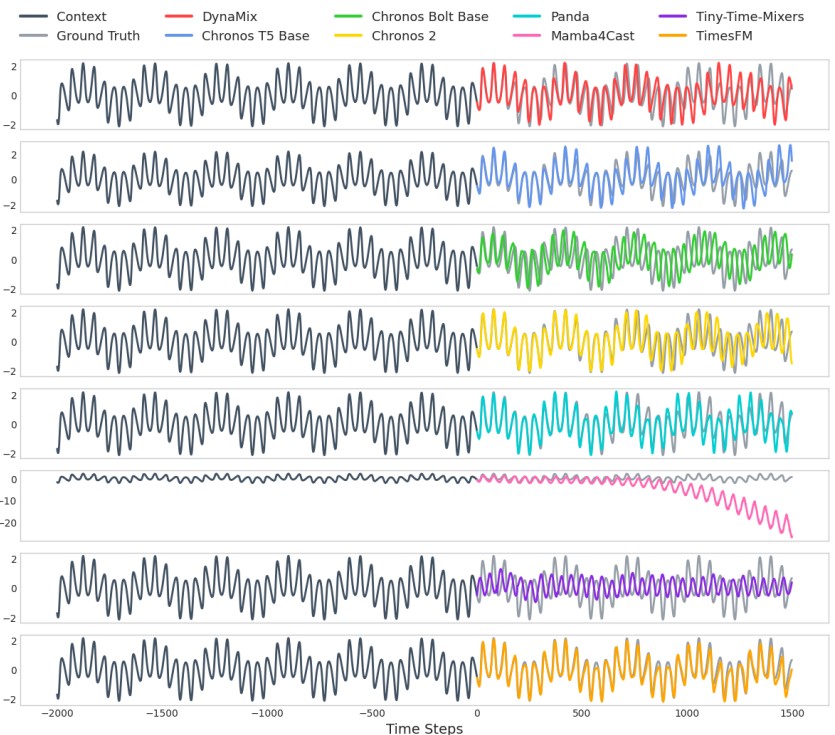

Figure 21: Comparison of zero-shot forecasting of Sprott A system for DynaMix vs. different TS foundation models.

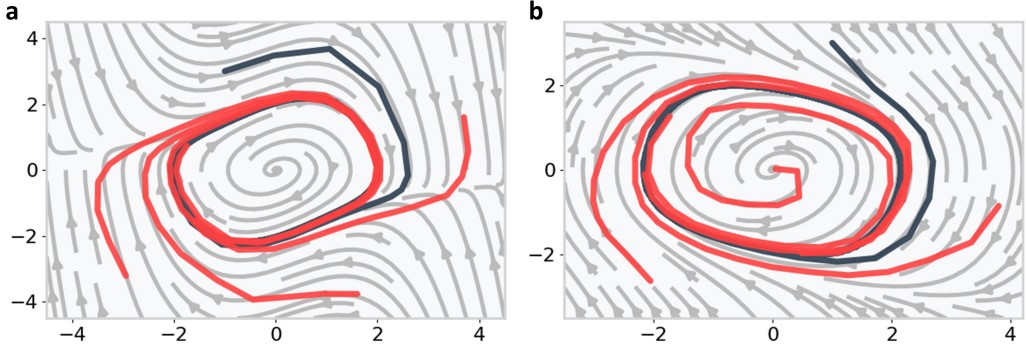

Figure 22: Zero-shot forecasts (red) for the **a**) Van-der-Pol system. **b**) Rayleigh oscillator (true vector field in lightgray) from different initial conditions outside the context range (darkgray).

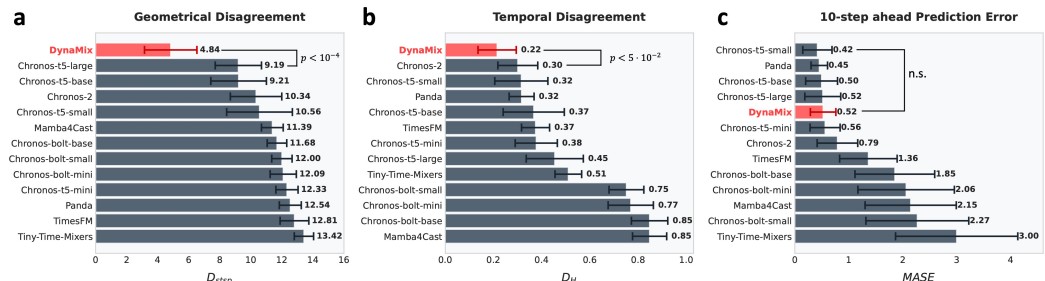

Figure 23: Zero-shot DSR performance across all 54 test set DS for DynaMix and various TS foundation models as in Fig. 4, but for context length $T_C = 512$. Numerical details in Table 10. Statistical testing based on Wilcoxon signed-rank tests.

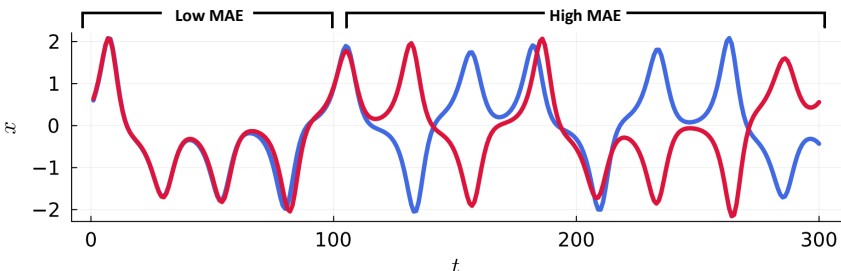

Figure 24: Exponential divergence of two initially nearby trajectories (in blue and red) illustrated for the chaotic Lorenz-63 system. The prediction error is still sensible on a short time scale (MAE $= 0.09$), but then rapidly increases and breaks down as a suitable metric in the longer-term (MAE $= 1.12$), *although both trajectories were drawn from the exact same system with the very same parameters*.

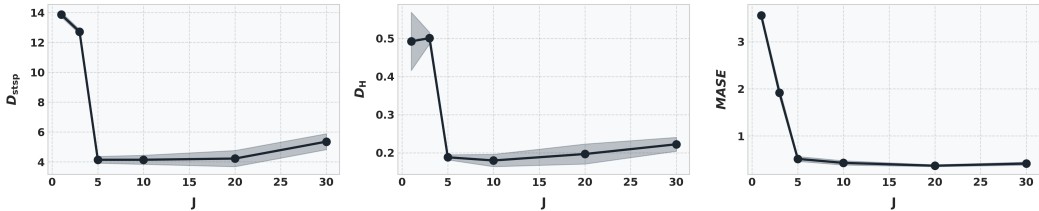

Figure 25: DynaMix' performance as a function of the number of experts $J$. While a minimum number of experts is necessary, performance already plateaus after a surprisingly small number of experts, suggesting there is a lot of room for scaling the model to larger datasets. Error bands = STD

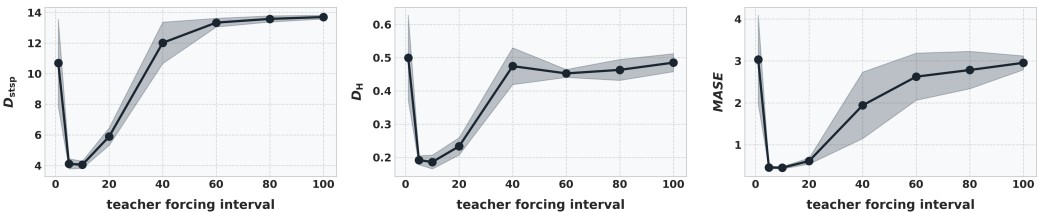

Figure 26: DSR performance for different teacher forcing intervals $\tau$, illustrating an optimal value is essential for successful training, see [60, 7]. Error bands = STD

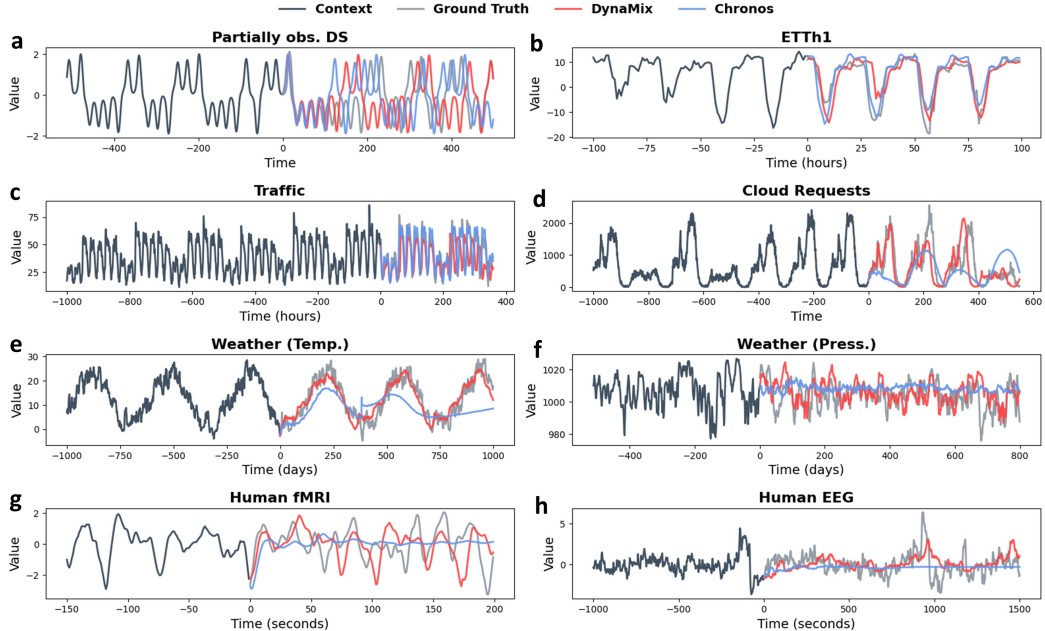

Figure 27: Comparison of DynaMix (red) to **Chronos** (blue) on zero-shot forecasts of various empirical time series: Forecasts of **a**) partially (1d) observed Lorenz-63 DS, **b**) electricity transformer temperature data, **c**) hourly car traffic data with weekly cycle, **d**) Huawei cloud request data, **e**) soil temperature development, **f**) air pressure data, **g**) human functional magnetic resonance imaging (fMRI) data, **h**) human electroencephalogram (EEG) data.

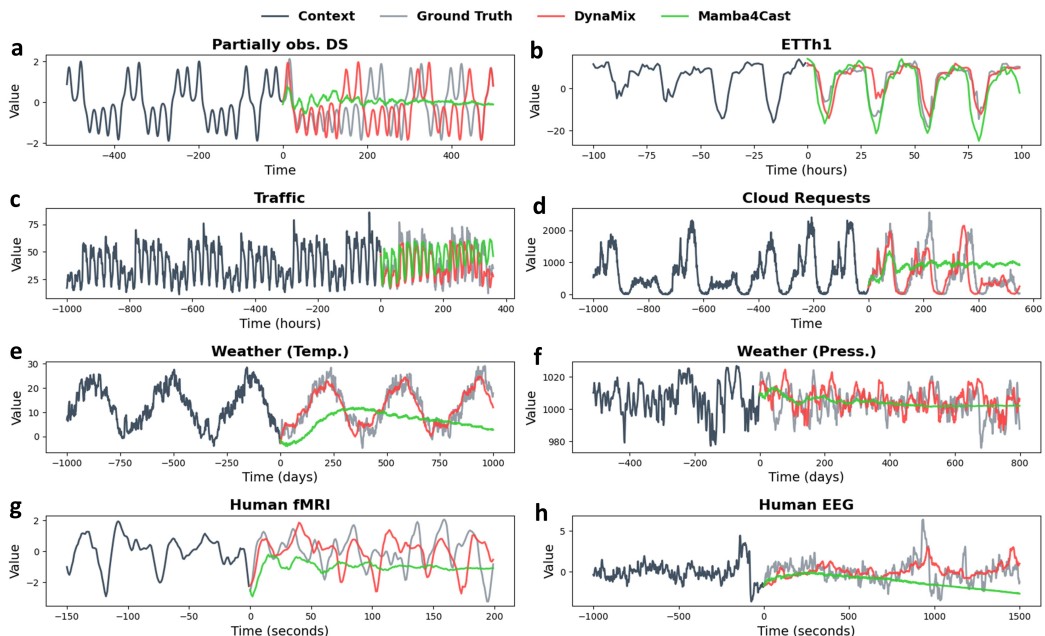

Figure 28: Comparison of DynaMix (red) to **Mamba4Cast** (green) on zero-shot forecasts of various empirical time series: Forecasts of **a**) partially (1d) observed Lorenz-63 DS, **b**) electricity transformer temperature data, **c**) hourly car traffic data with weekly cycle, **d**) Huawei cloud request data, **e**) soil temperature development, **f**) air pressure data, **g**) human functional magnetic resonance imaging (fMRI) data, **h**) human electroencephalogram (EEG) data.

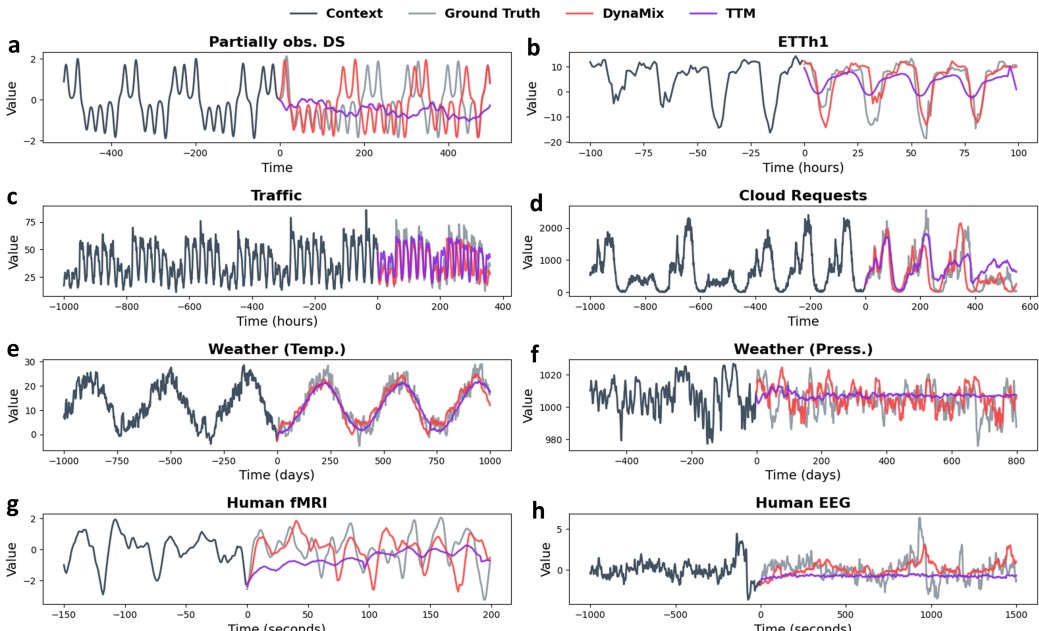

Figure 29: Comparison of DynaMix (red) to **Tiny-Time Mixers** (purple) on zero-shot forecasts of various empirical time series: Forecasts of **a**) partially (1d) observed Lorenz-63 DS, **b**) electricity transformer temperature data, **c**) hourly car traffic data with weekly cycle, **d**) Huawei cloud request data, **e**) soil temperature development, **f**) air pressure data, **g**) human functional magnetic resonance imaging (fMRI) data, **h**) human electroencephalogram (EEG) data.

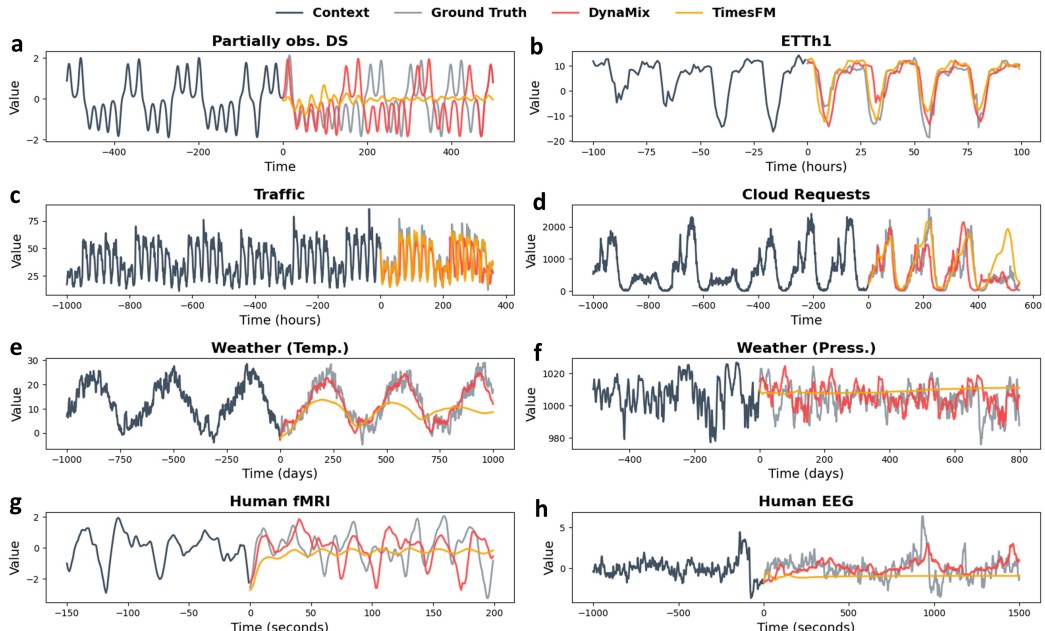

Figure 30: Comparison of DynaMix (red) to **TimesFM** (orange) on zero-shot forecasts of various empirical time series: ground truth): Forecasts of **a**) partially (1d) observed Lorenz-63 DS, **b**) electricity transformer temperature data, **c**) hourly car traffic data with weekly cycle, **d**) Huawei cloud request data, **e**) soil temperature development, **f**) air pressure data, **g**) human functional magnetic resonance imaging (fMRI) data, **h**) human electroencephalogram (EEG) data.

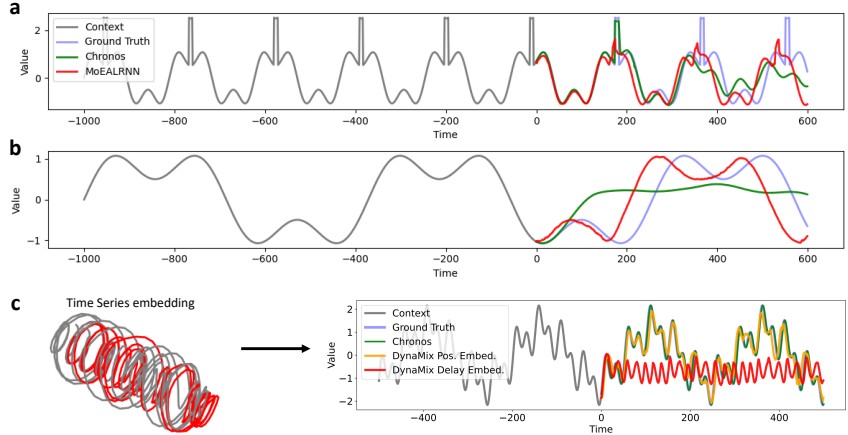

Figure 31: Failure of proper zero-shot DSR. **a**) DynaMix struggles with sharp peaks in an otherwise much more slowly evolving time series. **b**) As quantified in Fig. 3, selecting a too broad temporal resolution of the context time series can lead to improper reconstructions with, in this case, temporal disalignment w.r.t. the true signal. However, this is a general problem for any TS foundation model, and Chronos fares even much worse in this example. **c**) Zero-shot DSR may fail (red) as well if the context time series is not properly embedded. In this example, the required embedding dimension exceeded the model capacity. However, as shown in orange, this can be amended by using the positional encoding, eq. 6.

## A.5 Using DynaMix for non-stationary time series

Similar as in FEDformers [104] and related TS models which use specialized decomposition and filtering blocks for handling non-stationary data, one could add simple preprocessing operations to the DynaMix pipeline to separate out trend or other non-stationary components in the context signal. For a simple illustration, here we first apply a *Box-Cox* transformation to time series $x_t$,

$$x_t^{(\lambda)} = \begin{cases} \frac{x_t^\lambda - 1}{\lambda}, & \text{if } \lambda \neq 0 \\ \log(x_t), & \text{if } \lambda = 0 \end{cases} \tag{23}$$

where $\lambda$ is estimated by maximizing the log-likelihood. Next, trend components of the form

$$f(t; \boldsymbol{\theta}) = \theta_1 t^{\theta_2} + \theta_3 \ , \tag{24}$$

are inferred by least squares estimation and subtracted from the context signal. Standard DynaMix is then used to forecast the embedded residual context (see Sect. 3.3), after which the estimated trend model is added back on.

We tested this simple setup on the non-stationary *Air Passengers* dataset containing passenger counts of an airline (`https://www.kaggle.com/datasets/chirag19/air-passengers`). The results in Fig. 32 illustrate that this is in principle a viable direction, although constituting just a proof-of-concept at this stage.

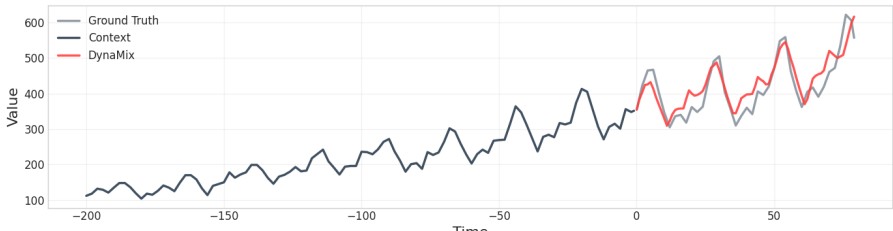

Figure 32: Example forecast for the non-stationary Air Passenger data using DynaMix with preprocessing pipeline.

## A.6 Scaling up context dimension

For reconstructing DS higher-dimensional than the context dimension defined by the architecture and used in training, one can use DynaMix without any modification by employing the delay-embedding theorems [82, 76]: Zero-shot infer the underlying DS from the observed TS and then delay-embed DynaMix' output into a sufficiently high-dimensional space which assures a diffeomorphism between original and reconstructed attractor. This idea is illustrated for the $6d$ Lorenz-96 system in Fig. 33. Here, only the first two dimensions of the simulated Lorenz-96 were provided as context, for which DynaMix then produced long-term forecasts. The good geometrical and temporal agreement between delay-embeddings of both the two ground truth and the forecasted TS confirms that DynaMix has correctly inferred the underlying $6d$ DS despite its own $3d$ structure.

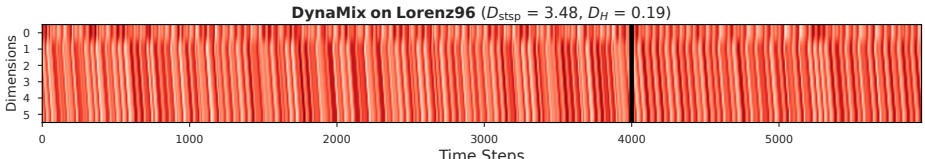

Figure 33: Zero-shot forecasting of partially observed ($2d$) Lorenz-96 system by DynaMix. Both context signal (ground truth) and DynaMix' forecast were subsequently delay-embedded into $6d$, the system's original dimension. The black line indicates the transition between context and forecast.

Alternatively, if more dimensions are observed than the gating network can handle by design, $N^* > N$, and are all to be utilized for DSR and forecasting, then the gating network needs to be

modified architecturally such that it can receive $N = N^*$ dimensional context signals. To allow for this *without changing the* $3d$ *DS training corpus*, observations $\boldsymbol{x} \in \mathbb{R}^3$ were embedded into a higher dimensional $N > 3$ space using a nonlinear transformation of the form

$$\tilde{\boldsymbol{x}} = f_{emb}(\boldsymbol{x}) = \left[\boldsymbol{x}^T \ \tanh(\boldsymbol{A}\boldsymbol{x})^T\right]^T \in \mathbb{R}^N, \tag{25}$$

where the entries in $\boldsymbol{A} \in \mathbb{R}^{E \times 3}$ are chosen randomly from $a_{ij} \sim \mathcal{U}(-1, 1)$, yielding an embedding dimension of $N = 3 + E$. DynaMix is then retrained on the embedded and standardized original 34 DS training set. Note that this embedding *does not change the nature of the dynamics of the trained-on systems in any way*, because $f_{emb}$ is an instantaneous (time-independent) transform that is only applied posthoc (*after* the DS has been simulated). It essentially places trajectories onto a $3d$ manifold embedded within an $N$ dimensional ambient space. Note that this yields a general recipe for any desired $N$.

As shown in Fig. 34 and Table 2, using this embedding DynaMix is able to generalize to all $6d$ of the Lorenz-96 system, although only trained on $3d$ systems, while all time series foundation models fail.

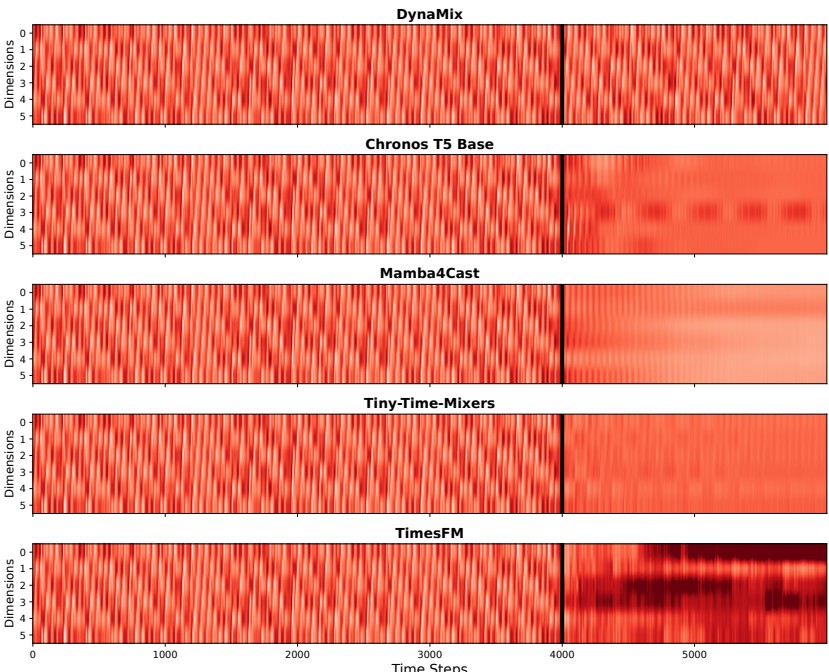

Figure 34: Comparison of zero-shot forecasting of $6d$ Lorenz-96 system by DynaMix vs. different TS foundation models. The black line indicates the transition between context and forecast.

Table 2: Zero-shot DSR performance across 5 different Lorenz-96 trajectories for DynaMix and various TS foundation models for context length $T_C = 4000$. Median±MAD of $D_{stsp}$ (geometrical disagreement), $D_H$ (temporal disagreement), and MASE (short-term prediction error).

| Model | $D_{\text{stsp}}$ | $D_H$ | MASE |
|---|---|---|---|
| DynaMix | $1.93 \pm 0.08$ | $0.07 \pm 0.00$ | $1.02 \pm 0.06$ |
| Chronos-t5-base | $12.74 \pm 1.18$ | $0.37 \pm 0.06$ | $2.44 \pm 1.30$ |
| Mamba4Cast | $13.34 \pm 0.36$ | $0.70 \pm 0.00$ | $2.93 \pm 0.45$ |
| TTM | $14.54 \pm 0.02$ | $0.26 \pm 0.02$ | $4.00 \pm 0.49$ |
| TimesFM | $11.00 \pm 0.18$ | $0.67 \pm 0.01$ | $4.82 \pm 0.56$ |

### A.7 Comparison of DynaMix to custom-trained DSR models

To put DynaMix' zero-shot DSR performance further into context, we compared it to different SOTA DSR models explicitly trained on the respective context data. Note that for the custom-trained models – unlike DynaMix – this constitutes a form of *in-domain* generalization, as the context signal and the forecast come from the same ergodic distribution. For these reasons, it is also difficult to establish a fair comparison, since hyper-parameters of the custom-trained models could be arbitrarily fine-tuned on the in-domain data (with often better results as the model size is further increased, indicating a tendency toward over-fitting). For the comparisons performed in here, we therefore decided to mainly rely on previously reported hyper-parameter settings, which often were already fine-tuned for evaluation on similar DS, without additional fine-tuning (which we did not do for DynaMix either).

Our first comparison model is the same **AL-RNN** used for DynaMix' experts (with $\approx 500$ trainable parameters), i.e. given by eq. 1. Training was performed as in Brenner et al. [8], and hyperparameter settings, as collected in Table 3, followed those in the original paper.

Table 3: Hyperparameter settings of custom AL-RNN.

| Hyperparameter | Setting |
|---|---|
| $M$ | 20 |
| $P$ | 8 (Lorenz-96: 14) |
| $\tau$ | 16 |
| $T$ | 200 |
| batch size | 16 |
| $\eta_{\text{start}}$ | $10^{-3}$ |
| $\eta_{\text{end}}$ | $10^{-5}$ |
| epochs | 2000 |

As another SOTA DSR model we trained **Neural-ODEs** [18], using the same type of MLP architecture, training setup, and same hyper-parameter settings as in [38], see Table 4. With this, the total number of trainable parameters is already more than twice (!) than that used for DynaMix ($\approx 10$k), and exceeded the number of data points ($\leq 4$k) manifold. Indeed, results became worse with less parameters, potentially indicating that Neural-ODEs particularly struggle on the short training segments.

Table 4: Hyperparameter settings of custom Neural-ODEs.

| Hyperparameter | Setting |
|---|---|
| hidden layer | [100, 100, 100] |
| activation | ReLU |
| $T$ | 30 |
| batch size | 32 |
| ODE solver | Tsit5 |
| $\eta_{\text{start}}$ | $10^{-3}$ |
| $\eta_{\text{end}}$ | $10^{-5}$ |
| epochs | 100000 |

Finally, **Reservoir Computers** (RCs) are widely used for DSR. Here we employed the architecture provided in Patel and Ott [67], trained as in Göring et al. [38] and using the same hyper-parameter settings, summarized in Table 5. Note that this endows RCs with about $25\times$ more parameters in total, of which $\leq 3$k were trainable, than DynaMix had at its disposal.

Table 5: Hyperparameter settings of custom Reservoir Computer.

| Hyperparameter | Setting |
|---|---|
| $M$ | 500 |
| $\rho$ | 1.0 |
| $\alpha$ | 0.7 |
| $\sigma$ | 0.2 |
| $\beta$ | 0.5 |

With these settings, performance results in Tables 6 and 7 indicate that DynaMix is within the same ballpark as the custom-trained DSR models, in contrast to all the TS foundation models. For the empirical time series, Table 8, DynaMix sometimes even seems to have an advantage over custom-trained models, potentially because it makes more efficient use of the relatively short context time series (for a fair comparison, custom-trained models were provided with the same positional embedding as DynaMix; using a standard delay embedding in $3d$ for the custom-trained models actually produced worse results). At the same time, DynaMix admits orders of magnitude faster inference times (see Fig. 5) and performs true *out-of-domain* generalization without parameter fine-tuning.

Table 6: Performance of DynaMix and custom trained DSR models across all 54 test set DS. Median$\pm$MAD of geometrical divergence ($D_{\text{stsp}}$), long-term temporal distance ($D_H$), and forecast error (MASE).

| Model | $D_{\text{stsp}}$ | $D_H$ | MASE |
|---|---|---|---|
| DynaMix | $3.8 \pm 1.4$ | $0.16 \pm 0.06$ | $0.35 \pm 0.17$ |
| AL-RNN | $3.86 \pm 2.48$ | $0.08 \pm 0.06$ | $0.47 \pm 0.23$ |
| Neural-ODE | $3.43 \pm 2.05$ | $0.09 \pm 0.04$ | $0.18 \pm 0.13$ |
| RC | $1.33 \pm 0.99$ | $0.08 \pm 0.07$ | $0.86 \pm 0.37$ |

Table 7: Performance of DynaMix and custom trained DSR models across 5 different Lorenz-96 trajectories. Median$\pm$MAD of geometrical divergence ($D_{\text{stsp}}$), long-term temporal distance ($D_H$), and forecast error (MASE).

| Model | $D_{\text{stsp}}$ | $D_H$ | MASE |
|---|---|---|---|
| DynaMix | $1.93 \pm 0.08$ | $0.07 \pm 0.00$ | $1.02 \pm 0.06$ |
| AL-RNN | $2.12 \pm 0.18$ | $0.15 \pm 0.03$ | $0.83 \pm 0.29$ |
| Neural-ODE | $0.76 \pm 0.06$ | $0.06 \pm 0.01$ | $0.34 \pm 0.16$ |
| RC | $1.26 \pm 0.51$ | $0.13 \pm 0.08$ | $1.10 \pm 0.42$ |

Table 8: Performance comparison on empirical time series in terms of geometrical divergence ($D_{\text{stsp}}$), long-term temporal distance ($D_H$), and forecast error (MAE). Best in red, second-best in blue.

| System | DynaMix | | | AL-RNN | | | RC | | | Neural-ODE | | |
|---|---|---|---|---|---|---|---|---|---|---|---|---|
| | $D_{\text{stsp}}$ | $D_H$ | MAE | $D_{\text{stsp}}$ | $D_H$ | MAE | $D_{\text{stsp}}$ | $D_H$ | MAE | $D_{\text{stsp}}$ | $D_H$ | MAE |
| ETTh1 | 0.22 | 0.08 | 4.92 | 0.26 | 0.08 | 6.82 | 0.60 | 0.75 | 44.41 | 0.70 | 0.09 | 4.55 |
| Traffic | 0.81 | 0.21 | 6.93 | 0.95 | 0.13 | 7.29 | 0.19 | 0.70 | 7.08 | 0.85 | 0.17 | 5.03 |
| Cloud Requests | 0.27 | 0.14 | 161.58 | 0.19 | 0.20 | 529.91 | 0.24 | 0.19 | 587.35 | 0.37 | 0.30 | 172.61 |
| Weather (Temp.) | 0.66 | 0.09 | 2.24 | 0.53 | 0.11 | 2.08 | 1.20 | 0.11 | 2.50 | 0.65 | 0.11 | 2.23 |
| Weather (Press.) | 0.39 | 0.19 | 9.03 | 0.28 | 0.41 | 8.84 | 0.55 | 0.45 | 61.31 | 1.13 | 0.34 | 11.08 |
| Human fMRI | 0.17 | 0.09 | 0.45 | 0.10 | 0.11 | 1.33 | 0.16 | 0.19 | 0.95 | 0.23 | 0.06 | 0.78 |
| Human EEG | 0.79 | 0.23 | 1.07 | 0.23 | 0.44 | 0.70 | 1.80 | 0.55 | 0.89 | 1.23 | 0.26 | 0.91 |

Table 9: Performance comparison for zero-shot inference across all DS in the dataset from [30] using $T_C = 2000$. Best value in red, second best in blue.

| System | DynaMix $D_{\mathrm{stsp}}$ | $D_H$ | MAE | Chronos-t5-large $D_{\mathrm{stsp}}$ | $D_H$ | MAE | Chronos-t5-base $D_{\mathrm{stsp}}$ | $D_H$ | MAE | Chronos-t5-small $D_{\mathrm{stsp}}$ | $D_H$ | MAE | Chronos-t5-mini $D_{\mathrm{stsp}}$ | $D_H$ | MAE | Chronos-bolt-base $D_{\mathrm{stsp}}$ | $D_H$ | MAE | Chronos-bolt-small $D_{\mathrm{stsp}}$ | $D_H$ | MAE | Chronos-bolt-mini $D_{\mathrm{stsp}}$ | $D_H$ | MAE | Chronos-2 $D_{\mathrm{stsp}}$ | $D_H$ | MAE | Panda $D_{\mathrm{stsp}}$ | $D_H$ | MAE | Mamba4Cast $D_{\mathrm{stsp}}$ | $D_H$ | MAE | TTM $D_{\mathrm{stsp}}$ | $D_H$ | MAE | TimesFM $D_{\mathrm{stsp}}$ | $D_H$ | MAE |
|---|---|---|---|---|---|---|---|---|---|---|---|---|---|---|---|---|---|---|---|---|---|---|---|---|---|---|---|---|---|---|---|---|---|---|---|---|---|---|---|
| Atmospheric Regime | 10.4 | nan | 0.0 | 9.2 | 0.4 | 0.0 | 10.5 | 0.4 | 0.0 | 12.0 | 0.5 | 0.0 | 13.3 | 0.5 | 0.0 | 11.6 | 0.9 | 0.1 | 12.3 | 0.8 | 0.2 | 12.3 | 0.9 | 0.1 | 10.9 | 0.5 | 0.1 | 12.0 | 0.3 | 0.1 | 11.2 | 0.8 | 0.7 | 13.9 | 0.4 | 1.6 | 11.8 | 0.4 | 0.1 |
| BeerRNN | 5.2 | 0.7 | 0.6 | 12.1 | 0.5 | 0.5 | 13.3 | 0.4 | 0.7 | 10.9 | 0.4 | 0.7 | 10.9 | 0.3 | 0.6 | 12.4 | 0.9 | 1.0 | 12.7 | 0.8 | 1.7 | 11.9 | 0.5 | 1.5 | 13.5 | 0.3 | 0.3 | 14.0 | 0.3 | 0.4 | 11.4 | 0.8 | 1.0 | 14.2 | 0.4 | 1.5 | 14.0 | 0.3 | 0.5 |
| Bouali | 5.9 | 0.1 | 0.1 | 7.0 | 0.3 | 0.2 | 12.1 | 0.2 | 0.5 | 11.6 | 0.2 | 0.5 | 13.1 | 0.2 | 0.4 | 11.2 | 0.6 | 2.3 | 12.0 | 0.6 | 3.1 | 12.3 | 0.5 | 2.0 | 6.7 | 0.1 | 0.6 | 13.0 | 0.2 | 0.6 | 12.9 | 0.7 | 0.4 | 14.0 | 0.3 | 5.6 | 9.7 | 0.4 | 1.1 |
| Burke Shaw | 3.8 | 0.2 | 0.3 | 8.0 | 0.1 | 0.5 | 5.6 | 0.1 | 0.4 | 5.1 | 0.1 | 0.2 | 11.1 | 0.2 | 0.9 | 11.0 | 0.6 | 1.3 | 11.0 | 0.7 | 0.9 | 11.7 | 0.8 | 1.3 | 12.4 | 0.3 | 0.9 | 12.3 | 0.2 | 0.4 | 10.2 | 0.7 | 1.2 | 12.9 | 0.4 | 2.1 | 12.4 | 0.4 | 0.8 |
| Ca Two Plus Quasip. | 2.8 | 0.1 | 0.3 | 9.6 | 0.5 | 1.6 | 9.5 | 0.3 | 0.5 | 12.6 | 0.2 | 1.2 | 12.7 | 0.4 | 1.4 | 9.4 | 0.7 | 2.6 | 9.6 | 0.7 | 1.3 | 11.7 | 0.6 | 0.4 | 8.4 | 0.2 | 1.1 | 12.8 | 0.2 | 0.4 | 11.1 | 0.6 | 1.2 | 13.8 | 0.2 | 4.6 | 13.4 | 0.3 | 0.8 |
| Chen Lee | 3.6 | 0.3 | 0.2 | 4.2 | 0.2 | 0.0 | 3.9 | 0.1 | 0.1 | 3.9 | 0.3 | 0.1 | 5.3 | 0.3 | 0.1 | 4.5 | 0.8 | 0.4 | 8.8 | 0.7 | 0.9 | 7.7 | 0.7 | 0.6 | 3.9 | 0.2 | 0.2 | 6.2 | 0.2 | 0.6 | 11.3 | 0.8 | 0.8 | 8.9 | 0.3 | 2.0 | 4.6 | 0.2 | 1.9 |
| Chua | 2.9 | 0.1 | 0.8 | 12.9 | 0.2 | 0.6 | 12.4 | 0.4 | 0.8 | 13.3 | 0.4 | 0.2 | 13.5 | 0.3 | 1.0 | 11.9 | 0.6 | 2.4 | 11.7 | 0.4 | 1.3 | 11.0 | 0.5 | 1.8 | 12.9 | 0.2 | 1.9 | 11.6 | 0.3 | 0.6 | 12.1 | 0.4 | 4.0 | 13.4 | 0.3 | 4.0 | 12.7 | 0.2 | 1.0 |
| Coullet | 0.9 | 0.0 | 0.2 | 10.0 | 0.5 | 0.6 | 9.7 | 0.4 | 0.3 | 3.9 | 0.1 | 0.2 | 11.4 | 0.3 | 1.0 | 11.4 | 0.7 | 1.1 | 12.8 | 0.7 | 5.0 | 13.3 | 0.4 | 0.7 | 9.8 | 0.3 | 0.3 | 7.3 | 0.2 | 0.6 | 13.1 | 0.8 | 1.5 | 15.0 | 0.3 | 3.5 | 9.3 | 0.2 | 0.3 |
| Dadras | 2.2 | 0.2 | 1.3 | 12.6 | 0.5 | 4.2 | 13.2 | 0.5 | 1.9 | 13.1 | 0.3 | 0.2 | 11.4 | 0.3 | 3.5 | 10.5 | 0.8 | 5.4 | 9.4 | 0.7 | 5.0 | 12.2 | 0.4 | 6.8 | 11.1 | 0.3 | 0.6 | 12.4 | 0.4 | 2.8 | 13.1 | 0.8 | 5.0 | 12.2 | 0.4 | 6.0 | 13.1 | 0.6 | 4.5 |
| Dequan Li | 8.5 | 0.1 | 0.8 | 10.0 | 0.5 | 0.7 | 9.2 | 0.2 | 0.5 | 13.1 | 0.3 | 0.3 | 10.5 | 0.4 | 1.4 | 12.4 | 0.8 | 0.8 | 12.7 | 0.8 | 0.8 | 12.8 | 0.4 | 0.6 | 10.0 | 0.3 | 0.3 | 12.8 | 0.4 | 1.0 | 13.1 | 0.8 | 3.3 | 13.0 | 0.4 | 1.7 | 13.2 | 0.3 | 1.3 |
| Finance | 3.1 | 0.1 | 0.3 | 8.5 | 0.1 | 0.7 | 9.2 | 0.3 | 0.5 | 13.4 | 0.3 | 0.6 | 12.2 | 0.4 | 0.9 | 12.0 | 0.7 | 2.8 | 11.7 | 0.6 | 4.0 | 12.4 | 0.6 | 3.7 | 12.8 | 0.4 | 0.8 | 13.2 | 0.2 | 0.3 | 12.6 | 0.8 | 4.9 | 14.0 | 0.2 | 4.2 | 13.2 | 0.5 | 2.7 |
| Genesio Tesi | 9.8 | 0.2 | 0.4 | 8.9 | 0.1 | 0.2 | 9.4 | 0.2 | 0.1 | 7.7 | 0.1 | 0.1 | 11.8 | 0.4 | 0.4 | 10.7 | 0.6 | 0.8 | 10.6 | 0.9 | 1.6 | 10.6 | 0.6 | 2.0 | 11.5 | 0.4 | 0.6 | 11.5 | 0.1 | 0.3 | 10.4 | 0.8 | 1.3 | 12.3 | 0.2 | 1.7 | 12.2 | 0.2 | 1.1 |
| Guckenheimer Holmes | 7.4 | 0.0 | 0.2 | 10.6 | 0.4 | 0.3 | 11.7 | 0.2 | 0.3 | 11.3 | 0.4 | 0.3 | 13.2 | 0.4 | 0.6 | 10.6 | 0.6 | 0.7 | 9.2 | 0.6 | 1.0 | 9.2 | 0.8 | 0.4 | 6.8 | 0.1 | 0.3 | 9.1 | 0.2 | 0.7 | 10.4 | 0.5 | 2.0 | 11.6 | 0.3 | 3.2 | 11.0 | 0.3 | 0.7 |
| Hadley | 8.6 | 0.3 | 0.3 | 10.8 | 0.1 | 0.2 | 11.7 | 0.6 | 0.6 | 11.9 | 0.4 | 0.4 | 12.6 | 0.2 | 0.6 | 10.9 | 0.7 | 0.6 | 8.9 | 0.8 | 2.1 | 8.5 | 0.6 | 3.4 | 12.1 | 0.2 | 0.4 | 11.9 | 0.2 | 1.7 | 11.7 | 0.6 | 3.4 | 12.5 | 0.3 | 5.4 | 11.7 | 0.3 | 0.4 |
| Halvorsen | 1.8 | 0.2 | 0.2 | 6.0 | 0.1 | 0.1 | 8.4 | 0.3 | 0.5 | 5.9 | 0.1 | 0.1 | 1.5 | 0.2 | 0.5 | 11.8 | 0.8 | 1.6 | 13.9 | 0.8 | 2.1 | 13.9 | 0.8 | 2.5 | 13.0 | 0.3 | 0.3 | 8.2 | 0.2 | 1.8 | 12.2 | 0.6 | 2.5 | 13.2 | 0.3 | 2.8 | 14.2 | 0.6 | 1.5 |
| Hindmarsh Rose | 2.4 | 1.1 | 0.2 | 9.7 | 0.3 | 1.3 | 6.7 | 0.5 | 0.7 | 9.7 | 0.4 | 0.3 | 12.7 | 0.3 | 0.9 | 7.9 | 0.7 | 0.5 | 8.4 | 0.8 | 2.2 | 8.4 | 0.8 | 2.2 | 3.7 | 0.2 | 0.2 | 7.0 | 0.2 | 1.8 | 12.8 | 0.6 | 2.5 | 14.0 | 0.2 | 2.7 | 4.0 | 0.6 | 1.5 |
| Isothermal Chemical | 8.7 | 0.2 | 0.1 | 10.0 | 0.6 | 1.7 | 8.4 | 0.5 | 2.4 | 10.5 | 0.4 | 1.5 | 9.0 | 0.3 | 0.9 | 9.2 | 0.7 | 3.6 | 5.9 | 0.7 | 4.6 | 9.7 | 0.9 | 5.1 | 9.1 | 0.3 | 1.4 | 12.2 | 0.3 | 1.8 | 13.3 | 0.9 | 6.4 | 15.6 | 0.4 | 6.1 | 4.0 | 0.4 | 1.5 |
| Itik Banks tumor | 3.2 | 0.4 | 0.2 | 10.1 | 0.4 | 1.3 | 9.7 | 0.5 | 1.0 | 7.5 | 0.3 | 0.5 | 11.1 | 0.4 | 1.7 | 10.8 | 0.8 | 1.6 | 10.5 | 0.5 | 1.6 | 11.3 | 0.5 | 1.7 | 11.7 | 0.2 | 1.2 | 12.7 | 0.3 | 0.3 | 11.1 | 0.9 | 1.6 | 13.9 | 0.4 | 2.8 | 12.6 | 0.3 | 1.4 |
| Jerk Circuit | 8.8 | 0.4 | 0.2 | 11.1 | 0.4 | 0.1 | 12.8 | 0.2 | 0.1 | 13.8 | 0.2 | 0.3 | 12.0 | 0.5 | 0.6 | 10.9 | 0.7 | 0.5 | 11.3 | 0.5 | 0.5 | 12.7 | 0.6 | 0.5 | 10.9 | 0.2 | 0.2 | 12.7 | 0.3 | 0.3 | 11.4 | 0.6 | 1.2 | 13.0 | 0.2 | 2.2 | 13.1 | 0.5 | 0.8 |
| Laser | 1.2 | 0.1 | 0.2 | 6.6 | 0.2 | 0.4 | 10.0 | 0.3 | 0.2 | 12.8 | 0.2 | 0.3 | 11.8 | 0.4 | 1.1 | 11.3 | 0.8 | 0.5 | 11.3 | 0.5 | 0.5 | 11.4 | 0.9 | 1.1 | 12.8 | 0.2 | 0.2 | 13.1 | 0.2 | 0.2 | 11.4 | 0.6 | 1.2 | 13.1 | 0.2 | 2.2 | 13.1 | 0.5 | 0.8 |
| Lorenz | 6.9 | 0.4 | 1.5 | 9.5 | 0.4 | 2.2 | 11.5 | 0.4 | 1.9 | 12.2 | 0.4 | 1.9 | 12.9 | 0.4 | 1.7 | 12.9 | 0.8 | 2.4 | 11.5 | 0.8 | 2.7 | 13.4 | 0.9 | 2.5 | 13.4 | 0.7 | 1.6 | 13.4 | 0.4 | 0.8 | 13.4 | 0.8 | 2.5 | 13.7 | 0.5 | 2.4 | 13.4 | 0.5 | 1.5 |
| Lorenz 84 | 7.7 | 0.8 | 0.8 | 6.3 | 0.4 | 0.8 | 8.0 | 0.5 | 0.3 | 8.0 | 0.5 | 0.4 | 7.1 | 0.4 | 0.9 | 0.5 | 0.9 | 0.7 | 0.1 | 1.0 | 0.8 | 0.1 | 1.0 | 0.9 | 6.4 | 0.3 | 0.3 | 7.1 | 0.4 | 0.9 | 9.7 | 0.8 | 0.9 | 12.5 | 0.3 | 1.5 | 8.6 | 0.5 | 2.4 |
| Lu Chen | 7.7 | 0.8 | 0.3 | 10.4 | 0.4 | 2.2 | 10.0 | 0.5 | 0.4 | 12.5 | 0.4 | 0.7 | 13.3 | 0.4 | 0.8 | 11.4 | 0.9 | 1.5 | 12.2 | 0.6 | 1.6 | 10.1 | 0.9 | 1.6 | 12.6 | 0.5 | 0.3 | 6.4 | 0.4 | 0.4 | 13.4 | 0.8 | 0.9 | 13.7 | 0.5 | 2.4 | 13.4 | 0.5 | 1.5 |
| Moore Spiegel | 4.3 | 0.2 | 0.3 | 9.3 | 0.5 | 0.5 | 9.6 | 0.2 | 0.5 | 11.5 | 0.5 | 0.5 | 12.5 | 0.7 | 5.8 | 11.4 | 0.9 | 1.5 | 9.5 | 0.5 | 1.5 | 10.1 | 0.9 | 1.3 | 9.7 | 0.5 | 1.9 | 11.0 | 0.3 | 0.5 | 9.4 | 0.8 | 2.8 | 12.5 | 0.3 | 1.5 | 12.4 | 0.2 | 2.8 |
| Multi Chua | 8.1 | 0.3 | 0.8 | 9.3 | 0.5 | 0.5 | 9.6 | 0.3 | 0.5 | 11.5 | 0.4 | 0.5 | 12.5 | 0.2 | 0.7 | 12.2 | 0.6 | 5.8 | 10.6 | 0.9 | 0.6 | 9.5 | 0.5 | 11.3 | 9.7 | 0.3 | 1.3 | 11.0 | 0.3 | 0.5 | 0.6 | 0.8 | 4.8 | 14.0 | 0.4 | 7.6 | 11.4 | 0.2 | 2.0 |
| Nose Hoover | 9.5 | 0.0 | 0.1 | 9.9 | 0.5 | 0.5 | 10.1 | 0.4 | 0.8 | 11.5 | 0.3 | 0.3 | 10.5 | 0.7 | 1.1 | 10.6 | 0.9 | 1.0 | 10.2 | 0.9 | 1.7 | 10.1 | 0.7 | 1.0 | 11.2 | 0.3 | 1.3 | 8.6 | 0.3 | 0.3 | 0.2 | 0.7 | 1.4 | 12.9 | 0.4 | 1.6 | 13.3 | 0.3 | 1.4 |
| Pehlivan Wei | 9.2 | 0.0 | 0.2 | 8.5 | 0.4 | 0.5 | 10.1 | 0.5 | 0.5 | 7.5 | 0.3 | 0.3 | 11.1 | 0.4 | 0.8 | 10.1 | 0.7 | 1.2 | 9.2 | 0.7 | 1.7 | 9.2 | 0.7 | 1.0 | 8.6 | 0.2 | 0.5 | 8.6 | 0.2 | 0.4 | 9.3 | 0.8 | 1.4 | 12.2 | 0.6 | 3.0 | 8.2 | 0.3 | 1.2 |
| Sakarya | 2.0 | 0.2 | 0.7 | 12.9 | 0.2 | 1.2 | 12.7 | 0.3 | 0.2 | 13.3 | 0.4 | 0.3 | 11.1 | 0.4 | 2.2 | 10.0 | 0.6 | 2.2 | 7.7 | 0.6 | 2.1 | 13.4 | 0.6 | 1.6 | 13.4 | 0.5 | 1.2 | 12.2 | 0.2 | 0.2 | 13.1 | 0.8 | 2.5 | 12.6 | 0.4 | 1.2 | 11.9 | 0.3 | 3.7 |
| San Um Srisuchinwong | 3.6 | 0.2 | 0.3 | 8.0 | 0.2 | 1.1 | 8.9 | 0.3 | 0.5 | 6.6 | 0.2 | 2.1 | 13.3 | 0.5 | 1.8 | 12.7 | 0.7 | 0.9 | 10.0 | 0.7 | 2.1 | 12.5 | 0.3 | 1.4 | 12.5 | 0.3 | 0.3 | 11.9 | 0.3 | 0.1 | 10.7 | 0.7 | 2.5 | 13.1 | 0.3 | 3.7 | 13.3 | 0.7 | 1.9 |
| Sprott A | 2.7 | 0.1 | 0.3 | 8.0 | 0.2 | 1.1 | 12.7 | 0.3 | 2.1 | 8.9 | 0.2 | 0.9 | 12.0 | 0.8 | 1.9 | 8.1 | 0.7 | 1.9 | 11.9 | 0.5 | 1.5 | 9.3 | 0.5 | 1.4 | 9.3 | 0.5 | 0.3 | 12.2 | 0.3 | 1.1 | 9.9 | 0.7 | 2.5 | 12.0 | 0.3 | 3.7 | 9.8 | 0.2 | 1.0 |
| Sprott B | 2.9 | 0.2 | 0.4 | 4.3 | 0.2 | 1.1 | 5.6 | 0.4 | 0.9 | 11.7 | 0.5 | 1.3 | 12.7 | 0.8 | 1.3 | 11.5 | 0.6 | 6.8 | 8.1 | 0.5 | 6.8 | 12.5 | 0.2 | 0.7 | 13.1 | 0.5 | 0.7 | 11.9 | 0.1 | 0.1 | 12.5 | 0.8 | 3.0 | 13.5 | 0.2 | 5.7 | 13.5 | 0.6 | 0.5 |
| Sprott C | 2.4 | 0.1 | 0.2 | 6.3 | 0.4 | 2.3 | 12.4 | 0.4 | 1.3 | 11.9 | 0.5 | 0.7 | 12.9 | 0.6 | 1.3 | 10.0 | 0.8 | 1.3 | 11.5 | 0.6 | 5.2 | 13.7 | 0.5 | 2.3 | 13.7 | 0.5 | 2.3 | 12.8 | 0.3 | 1.1 | 13.3 | 0.6 | 5.2 | 13.3 | 0.2 | 6.2 | 13.4 | 0.2 | 0.5 |
| Sprott D | 3.0 | 0.1 | 0.2 | 10.4 | 0.6 | 0.3 | 10.0 | 0.4 | 0.4 | 12.9 | 0.3 | 2.6 | 13.4 | 0.3 | 0.7 | 9.9 | 0.7 | 7.7 | 10.9 | 0.6 | 1.3 | 9.1 | 0.2 | 1.1 | 9.1 | 0.2 | 1.1 | 9.8 | 0.4 | 0.3 | 12.9 | 0.6 | 1.3 | 14.0 | 0.4 | 4.0 | 7.9 | 0.2 | 4.5 |
| Sprott F | 5.0 | 0.3 | 0.3 | 9.6 | 0.2 | 0.2 | 9.2 | 0.3 | 0.6 | 10.3 | 0.3 | 0.8 | 12.6 | 0.4 | 1.3 | 10.4 | 0.7 | 1.3 | 10.2 | 0.6 | 0.6 | 9.5 | 0.5 | 1.2 | 12.3 | 0.2 | 0.3 | 12.7 | 0.5 | 0.3 | 1.4 | 0.8 | 1.6 | 13.3 | 0.6 | 2.8 | 13.0 | 0.6 | 1.7 |
| Sprott G | 3.8 | 0.1 | 0.3 | 7.0 | 0.6 | 0.7 | 7.7 | 0.5 | 0.2 | 10.0 | 0.4 | 0.6 | 11.9 | 0.3 | 1.9 | 10.9 | 0.9 | 1.9 | 9.4 | 0.9 | 1.9 | 8.0 | 0.2 | 0.7 | 8.0 | 0.1 | 0.7 | 9.2 | 0.3 | 0.3 | 4.1 | 0.7 | 2.0 | 13.1 | 0.2 | 1.9 | 8.3 | 0.4 | 1.4 |
| Sprott I | 2.7 | 0.1 | 0.5 | 9.7 | 0.5 | 0.7 | 11.3 | 0.2 | 0.6 | 9.0 | 0.3 | 0.6 | 11.9 | 0.3 | 3.5 | 9.8 | 0.8 | 3.5 | 8.4 | 0.8 | 4.9 | 8.6 | 0.4 | 2.2 | 7.2 | 0.4 | 0.4 | 7.5 | 0.3 | 0.4 | 2.2 | 0.7 | 2.0 | 14.4 | 0.2 | 8.8 | 8.3 | 0.4 | 1.0 |
| Sprott L | 3.6 | 0.1 | 0.6 | 8.0 | 0.5 | 0.3 | 6.8 | 0.4 | 0.3 | 8.2 | 0.2 | 1.8 | 11.6 | 0.3 | 1.8 | 10.4 | 0.8 | 2.8 | 11.4 | 0.9 | 2.8 | 8.8 | 0.6 | 3.7 | 8.8 | 0.2 | 6.1 | 11.9 | 0.1 | 0.4 | 10.8 | 0.8 | 2.9 | 13.4 | 0.4 | 2.9 | 8.1 | 0.4 | 0.2 |
| Sprott M | 3.6 | 0.2 | 0.2 | 9.8 | 0.1 | 0.4 | 10.5 | 0.4 | 0.4 | 12.4 | 0.1 | 3.7 | 12.9 | 0.4 | 3.7 | 10.4 | 0.8 | 3.7 | 9.9 | 0.8 | 2.8 | 12.5 | 0.3 | 1.6 | 12.5 | 0.2 | 0.8 | 7.2 | 0.3 | 0.3 | 11.6 | 0.7 | 1.6 | 14.3 | 0.4 | 6.5 | 13.4 | 0.4 | 1.6 |
| Sprott O | 2.0 | 0.1 | 1.4 | 3.6 | 0.1 | 0.4 | 4.6 | 0.0 | 0.4 | 9.7 | 0.1 | 0.5 | 12.4 | 0.2 | 2.6 | 10.8 | 1.0 | 1.6 | 10.4 | 0.9 | 2.6 | 9.3 | 0.2 | 0.7 | 7.2 | 0.1 | 0.2 | 7.2 | 0.2 | 0.3 | 10.7 | 0.7 | 1.6 | 14.0 | 0.4 | 3.7 | 9.5 | 0.4 | 0.7 |
| Sprott P | 3.8 | 0.2 | 1.4 | 10.8 | 0.3 | 3.9 | 11.9 | 0.4 | 2.6 | 12.7 | 0.3 | 2.6 | 12.7 | 0.3 | 2.6 | 10.6 | 0.5 | 5.7 | 10.6 | 0.7 | 9.6 | 10.5 | 0.3 | 8.7 | 10.5 | 0.3 | 8.7 | 10.9 | 0.2 | 0.9 | 11.1 | 0.8 | 6.3 | 12.2 | 0.4 | 10.2 | 11.8 | 0.3 | 4.4 |
| Sprott Q | 4.5 | 0.2 | 0.2 | 10.1 | 0.3 | 0.2 | 7.6 | 0.1 | 0.3 | 7.3 | 0.1 | 0.5 | 11.8 | 0.2 | 2.1 | 11.6 | 0.6 | 3.6 | 11.6 | 0.7 | 1.0 | 9.8 | 0.3 | 1.1 | 9.8 | 0.1 | 1.1 | 11.6 | 0.3 | 0.9 | 10.9 | 0.7 | 1.6 | 13.1 | 0.4 | 7.5 | 8.8 | 0.3 | 0.4 |
| Sprott R | 4.3 | 0.3 | 0.3 | 9.3 | 0.1 | 0.3 | 9.9 | 0.5 | 0.5 | 11.0 | 0.5 | 0.5 | 12.6 | 0.6 | 0.5 | 10.4 | 0.7 | 0.5 | 11.6 | 0.6 | 1.2 | 9.6 | 0.4 | 1.2 | 8.1 | 0.4 | 1.2 | 8.0 | 0.3 | 0.3 | 11.8 | 0.8 | 1.3 | 13.8 | 0.3 | 2.5 | 11.3 | 0.3 | 0.9 |
| Sprott S | 4.2 | 0.2 | 0.2 | 10.1 | 0.9 | 2.2 | 9.9 | 0.6 | 0.7 | 12.3 | 0.5 | 1.2 | 14.5 | 0.4 | 3.1 | 9.9 | 0.8 | 6.1 | 10.0 | 0.5 | 1.2 | 9.6 | 0.4 | 2.3 | 13.5 | 0.3 | 2.3 | 6.3 | 0.3 | 0.3 | 11.8 | 0.8 | 4.1 | 11.6 | 0.3 | 10.2 | 12.8 | 0.3 | 3.7 |
| Sprott Torus | 2.2 | 0.2 | 0.2 | 5.4 | 0.7 | 0.4 | 10.4 | 0.7 | 0.3 | 14.3 | 0.6 | 0.3 | 13.9 | 0.6 | 6.3 | 10.9 | 0.7 | 8.3 | 9.2 | 0.8 | 8.3 | 13.5 | 0.3 | 0.3 | 13.5 | 0.3 | 0.3 | 6.3 | 0.3 | 0.3 | 13.6 | 0.8 | 4.8 | 11.6 | 0.3 | 4.8 | 14.3 | 0.5 | 1.0 |
| Thomas | 1.6 | 0.3 | 4.8 | 10.6 | 0.5 | 6.0 | 10.1 | 0.8 | 6.0 | 9.9 | 0.4 | 5.7 | 13.9 | 0.5 | 4.7 | 6.2 | 0.8 | 0.6 | 9.4 | 0.9 | 1.9 | 5.3 | 0.8 | 2.8 | 13.2 | 0.3 | 3.4 | 13.7 | 0.3 | 3.5 | 0.7 | 0.5 | 6.1 | 13.3 | 0.2 | 3.1 | 13.8 | 0.6 | 3.3 |
| Thomas Labyrinth | 4.1 | 2.4 | 1.1 | 12.5 | 0.2 | 2.6 | 12.5 | 0.3 | 3.0 | 12.4 | 0.4 | 3.0 | 12.3 | 0.4 | 2.6 | 11.6 | 0.5 | 2.6 | 10.9 | 0.8 | 1.9 | 12.8 | 0.5 | 3.4 | 12.1 | 0.3 | 3.0 | 12.6 | 0.2 | 6.1 | 12.3 | 0.5 | 2.4 | 12.4 | 0.5 | 2.0 | 12.6 | 0.5 | 2.5 |
| Torus | 11.9 | 0.9 | 1.1 | 9.8 | 0.1 | 0.1 | 9.4 | 0.2 | 0.1 | 10.9 | 0.4 | 0.1 | 12.3 | 0.3 | 0.2 | 10.6 | 0.6 | 0.8 | 11.2 | 0.8 | 0.9 | 11.6 | 0.8 | 0.4 | 11.5 | 0.3 | 0.3 | 12.6 | 0.2 | 0.3 | 12.3 | 0.8 | 0.8 | 12.6 | 0.5 | 1.1 | 10.9 | 0.5 | 0.4 |
| Tsucs 2 | 1.1 | 0.1 | 0.5 | 11.6 | 0.3 | 0.7 | 11.2 | 0.5 | 0.7 | 10.6 | 0.3 | 0.8 | 11.6 | 0.6 | 0.4 | 8.2 | 0.7 | 1.0 | 8.2 | 0.7 | 1.0 | 4.8 | 0.3 | 0.3 | 4.8 | 0.3 | 0.3 | 7.8 | 0.2 | 0.2 | 8.4 | 0.8 | 0.8 | 10.9 | 0.5 | 1.2 | 13.2 | 0.5 | 0.5 |
| Vallise El Nino | 4.9 | nan | 0.6 | 8.4 | 0.3 | 0.5 | 11.2 | 0.5 | 1.0 | 12.5 | 0.5 | 1.0 | 12.7 | 0.4 | 1.9 | 12.3 | 0.7 | 1.4 | 13.9 | 0.7 | 1.7 | 13.2 | 0.4 | 1.3 | 13.2 | 0.4 | 0.4 | 13.1 | 0.2 | 0.2 | 13.9 | 0.8 | 3.0 | 13.7 | 0.4 | 3.8 | 13.3 | 0.5 | 2.3 |
| Wang Sun | 4.9 | 0.1 | 0.6 | 13.2 | 0.4 | 0.7 | 12.2 | 0.5 | 1.0 | 11.8 | 0.3 | 0.6 | 13.7 | 0.6 | 3.3 | 10.9 | 0.7 | 1.5 | 10.9 | 0.7 | 1.7 | 12.7 | 0.4 | 1.3 | 12.7 | 0.4 | 0.4 | 12.3 | 0.2 | 0.5 | 13.3 | 0.8 | 1.6 | 11.1 | 0.4 | 1.0 | 12.3 | 0.5 | 0.9 |
| Winsmi Reduced | 8.3 | 0.5 | 0.6 | 11.4 | 0.4 | 0.6 | 12.2 | 0.5 | 0.1 | 13.5 | 0.5 | 0.1 | 13.7 | 0.6 | 3.8 | 10.9 | 0.8 | 3.5 | 10.4 | 0.6 | 2.6 | 10.7 | 0.2 | 0.6 | 10.9 | 0.2 | 0.6 | 12.0 | 0.8 | 0.6 | 12.0 | 0.8 | 3.3 | 14.2 | 0.4 | 4.7 | 13.6 | 0.2 | 1.2 |
| Yu Wang | 4.9 | 0.3 | 0.3 | 9.8 | 0.3 | 0.6 | 8.4 | 0.3 | 0.3 | 12.4 | 0.3 | 0.3 | 13.7 | 0.4 | 3.3 | 12.4 | 0.7 | 3.7 | 10.9 | 0.6 | 4.4 | 13.1 | 0.2 | 1.6 | 12.8 | 0.2 | 0.6 | 12.5 | 0.2 | 3.4 | 12.5 | 0.7 | 2.9 | 13.5 | 0.2 | 3.4 | 13.2 | 0.5 | 2.6 |
| Yu Wang 2 | 2.7 | 0.1 | 0.3 | 9.6 | 0.5 | 0.6 | 10.0 | 0.6 | 0.2 | 12.4 | 0.3 | 0.5 | 12.4 | 0.3 | 2.3 | 9.9 | 0.6 | 0.5 | 9.5 | 0.7 | 1.5 | 13.1 | 0.2 | 0.9 | 9.0 | 0.2 | 0.2 | 11.5 | 0.3 | 0.6 | 0.6 | 0.7 | 2.9 | 12.6 | 0.2 | 3.4 | 10.6 | 0.5 | 1.7 |
| Zhou Chen | 4.6 | nan | 0.4 | 11.1 | 0.3 | 1.0 | 5.9 | 0.4 | 0.5 | 11.6 | 0.3 | 1.8 | 14.2 | 0.4 | 1.5 | 12.8 | 0.7 | 1.5 | 12.7 | 0.5 | 2.1 | 13.5 | 0.4 | 1.3 | 12.8 | 0.3 | 0.3 | 9.1 | 0.3 | 0.3 | 13.1 | 0.5 | 3.7 | 13.2 | 0.2 | 5.1 | 14.9 | 0.4 | 1.3 |

Table 10: Performance comparison for zero-shot inference across all DS in the dataset from [30] using $T_C = 512$. Best value in red, second best in blue.

| System | DynaMix $D_{ssp}$ | $D_H$ | MAE | Chronos-t5-large $D_{ssp}$ | $D_H$ | MAE | Chronos-t5-base $D_{ssp}$ | $D_H$ | MAE | Chronos-t5-small $D_{ssp}$ | $D_H$ | MAE | Chronos-t5-mini $D_{ssp}$ | $D_H$ | MAE | Chronos-bolt-base $D_{ssp}$ | $D_H$ | MAE | Chronos-bolt-small $D_{ssp}$ | $D_H$ | MAE | Chronos-bolt-mini $D_{ssp}$ | $D_H$ | MAE | Chronos-2 $D_{ssp}$ | $D_H$ | MAE | Panda $D_{ssp}$ | $D_H$ | MAE | Mamba4Cast $D_{ssp}$ | $D_H$ | MAE | TTM $D_{ssp}$ | $D_H$ | MAE | TimesFM $D_{ssp}$ | $D_H$ | MAE |
|---|---|---|---|---|---|---|---|---|---|---|---|---|---|---|---|---|---|---|---|---|---|---|---|---|---|---|---|---|---|---|---|---|---|---|---|---|---|---|---|
| Atmospheric Regime | 8.4 | 0.6 | 2.3 | 9.7 | 0.5 | 0.9 | 11.1 | 0.4 | 0.5 | 10.5 | 0.6 | 0.7 | 10.1 | 0.5 | 1.0 | 11.9 | 0.9 | 2.2 | 12.0 | 0.9 | 1.9 | 13.5 | 0.9 | 1.8 | 10.2 | 0.3 | 0.8 | 13.2 | 0.3 | 1.0 | 11.3 | 0.9 | 1.3 | 14.2 | 0.6 | 3.1 | 13.9 | 0.4 | 1.2 |
| BeerRNN | 11.7 | 0.8 | 1.2 | 8.0 | 0.5 | 0.2 | 3.3 | 0.2 | 0.2 | 9.5 | 0.2 | 0.2 | 7.3 | 0.4 | 0.2 | 12.3 | 0.9 | 1.2 | 12.5 | 0.9 | 1.2 | 12.1 | 0.9 | 1.2 | 12.6 | 0.4 | 1.3 | 13.9 | 0.3 | 0.5 | 13.0 | 0.9 | 1.4 | 14.1 | 0.6 | 1.3 | 14.0 | 0.4 | 0.8 |
| Bouali | 4.1 | 0.1 | 0.5 | 8.3 | 0.5 | 0.5 | 7.3 | 0.3 | 0.3 | 8.6 | 0.1 | 0.2 | 11.1 | 0.1 | 0.5 | 9.6 | 0.8 | 1.0 | 9.6 | 0.7 | 1.1 | 12.2 | 0.7 | 2.2 | 7.6 | 0.4 | 0.5 | 13.1 | 0.4 | 0.5 | 12.9 | 0.8 | 2.3 | 13.4 | 0.4 | 2.3 | 12.7 | 0.3 | 0.7 |
| Burke Shaw | 4.1 | 0.2 | 0.5 | 10.8 | 0.5 | 1.2 | 9.0 | 0.1 | 1.5 | 13.0 | 0.4 | 1.1 | 13.2 | 0.4 | 1.5 | 12.1 | 0.8 | 1.3 | 12.1 | 0.8 | 2.4 | 12.2 | 0.7 | 2.4 | 11.1 | 0.4 | 1.0 | 13.4 | 0.4 | 0.4 | 12.2 | 0.9 | 3.7 | 13.7 | 0.5 | 3.6 | 13.9 | 0.4 | 1.5 |
| Ca Two Plus Quasip. | 2.3 | 0.1 | 0.7 | 7.2 | 0.1 | 0.2 | 8.3 | 0.5 | 0.5 | 9.2 | 0.4 | 0.3 | 13.1 | 0.4 | 0.2 | 11.6 | 0.9 | 1.6 | 11.3 | 0.9 | 1.1 | 12.8 | 0.7 | 1.2 | 7.0 | 0.3 | 1.0 | 12.3 | 0.2 | 0.3 | 11.1 | 0.8 | 3.0 | 13.8 | 0.5 | 3.6 | 13.4 | 0.4 | 1.9 |
| Chen Lee | 3.9 | 0.4 | 0.3 | 4.9 | 0.3 | 0.3 | 4.0 | 0.5 | 0.5 | 4.8 | 0.3 | 0.3 | 4.6 | 0.3 | 0.3 | 7.9 | 0.6 | 1.6 | 7.9 | 0.7 | 1.1 | 8.0 | 0.7 | 2.0 | 4.9 | 0.3 | 0.2 | 6.2 | 0.3 | 0.3 | 11.0 | 0.9 | 0.9 | 6.4 | 0.4 | 2.6 | 13.4 | 0.3 | 1.9 |
| Chua | 5.9 | 0.1 | 0.2 | 7.4 | 0.2 | 0.6 | 5.9 | 0.2 | 0.7 | 7.6 | 0.7 | 0.7 | 11.7 | 0.2 | 0.4 | 12.2 | 0.6 | 1.9 | 12.2 | 0.6 | 3.8 | 11.7 | 0.8 | 1.7 | 9.4 | 0.2 | 0.5 | 12.2 | 0.4 | 0.6 | 11.3 | 0.5 | 3.4 | 13.4 | 0.4 | 2.7 | 10.7 | 0.2 | 0.5 |
| Coullet | 1.2 | 0.0 | 0.2 | 4.9 | 0.0 | 0.3 | 0.7 | 0.0 | 0.2 | 2.8 | 0.0 | 0.1 | 7.5 | 0.2 | 0.2 | 13.0 | 0.8 | 2.8 | 11.8 | 0.8 | 2.8 | 11.8 | 0.8 | 2.0 | 9.8 | 0.2 | 0.2 | 9.7 | 0.3 | 0.3 | 12.3 | 0.8 | 2.4 | 14.9 | 0.5 | 3.3 | 10.7 | 0.2 | 0.5 |
| Dadras | 4.1 | 0.2 | 1.1 | 7.7 | 0.6 | 0.5 | 4.7 | 0.5 | 0.5 | 13.1 | 0.3 | 0.3 | 12.1 | 0.4 | 0.2 | 12.9 | 0.8 | 1.7 | 12.9 | 0.7 | 2.8 | 13.0 | 0.4 | 2.2 | 9.2 | 0.5 | 0.5 | 12.0 | 0.4 | 0.9 | 11.9 | 0.8 | 1.7 | 12.8 | 0.4 | 2.9 | 11.8 | 0.4 | 0.5 |
| Dequan Li | 12.7 | 0.5 | 2.1 | 11.1 | 1.1 | 1.1 | 11.0 | 0.5 | 1.0 | 12.1 | 0.9 | 0.9 | 12.7 | 0.4 | 0.5 | 11.7 | 0.8 | 3.1 | 12.3 | 0.7 | 2.3 | 13.4 | 0.4 | 3.1 | 11.5 | 0.3 | 1.3 | 13.3 | 0.2 | 0.9 | 11.9 | 0.8 | 3.1 | 13.5 | 0.4 | 4.5 | 12.8 | 0.3 | 1.2 |
| Finance | 3.2 | 0.1 | 0.6 | 8.5 | 0.6 | 1.6 | 8.6 | 0.4 | 0.4 | 9.1 | 0.4 | 0.4 | 11.6 | 0.2 | 0.4 | 10.7 | 0.8 | 1.4 | 10.7 | 0.8 | 2.0 | 13.4 | 0.8 | 1.7 | 11.0 | 0.3 | 1.0 | 13.3 | 0.3 | 0.8 | 11.1 | 0.9 | 3.1 | 13.7 | 0.5 | 4.2 | 13.6 | 0.4 | 3.1 |
| Genesio Tesi | 8.2 | 0.2 | 0.9 | 7.9 | 0.4 | 1.0 | 8.5 | 0.2 | 0.2 | 7.7 | 0.4 | 1.0 | 10.4 | 0.3 | 0.2 | 10.4 | 0.7 | 0.6 | 10.6 | 0.7 | 1.8 | 10.3 | 0.8 | 2.5 | 7.4 | 0.3 | 1.2 | 10.8 | 0.4 | 0.4 | 11.0 | 0.7 | 1.4 | 12.6 | 0.5 | 5.4 | 10.6 | 0.4 | 1.0 |
| Guckenheimer Holmes | 9.6 | 0.1 | 0.5 | 11.8 | 0.1 | 0.8 | 12.0 | 0.1 | 0.4 | 12.6 | 0.3 | 0.5 | 9.6 | 0.4 | 1.1 | 9.6 | 0.7 | 0.6 | 9.6 | 0.7 | 2.5 | 9.4 | 0.6 | 4.7 | 8.2 | 0.1 | 0.1 | 10.8 | 0.3 | 0.6 | 10.2 | 0.7 | 1.0 | 12.7 | 0.5 | 3.5 | 11.5 | 0.3 | 0.9 |
| Hadley | 11.3 | 0.3 | 0.3 | 10.7 | 0.5 | 2.3 | 9.7 | 0.5 | 0.7 | 12.6 | 0.3 | 1.5 | 12.6 | 0.6 | 1.8 | 11.9 | 0.8 | 5.1 | 11.9 | 0.8 | 4.3 | 11.9 | 0.6 | 4.6 | 10.9 | 0.2 | 4.7 | 12.6 | 0.3 | 0.8 | 11.4 | 0.9 | 6.3 | 12.5 | 0.5 | 7.1 | 12.4 | 0.5 | 3.8 |
| Halvorsen | 4.3 | 0.3 | 0.6 | 9.7 | 0.6 | 0.5 | 8.2 | 0.2 | 0.7 | 12.5 | 0.2 | 0.7 | 13.1 | 0.2 | 0.6 | 10.3 | 0.8 | 5.0 | 10.3 | 0.8 | 4.3 | 9.7 | 0.8 | 4.3 | 11.8 | 0.3 | 2.3 | 13.1 | 0.2 | 0.3 | 13.3 | 0.9 | 3.4 | 13.4 | 0.5 | 3.5 | 13.3 | 0.3 | 1.6 |
| Hindmarsh Rose | 6.5 | 0.4 | 0.7 | 9.8 | 0.2 | 0.2 | 4.9 | 0.1 | 0.1 | 9.6 | 0.1 | 0.2 | 12.7 | 0.6 | 0.3 | 11.6 | 0.7 | 1.6 | 14.3 | 0.7 | 2.0 | 12.9 | 0.8 | 1.7 | 8.1 | 0.3 | 0.6 | 9.6 | 0.3 | 0.3 | 13.2 | 0.8 | 2.5 | 14.6 | 0.6 | 1.4 | 14.3 | 0.5 | 1.1 |
| Isothermal Chemical | 2.0 | 0.2 | 0.2 | 2.5 | 0.1 | 0.2 | 4.0 | 0.1 | 0.3 | 8.3 | 0.2 | 0.2 | 7.0 | 0.4 | 0.2 | 7.0 | 0.7 | 1.1 | 10.8 | 0.8 | 1.6 | 10.8 | 0.8 | 1.6 | 6.5 | 0.3 | 0.4 | 5.8 | 0.3 | 0.3 | 0.1 | 0.9 | 2.9 | 15.6 | 0.6 | 3.5 | 7.5 | 0.3 | 1.3 |
| Itik Banks tumor | 5.7 | 0.2 | 0.5 | 8.2 | 0.2 | 0.5 | 5.4 | 0.2 | 0.1 | 9.8 | 0.3 | 0.3 | 11.8 | 0.6 | 0.4 | 11.7 | 0.6 | 3.4 | 11.5 | 0.9 | 3.8 | 11.9 | 0.9 | 2.9 | 9.8 | 0.4 | 0.3 | 12.5 | 0.4 | 0.4 | 12.0 | 0.9 | 2.9 | 12.9 | 0.7 | 4.1 | 13.4 | 0.3 | 1.1 |
| Jerk Circuit | 4.3 | 0.3 | 0.8 | 7.5 | 0.6 | 0.5 | 9.1 | 0.6 | 0.6 | 9.1 | 0.2 | 0.3 | 11.7 | 0.3 | 0.3 | 11.7 | 0.7 | 1.0 | 12.5 | 0.9 | 2.8 | 12.9 | 0.9 | 0.8 | 10.6 | 0.4 | 0.7 | 13.5 | 0.4 | 0.5 | 11.8 | 0.8 | 0.6 | 14.1 | 0.5 | 1.9 | 13.8 | 0.4 | 1.1 |
| Laser | 4.5 | 0.2 | 0.2 | 9.8 | 0.5 | 3.0 | 8.2 | 0.3 | 0.3 | 11.5 | 0.2 | 0.5 | 12.8 | 0.2 | 0.4 | 13.0 | 0.8 | 2.0 | 13.0 | 0.5 | 2.8 | 13.0 | 0.7 | 1.7 | 7.4 | 0.4 | 0.4 | 13.1 | 0.3 | 0.3 | 11.8 | 0.8 | 2.5 | 13.3 | 0.5 | 2.6 | 13.8 | 0.4 | 1.0 |
| Lorenz | 3.4 | 0.2 | 0.5 | 10.7 | 0.5 | 1.5 | 8.5 | 0.2 | 0.3 | 13.0 | 0.7 | 0.7 | 12.6 | 0.7 | 0.5 | 12.6 | 0.9 | 2.0 | 12.2 | 0.7 | 1.3 | 13.2 | 0.6 | 3.0 | 11.8 | 0.4 | 0.8 | 12.7 | 0.3 | 0.3 | 11.8 | 0.8 | 2.7 | 13.1 | 0.5 | 3.1 | 13.1 | 0.3 | 2.1 |
| Lorenz 84 | 7.9 | 0.2 | 0.5 | 5.3 | 0.2 | 1.5 | 11.0 | 0.4 | 0.4 | 8.9 | 0.2 | 1.0 | 11.0 | 0.4 | 0.4 | 12.0 | 0.7 | 1.3 | 12.2 | 0.8 | 1.4 | 11.5 | 0.9 | 1.3 | 7.4 | 0.2 | 0.4 | 13.4 | 0.3 | 0.3 | 11.2 | 0.9 | 1.2 | 13.4 | 0.6 | 1.2 | 13.4 | 0.5 | 1.4 |
| Lu Chen | 9.5 | 0.8 | 0.8 | 9.8 | 0.5 | 0.4 | 9.0 | 0.3 | 0.3 | 12.6 | 0.7 | 1.0 | 9.7 | 0.3 | 0.5 | 11.5 | 1.0 | 1.7 | 11.7 | 0.8 | 2.7 | 12.7 | 0.9 | 1.6 | 11.8 | 0.2 | 0.8 | 13.7 | 0.3 | 0.4 | 11.2 | 0.9 | 1.1 | 14.2 | 0.7 | 1.9 | 12.6 | 0.5 | 0.9 |
| Moore-Spiegel | 9.2 | 0.6 | 0.4 | 9.1 | 0.4 | 0.5 | 12.0 | 0.2 | 0.4 | 12.5 | 0.3 | 0.2 | 11.5 | 0.6 | 0.5 | 11.5 | 0.9 | 2.0 | 13.8 | 0.9 | 4.6 | 13.6 | 0.9 | 3.1 | 7.0 | 0.5 | 1.2 | 11.7 | 0.3 | 0.3 | 0.1 | 0.9 | 2.2 | 13.1 | 0.7 | 1.5 | 12.7 | 0.5 | 2.1 |
| Multi Chua | 12.5 | 0.5 | 0.9 | 12.0 | 0.6 | 1.1 | 11.9 | 0.6 | 0.6 | 13.4 | 0.4 | 1.0 | 13.0 | 0.4 | 0.8 | 12.6 | 0.5 | 1.5 | 12.6 | 0.5 | 1.4 | 13.6 | 0.5 | 1.2 | 11.4 | 0.2 | 0.5 | 13.0 | 0.2 | 0.3 | 10.7 | 0.5 | 0.7 | 13.5 | 0.5 | 2.1 | 12.7 | 0.3 | 1.9 |
| Nose Hoover | 11.4 | 0.5 | 1.9 | 10.2 | 0.9 | 3.2 | 10.8 | 0.5 | 0.9 | 10.8 | 0.6 | 0.5 | 12.9 | 0.8 | 0.6 | 11.2 | 0.9 | 2.0 | 12.0 | 0.9 | 1.4 | 10.8 | 0.7 | 6.8 | 13.1 | 0.2 | 0.8 | 12.9 | 0.3 | 0.3 | 10.0 | 0.9 | 0.5 | 13.3 | 0.5 | 3.3 | 12.8 | 0.3 | 1.5 |
| Pehlivan Wei | 4.8 | 0.2 | 1.9 | 4.6 | 0.1 | 0.7 | 6.8 | 0.4 | 0.4 | 9.0 | 0.3 | 1.0 | 0.7 | 0.3 | 1.3 | 12.0 | 0.9 | 5.6 | 12.0 | 0.9 | 2.5 | 12.6 | 0.8 | 6.5 | 8.1 | 0.2 | 0.5 | 8.3 | 0.4 | 0.4 | 12.4 | 0.9 | 4.1 | 12.7 | 0.5 | 7.7 | 9.1 | 0.3 | 3.3 |
| Sakarya | 3.1 | 0.3 | 0.6 | 2.6 | 0.2 | 0.2 | 5.0 | 0.9 | 0.8 | 8.2 | 0.3 | 0.6 | 10.7 | 0.3 | 0.6 | 10.7 | 0.7 | 0.5 | 12.7 | 0.9 | 6.8 | 12.6 | 0.8 | 0.9 | 10.8 | 0.5 | 0.5 | 11.5 | 0.3 | 0.3 | 0.0 | 0.9 | 0.8 | 14.0 | 0.5 | 2.5 | 13.8 | 0.3 | 1.5 |
| San Um Srisuchinwong | 5.5 | 0.1 | 0.6 | 5.5 | 0.2 | 1.2 | 10.5 | 0.8 | 1.3 | 7.5 | 0.3 | 0.2 | 13.0 | 0.5 | 0.8 | 13.0 | 0.8 | 4.1 | 13.0 | 0.7 | 0.5 | 12.0 | 0.5 | 2.5 | 10.3 | 0.2 | 2.2 | 13.0 | 0.3 | 0.2 | 12.4 | 0.9 | 1.1 | 14.0 | 0.6 | 0.9 | 11.2 | 0.2 | 1.4 |
| Sprott A | 4.9 | 0.1 | 1.0 | 9.3 | 0.3 | 0.5 | 8.5 | 0.6 | 0.8 | 8.5 | 0.3 | 0.4 | 11.7 | 0.3 | 0.8 | 11.8 | 0.8 | 4.1 | 11.8 | 0.7 | 4.5 | 12.0 | 0.6 | 2.1 | 10.4 | 0.2 | 1.1 | 13.0 | 0.3 | 0.4 | 9.8 | 0.8 | 2.1 | 12.6 | 0.6 | 6.0 | 11.2 | 0.2 | 1.1 |
| Sprott B | 2.6 | 0.2 | 0.9 | 9.2 | 0.3 | 0.5 | 11.0 | 0.6 | 0.6 | 13.2 | 0.4 | 0.3 | 12.6 | 0.4 | 1.5 | 12.6 | 0.8 | 2.7 | 12.5 | 0.8 | 2.7 | 12.5 | 0.6 | 4.4 | 13.7 | 0.7 | 2.7 | 12.9 | 0.3 | 0.6 | 13.9 | 0.8 | 3.9 | 12.4 | 0.6 | 6.0 | 13.7 | 0.3 | 2.5 |
| Sprott C | 2.0 | 0.2 | 0.5 | 8.5 | 0.3 | 0.6 | 12.0 | 0.5 | 1.1 | 13.3 | 0.2 | 0.8 | 12.4 | 0.4 | 0.8 | 13.0 | 0.6 | 1.2 | 11.9 | 0.6 | 2.3 | 12.7 | 0.5 | 2.5 | 11.6 | 0.7 | 0.7 | 12.9 | 0.3 | 0.3 | 13.5 | 0.8 | 2.6 | 12.9 | 0.5 | 2.9 | 13.5 | 0.4 | 1.9 |
| Sprott D | 3.8 | 0.1 | 0.6 | 7.6 | 0.2 | 0.3 | 5.5 | 0.1 | 0.2 | 5.5 | 0.1 | 1.1 | 12.9 | 0.4 | 0.7 | 12.1 | 0.7 | 0.8 | 12.1 | 0.7 | 4.6 | 12.1 | 0.7 | 3.1 | 10.4 | 0.3 | 1.2 | 12.0 | 0.4 | 0.4 | 12.3 | 0.7 | 1.9 | 13.8 | 0.5 | 2.3 | 13.2 | 0.5 | 0.6 |
| Sprott F | 4.9 | 0.3 | 0.5 | 10.8 | 0.4 | 0.3 | 6.2 | 0.2 | 0.5 | 12.6 | 0.2 | 0.3 | 13.0 | 0.3 | 0.2 | 13.0 | 0.8 | 1.2 | 12.4 | 0.6 | 1.3 | 12.9 | 0.8 | 1.2 | 13.2 | 0.5 | 0.5 | 12.0 | 0.3 | 0.3 | 11.8 | 0.9 | 2.6 | 13.4 | 0.5 | 1.8 | 10.6 | 0.3 | 1.2 |
| Sprott G | 2.4 | 0.0 | 0.3 | 4.2 | 0.1 | 0.1 | 8.9 | 0.1 | 0.2 | 7.1 | 0.2 | 0.2 | 7.7 | 0.3 | 0.5 | 3.6 | 0.9 | 3.6 | 12.4 | 0.7 | 4.0 | 12.9 | 0.8 | 2.5 | 6.6 | 0.3 | 0.6 | 10.5 | 0.2 | 0.2 | 11.0 | 0.9 | 1.4 | 14.0 | 0.6 | 2.0 | 9.1 | 0.4 | 0.9 |
| Sprott I | 3.0 | 0.2 | 0.2 | 8.6 | 0.5 | 0.2 | 10.8 | 0.1 | 0.2 | 7.4 | 0.3 | 0.0 | 10.1 | 0.3 | 1.0 | 11.9 | 0.8 | 1.2 | 11.9 | 0.8 | 2.5 | 9.4 | 0.9 | 1.0 | 8.6 | 0.3 | 0.3 | 8.2 | 0.2 | 0.1 | 10.9 | 0.9 | 0.6 | 14.4 | 0.6 | 3.2 | 9.1 | 0.4 | 0.3 |
| Sprott L | 2.7 | 0.2 | 0.2 | 9.2 | 0.2 | 0.2 | 8.2 | 0.1 | 0.2 | 10.2 | 0.3 | 0.1 | 11.3 | 0.2 | 0.3 | 12.6 | 0.9 | 2.6 | 12.3 | 0.9 | 2.3 | 13.3 | 0.9 | 1.8 | 6.7 | 0.2 | 1.8 | 12.0 | 0.2 | 0.2 | 12.2 | 0.8 | 0.8 | 14.2 | 0.5 | 2.5 | 11.7 | 0.4 | 0.3 |
| Sprott M | 3.4 | 0.2 | 0.2 | 10.9 | 0.2 | 1.9 | 12.4 | 0.2 | 0.4 | 12.8 | 0.4 | 0.2 | 13.5 | 0.4 | 0.6 | 12.2 | 0.8 | 1.2 | 12.3 | 0.8 | 2.1 | 13.0 | 0.7 | 1.2 | 7.6 | 0.2 | 0.7 | 13.7 | 0.2 | 0.2 | 10.7 | 0.9 | 1.8 | 14.3 | 0.7 | 3.2 | 13.5 | 0.5 | 0.6 |
| Sprott O | 3.5 | 0.1 | 0.4 | 9.2 | 0.2 | 0.4 | 6.7 | 0.1 | 0.1 | 8.1 | 0.4 | 0.2 | 11.6 | 0.4 | 0.5 | 11.1 | 0.8 | 1.9 | 12.7 | 0.8 | 2.1 | 11.4 | 1.0 | 2.1 | 9.5 | 0.3 | 0.3 | 12.0 | 0.1 | 0.1 | 10.4 | 0.8 | 1.4 | 14.0 | 0.7 | 3.2 | 9.6 | 0.5 | 0.6 |
| Sprott P | 5.7 | 0.2 | 0.4 | 9.2 | 0.5 | 0.9 | 10.5 | 0.4 | 0.1 | 11.1 | 0.4 | 0.6 | 13.3 | 0.4 | 0.5 | 13.3 | 0.8 | 6.8 | 7.6 | 0.6 | 5.7 | 8.7 | 0.8 | 2.1 | 7.6 | 0.2 | 0.2 | 11.4 | 0.3 | 0.1 | 10.8 | 0.8 | 1.8 | 10.4 | 0.4 | 3.2 | 9.1 | 0.5 | 1.9 |
| Sprott Q | 5.8 | 0.2 | 0.8 | 9.5 | 0.5 | 0.5 | 9.4 | 0.5 | 0.3 | 10.0 | 0.7 | 0.4 | 11.0 | 0.5 | 1.3 | 13.0 | 0.8 | 2.8 | 7.6 | 0.7 | 2.2 | 10.6 | 0.7 | 2.2 | 7.6 | 0.2 | 0.4 | 9.8 | 0.3 | 0.4 | 9.7 | 0.9 | 1.1 | 13.2 | 0.4 | 6.0 | 8.9 | 0.3 | 0.8 |
| Sprott R | 5.1 | 0.3 | 0.7 | 6.9 | 0.4 | 0.3 | 6.7 | 0.3 | 0.3 | 12.5 | 0.2 | 0.5 | 10.2 | 0.3 | 0.7 | 8.8 | 0.9 | 2.8 | 13.0 | 0.9 | 2.2 | 10.6 | 0.9 | 2.2 | 7.4 | 0.3 | 1.3 | 9.8 | 0.4 | 0.2 | 1.4 | 0.9 | 2.2 | 13.8 | 0.6 | 6.0 | 12.3 | 0.4 | 1.6 |
| Sprott S | 2.9 | 0.0 | 0.0 | 9.6 | 0.6 | 0.6 | 10.7 | 0.5 | 0.5 | 11.7 | 0.5 | 0.3 | 0.2 | 0.7 | 0.2 | 10.2 | 0.9 | 0.8 | 12.5 | 0.9 | 3.4 | 11.9 | 0.9 | 2.2 | 11.7 | 0.3 | 0.7 | 12.9 | 0.5 | 0.5 | 11.4 | 0.9 | 2.2 | 12.8 | 0.5 | 6.0 | 11.3 | 0.4 | 1.9 |
| Sprott Torus | 3.6 | 0.2 | 0.1 | 6.8 | 0.5 | 0.5 | 12.0 | 0.3 | 0.5 | 13.0 | 0.3 | 0.5 | 7.0 | 0.3 | 0.3 | 7.9 | 0.9 | 1.1 | 7.9 | 0.9 | 1.1 | 13.7 | 0.9 | 5.2 | 10.7 | 0.4 | 0.6 | 8.5 | 0.3 | 0.3 | 12.2 | 0.9 | 1.1 | 14.9 | 0.5 | 1.5 | 12.8 | 0.3 | 0.9 |
| Thomas | 2.3 | 0.2 | 3.2 | 12.4 | 0.2 | 2.9 | 12.0 | 0.5 | 3.3 | 13.4 | 0.7 | 3.3 | 12.3 | 0.8 | 2.9 | 12.0 | 0.9 | 4.4 | 13.7 | 0.9 | 1.1 | 13.7 | 0.8 | 4.2 | 12.9 | 0.4 | 0.6 | 13.6 | 0.4 | 0.3 | 11.9 | 0.9 | 4.0 | 13.6 | 0.6 | 4.6 | 12.8 | 0.3 | 4.3 |
| Thomas Labyrinth | 6.5 | 0.1 | 3.2 | 12.5 | 0.2 | 3.2 | 12.1 | 0.3 | 3.3 | 11.8 | 0.3 | 3.3 | 11.5 | 0.6 | 3.3 | 12.4 | 0.8 | 4.1 | 12.0 | 0.9 | 5.2 | 12.6 | 0.9 | 3.6 | 11.7 | 0.3 | 2.8 | 12.5 | 0.4 | 0.4 | 12.5 | 0.8 | 2.5 | 12.5 | 0.3 | 2.5 | 12.6 | 0.3 | 2.6 |
| Torus | 10.9 | 0.9 | 0.9 | 12.4 | 0.7 | 0.0 | 11.9 | 0.2 | 0.0 | 12.0 | 0.4 | 0.0 | 12.0 | 0.6 | 0.1 | 11.9 | 1.0 | 0.5 | 11.9 | 0.9 | 0.7 | 12.6 | 1.0 | 0.6 | 11.5 | 0.4 | 0.4 | 12.1 | 0.2 | 0.2 | 12.6 | 0.9 | 0.9 | 13.1 | 0.6 | 1.3 | 12.5 | 0.3 | 2.4 |
| Tsucs 2 | 7.2 | 0.6 | 1.9 | 10.9 | 0.7 | 1.9 | 10.3 | 0.2 | 0.4 | 7.4 | 0.3 | 2.0 | 8.5 | 0.6 | 2.4 | 8.2 | 0.8 | 2.8 | 7.5 | 0.7 | 1.3 | 7.5 | 0.7 | 3.4 | 7.2 | 0.2 | 1.0 | 6.2 | 0.2 | 1.1 | 8.7 | 0.9 | 3.3 | 9.8 | 0.5 | 4.2 | 12.5 | 0.3 | 3.7 |
| Vallise El Nino | 1.8 | 0.2 | 0.2 | 10.1 | 0.5 | 0.2 | 10.3 | 0.2 | 0.2 | 13.2 | 0.3 | 0.4 | 11.6 | 0.4 | 0.3 | 13.6 | 0.8 | 1.7 | 13.6 | 0.7 | 1.9 | 13.6 | 0.7 | 1.5 | 9.5 | 0.2 | 0.2 | 13.2 | 0.4 | 0.4 | 13.6 | 0.7 | 1.0 | 13.7 | 0.5 | 1.6 | 11.7 | 0.5 | 1.9 |
| Wang Sun | 5.3 | 0.2 | 0.5 | 11.8 | 0.5 | 1.7 | 5.5 | 0.2 | 0.2 | 13.2 | 0.5 | 0.5 | 9.3 | 0.5 | 0.4 | 11.9 | 0.7 | 2.7 | 11.9 | 0.8 | 2.1 | 11.2 | 0.7 | 2.2 | 10.8 | 0.3 | 0.4 | 12.6 | 0.3 | 0.3 | 13.3 | 0.7 | 1.0 | 12.6 | 0.5 | 1.4 | 10.5 | 0.4 | 1.5 |
| Winsmi Reduced | 5.0 | 0.3 | 1.8 | 8.9 | 0.4 | 3.4 | 11.2 | 0.4 | 0.4 | 13.6 | 0.6 | 0.1 | 10.4 | 0.5 | 3.7 | 11.7 | 0.7 | 2.7 | 11.7 | 0.9 | 5.7 | 11.9 | 0.9 | 4.4 | 6.3 | 0.1 | 0.3 | 13.1 | 0.1 | 0.1 | 11.4 | 0.7 | 2.1 | 14.3 | 0.5 | 4.5 | 13.8 | 0.3 | 2.4 |
| Yu Wang | 6.4 | 0.3 | 0.7 | 9.8 | 0.2 | 0.7 | 11.3 | 0.4 | 0.5 | 10.3 | 0.6 | 0.7 | 8.1 | 0.6 | 0.6 | 12.7 | 0.9 | 2.9 | 12.0 | 0.8 | 1.8 | 12.8 | 0.7 | 2.1 | 12.8 | 0.4 | 0.9 | 0.7 | 0.4 | 0.4 | 12.0 | 0.8 | 2.7 | 14.3 | 0.5 | 1.8 | 13.4 | 0.3 | 1.7 |
| Yu Wang 2 | 3.4 | 0.1 | 0.2 | 10.0 | 0.2 | 0.2 | 10.6 | 0.4 | 0.4 | 12.7 | 0.2 | 0.4 | 12.2 | 0.2 | 0.6 | 9.4 | 0.9 | 2.9 | 9.4 | 0.8 | 5.0 | 8.6 | 0.7 | 4.0 | 12.8 | 0.4 | 0.4 | 0.7 | 0.4 | 0.3 | 11.6 | 0.9 | 1.9 | 10.7 | 0.3 | 1.7 | 13.5 | 0.3 | 2.0 |
| Zhou Chen | 10.0 | 0.4 | 0.8 | 10.4 | 0.2 | 0.4 | 13.7 | 0.5 | 2.6 | 12.7 | 0.4 | 1.2 | 12.8 | 0.3 | 3.1 | 11.6 | 0.5 | 4.1 | 11.6 | 0.5 | 4.4 | 13.5 | 0.5 | 4.3 | 13.0 | 0.3 | 2.3 | 11.8 | 0.4 | 0.7 | 12.8 | 0.6 | 5.1 | 11.6 | 0.4 | 7.0 | 14.3 | 0.5 | 5.4 |

