# OpenReview forum: "True Zero-Shot Inference of Dynamical Systems Preserving Long-Term Statistics"
_NeurIPS.cc/2025/Conference — NeurIPS 2025 poster_

### Official Review · Reviewer_H6rs · 2025-06-11

**Clarity:** 4
**Significance:** 4
**Originality:** 4
**Rating:** 5
**Confidence:** 4

**Summary:**

This paper presents DynaMix, a foundation model for dynamical systems based on a mixture of experts with ALRNN models.

The model is pretrained on a large amount of dynamical systems (synthetic 3d data from chaotic systems), and can be used for zero-shot reconstruction of unseen dynamical systems.

DynaMix is tested on both simulated and real-world datasets, and greatly outperforms existing time-series foundation models while also being very computationally efficient.

**Questions:**

See questions in the weaknesses section above.

**Ethical Concerns:**

["NO or VERY MINOR ethics concerns only"]

**Final Justification:**

This is a very solid paper, which is novel, well written and motivated. I believe could have a great impact in the dynamical systems community.

**Limitations:**

Yes

**Quality:**

4

**Strengths And Weaknesses:**

__STRENGTHS__

1. Novelty: The paper introduces a new class of foundation model for time series, specifically designed for dynamical systems reconstruction, which is a direction not previously explored to this extent as far as I know.

2. Performance: DynaMix demonstrates superior long-term forecasting performance compared to state-of-the-art time series foundation models, with orders of magnitude faster inference.

3. Motivation: The model is grounded on the hypothesis that many real-world time series exhibit local dynamics reflective of underlying dynamical systems.

4. Clarity on Limitations: the paper clearly acknowledges that the model is not suited for more complex time series, setting realistic expectations.

5. Thorough Literature Review: The paper provides a comprehensive overview of existing approaches in both time series modeling and dynamical systems.


__WEAKNESSESS__


The paper lacks few analyses/clarifications to better understand the role of different components of the model

1. Why did you choose the ALRNN? How does the model perform with other RNN architectures like LSTM/GRU?
2. How does the model perform without the temporal delay embeddings? Can SOTA models be improved with similar ideas?

Finally, the model is not suited for more complex time series that stationary ones, but this is a limitation that is acknowledged by the authors and I still find the paper valuable as is.

---

> ### Author Rebuttal · Authors · 2025-07-30
>
> We thank the referee for the careful reading and very supportive feedback on our paper, and for raising a couple of great points which we now have addressed with further results (provided in tabular rather than graphical form, as new figures are not allowed this time).
>
> **Weaknesses**
>
> **W1 (further analyses/ other RNNs):** Fair question. We had initially chosen the AL-RNN as a recent SOTA model proposed specifically for DSR, which combines a couple of favorable features from a dynamical systems perspective, like direct translation into symbolic dynamics and providing topologically minimally forms [1] (which we are happy to comment on in our revision). However, we have now also tested other RNNs, like different variants of PLRNNs [2], LSTMs as suggested, vanilla RNNs, and reservoir computers which are also a popular choice in the DSR context [3]. As can be seen in Table R3 below, while AL-RNNs are on par with other PLRNN variants, other types of RNNs (vanilla, LSTM, RC) perform significantly worse, further supporting our specific choice. On the side, we discovered that we can improve DynaMix’ performance even further by including noise into the AL-RNN experts during training (see line “Expert: probabilistic ALRNN”).
>
> More generally, we also performed a series of other ablation studies to clarify the role of different model components, also included in Table R3 below. While the STF-based training itself (see also Fig. 24 in Appx.), the attention mechanism, a sufficient number of experts ($\geq 5$), and a sufficient diversity of the training set with different chaos typologies, are all crucial components, the precise form of the context mapping (whether CNN or linear layer) and the exact expert model *within the class of PLRNNs* has less of an influence on the overall architecture’s success.
>
> Table R3: Ablation experiments \& different expert models: Mean across 5 different training runs of the median performance across the 54 DS test sets.
> | Component  | D_stsp | D_H | MASE |
> | --- | --- | --- | --- |
> | full model | 3.78 ± 0.25 | 0.17 ± 0.01 | 0.36 ± 0.04 |
> | no MLP | 4.02 ± 0.54 | 0.17 ± 0.02 | 0.37 ± 0.04 |
> | no CNN | 10.41 ± 0.60 | 0.47 ± 0.02 | 3.34 ± 0.24 |
> | no Attention | 11.63 ± 1.18 | 0.52 ± 0.04 | 3.37 ± 0.22 |
> | CNN to linear layer | 5.65 ± 0.22 | 0.22 ± 0.01 | 0.57 ± 0.06 |
> | no STF | 13.66 ± 0.14 | 0.49 ± 0.03 | 3.02 ± 0.41 |
> | #experts<5| 13.82 ± 0.14 | 0.49 ± 0.06 | 3.50 ± 0.09 |
> | Restricted training set: Lorenz-like DS only | 11.82 ± 0.75 | 0.55 ± 0.02 | 3.34 ± 0.25 |
> | Expert: clipped shallow PLRNN | 3.78 ± 0.26 | 0.16 ± 0.01 | 0.34 ± 0.03 |
> | Expert: probabilistic ALRNN | 3.07 ± 0.10 | 0.16 ± 0.01 | 0.40 ± 0.05 |
> | Expert: vanilla RNN | 5.72 ± 0.42 | 0.22 ± 0.01 | 0.61 ± 0.03 |
> | Expert: LSTM | 8.03 ± 4.31 | 0.32 ± 0.16 | 1.61 ± 1.64 |
> | Expert: Reservoir Computer | 13.93 ± 0.10 | 0.54 ± 0.01 | 3.86 ± 0.18 |
>
> **W2 (delay embedding):** Some form of embedding is indeed necessary to achieve strong DSR results (cf. sect. 3.3), but this does not necessarily need to be a delay embedding, but could also be a positional encoding (eq. 6) as similarly employed in Transformer-based models like the different versions of Chronos. Indeed, for all empirical examples in Fig. 8 we had used positional encoding for DynaMix, i.e. DynaMix outperformed the other FMs in terms of DSR (and often forecasting) even though all models received the same types of inputs (scalar time series and positional embedding). However, we have also tested TTM with delay embedding now (the only one of the FMs which actually allows for this type of multivariate embedding), hardly improving the results as provided below.
>
> Table R6: Reconstruction results for TTM with delay embedding
> | Model | D_stsp | D_H | MASE |
> |-------|--------|-----|-----|
> | original TTM | 13.30 ± 0.68 | 0.32 ± 0.07 | 3.04 ± 1.25 |
> | TTM with delay embedding | 13.03 ± 0.61 | 0.29 ± 0.06 | 3.94 ± 1.77 |
>
> **W3 (non-stationary TS):** Thank you. Non-stationary time series are indeed still an open issue, with which other models struggle as well. As a proof of concept, however, we have now designed a pre-processing pipeline, which (similar as in FEDformers [4]) first estimates trend components from the context signal and adds them back on for inference. We have tested this idea on the non-stationary Air Passengers TS benchmark (cf. [5]), and compare performance of DynaMix with this enhancement to the best competitor (Chronos-t5-base; see Fig. 4c \& Table 1) in Table R7 below. While we are not allowed to provide figures in the rebuttal which would illustrate the approach and results more clearly, we will include these in our revision. This is just a proof-of-concept at this stage, but it illustrates the approach in principle works and could be extended and potentially optimized.
>
> Table R7: Forecasting results for *non-stationary* Air Passenger TS benchmark
> | Model | D_stsp | D_H | MASE |
> |-------|--------|-----|-----|
> | DynaMix | 2.25 | 0.04 | 0.54 |
> | Chronos-t5-base | 3.49 | 0.06 | 1.16 |
>
> **References**
>
> [1] Brenner, M., et al. (2024). Almost-linear RNNs yield highly interpretable symbolic codes in dynamical systems reconstruction. Advances in Neural Information Processing Systems, 37, 36829–36868.
>
> [2] Hess, F., et al. (2023). Generalized Teacher Forcing for Learning Chaotic Dynamics. Proceedings of the 40th International Conference on Machine Learning, 13017–13049. PMLR.
>
> [3] Patel, D., & Ott, E. (2023). Using machine learning to anticipate tipping points and extrapolate to post-tipping dynamics of non-stationary dynamical systems. Chaos: An Interdisciplinary Journal of Nonlinear Science, 33(2).
>
> [4] Zhou et al. (2022) FEDformer: Frequency Enhanced Decomposed Transformer for Long-term Series Forecasting. Proceedings of the 39th International Conference on Machine Learning, PMLR 162:27268-27286
>
> [5] Ansari, A. F., et al. (2024). Chronos: Learning the language of time series. arXiv preprint arXiv:2403.07815.

---

> > ### Comment · Reviewer_H6rs · 2025-08-03
> >
> > Thank you for the replies and additional experiments that further confirm the benefits of Dynamix. I will confirm my accept score.

---

> > > ### Author Response · Authors · 2025-08-03
> > >
> > > We, once again, thank the referee very much for all the support and appreciation of our work!

---

### Official Review · Reviewer_gq5P · 2025-06-17

**Clarity:** 4
**Significance:** 3
**Originality:** 3
**Rating:** 5
**Confidence:** 3

**Summary:**

This paper proposes DynaMix, a foundation model for dynamical systems. The model is a mixture-of-experts based on an RNN and trained via sparse teacher forcing. The authors perform an extensive empirical evaluation of their method across a range of dynamical systems and time series forecasting tasks. DynaMix exhibits strong predictive performance while simultaneously requiring less inference time than existing well-known time series foundation models.

**Questions:**

- From my understanding, the context dimension $N$ is fixed and shared across tasks during model training. L153-165 describe techniques for applying the model to signals of smaller dimension. Is it possible to apply the model to signals of higher dimension, or is this a limitation of the model?
- Is the model applicable in settings where your signal is not observed uniformly in time?

**Ethical Concerns:**

["NO or VERY MINOR ethics concerns only"]

**Final Justification:**

I think this is a solid, high-quality paper. The paper itself is very clear and insightful, and the proposed methodology seems highly performant. The authors clearly demonstrate the reasons for the failures of existing methods which provides a good deal of insight.

The reason I am not voting for a higher score of (6) is that, while this is a strong paper, I don't think that it quite constitutes "groundbreaking impact", as the methodology is largely within the existing paradigm.

**Limitations:**

yes

**Quality:**

4

**Strengths And Weaknesses:**

# Strengths

- The proposed architecture is clearly motivated and seems well-suited for the task at hand
- The empirical evaluation in the paper is thorough and convincing. The proposed model architecture exhibits clear gains compared against well-known time series foundation models in terms of zero-shot DSR (Fig 4) and time series forecasting (Tab 1) while being significantly faster (Fig 5).
- I especially appreciate that the authors clearly demonstrate various short-comings of existing methods (Section 4.2) rather than only reporting metrics. This provides significant insight.
- Section 4.3, on MoE similarity, is an interesting qualitative evaluation of the model.
- The paper is generally very clearly written and well-presented throughout

# Weaknesses

- The paper could potentially be strengthened by additional ablations. It wasn't clear to me which components of the model (the AL-RNN, the MoE, the specifics of the training setup, etc.) were most important
- It also would have been interesting to see how the proposed method compares to DSR methods which are not based on foundation models -- while these certainly have drawbacks (e.g., requiring task-specific training or fine-tuning), it is unclear if this would result in better performance.

# Minor Comments
- I found Section 3.2 a bit dense; while things are explained in the appendix, it would be ideal to provide more details in the main paper given that this seems to be an integral part of the method
- The curves in Figure 8 are difficult to see/distinguish from one another

---

> ### Author Rebuttal · Authors · 2025-07-30
>
> We thank the referee for their thorough assessment of our work, and the very supportive and encouraging feedback!
>
> **Weaknesses**
>
> **W1 (ablations):** Thank you for bringing this up, we have now performed a series of ablation experiments, summarized in Table R3 below. While the STF-based training itself (see also Fig. 24 in Appx.), the attention mechanism, a sufficient number of experts ($\geq 5$), and a sufficient diversity of the training set with different chaos typologies, are all crucial components, the precise form of the context mapping (whether CNN or linear layer) and the exact expert model *within the class of PLRNNs* has less of an influence on the overall architecture’s success. Using as experts LSTMs, vanilla RNNs or reservoir computers (RCs) instead of any of the PLRNN-variants, however, led to decay in performance. On the side, we discovered that we can improve DynaMix’ performance even further by including noise into the AL-RNN experts during training (see line “Expert: probabilistic ALRNN”).
>
> Table R3: Ablation experiments \& different expert models: Mean across 5 different training runs of the median performance across the 54 DS test sets.
> | Component  | D_stsp | D_H | MASE |
> | --- | --- | --- | --- |
> | full model | 3.78 ± 0.25 | 0.17 ± 0.01 | 0.36 ± 0.04 |
> | no MLP | 4.02 ± 0.54 | 0.17 ± 0.02 | 0.37 ± 0.04 |
> | no CNN | 10.41 ± 0.60 | 0.47 ± 0.02 | 3.34 ± 0.24 |
> | no Attention | 11.63 ± 1.18 | 0.52 ± 0.04 | 3.37 ± 0.22 |
> | CNN to linear layer | 5.65 ± 0.22 | 0.22 ± 0.01 | 0.57 ± 0.06 |
> | no STF | 13.66 ± 0.14 | 0.49 ± 0.03 | 3.02 ± 0.41 |
> | #experts<5| 13.82 ± 0.14 | 0.49 ± 0.06 | 3.50 ± 0.09 |
> | Restricted training set: Lorenz-like DS only | 11.82 ± 0.75 | 0.55 ± 0.02 | 3.34 ± 0.25 |
> | Expert: clipped shallow PLRNN | 3.78 ± 0.26 | 0.16 ± 0.01 | 0.34 ± 0.03 |
> | Expert: probabilistic ALRNN | 3.07 ± 0.10 | 0.16 ± 0.01 | 0.40 ± 0.05 |
> | Expert: vanilla RNN | 5.72 ± 0.42 | 0.22 ± 0.01 | 0.61 ± 0.03 |
> | Expert: LSTM | 8.03 ± 4.31 | 0.32 ± 0.16 | 1.61 ± 1.64 |
> | Expert: Reservoir Computer | 13.93 ± 0.10 | 0.54 ± 0.01 | 3.86 ± 0.18 |
>
> **W2 (comparison to non-FMs):** Very good point. To put our model’s zero-shot performance into perspective, we have now trained various DSR models (AL-RNN trained by STF [1], reservoir computers for DSR [2], and neural ODEs for DSR [3,4]) on the same data used in the paper, with results shown below in Table R2. As evident from this table, DynaMix’ zero-shot *generalization* performance is completely on-par with that of SOTA DSR models specifically trained on these data, even *without any fine-tuning or retraining* (performance across the 54 DS test sets was statistically largely indistinguishable according to t-tests, while performance on the empirical TS was on average even slightly better).
>
> Table R2: Performance for *custom trained* DSR (ALRNN, RC, Neural ODE) models across all 54 test set DS (as in Fig.4; median+/-MAD) and for the empirical time series (as in Fig. 8). Best in bold.
> | | DynaMix | | | ALRNN | | | RC | | | NODE | | |
> |---|---|---|---|---|---|---|---|---|---|---|---|---|
> | Dataset | D_stsp | D_H | MASE | D_stsp | D_H | MASE | D_stsp | D_H | MASE | D_stsp | D_H | MASE |
> | DS testsets | 3.92 ± 1.47 | 0.19 ± 0.09 | 0.36 ± 0.17 | 3.65 ± 2.54 | **0.08 ± 0.06** | 0.27 ± 0.16 | 5.65 ± 4.77 | 0.23 ± 0.19 | 0.96 ± 0.50 | **3.51 ± 2.32** | 0.12 ± 0.07 | **0.23 ± 0.13** |
> | ETTh1 | 0.8 | **0.08** | 2.13 | **0.33** | 0.13 | **1.63** | 1.01 | 0.27 | 2.99 | 12.76 | 0.31 | 2.81 |
> | Traffic | 0.68 | 0.21 | **1.58** | **0.33** | **0.16** | 2.43 | 1.5 | 0.43 | 2.96 | NaN | 0.5 | 50.65 |
> | Cloud Requests | **0.17** | 0.14 | **1.05** | 0.52 | **0.13** | 4.62 | 3.99 | 0.31 | 2.4 | 11.98 | 0.3 | 6.51 |
> | Weather (Temp.) | **0.59** | **0.09** | **2.74** | 0.91 | 0.16 | 3.63 | 11.26 | 0.66 | 8.8 | 12.4 | 0.51 | 8.82 |
> | Weather (Press.) | 0.41 | 0.19 | 5.25 | **0.23** | 0.28 | 5.6 | 0.26 | 0.29 | 5.95 | 11.42 | **0.16** | **4.9** |
> | Human fMRI | 0.63 | 0.09 | **1.91** | 0.61 | **0.08** | 2.47 | **0.15** | 0.17 | 3.22 | 7.81 | 0.35 | 2.77 |
> | Human EEG | **0.78** | 0.23 | 8.2 | 2.33 | 0.54 | **3.29** | 6.55 | 0.2 | 4.8 | 6.26 | **0.16** | 5.04 |
> | Avg. real-world TS | **0.58 ± 0.22** | **0.14 ± 0.06** | **3.26 ± 2.57** | 0.75 ± 0.73 | 0.21 ± 0.15 | 3.38 ± 1.37 | 3.53 ± 4.11 | 0.33 ± 0.16 | 4.45 ± 2.29 | 10.44 ± 2.72 | 0.33 ± 0.14 | 11.64 ± 17.32 |
>
> **Minor**
>
> **M1)** Agreed, this sect. is rather short, and we will move more details from Appx. A.2 back to the main text in the revision.
>
> **M2)** Yes, fair point. One solution here is to only show a subset of the empirical data in the main text (moving the others to the Appx.), and using the space instead to show predictions from each foundation model in separate panels (or, alternatively, to focus only on the comparison between DynaMix and its strongest competitor in each panel).
>
> **Questions**
>
> **Q1)** Yes, this is possible. First, please note that the empirical examples (incl. human EEG or the air pressure data) are inherently likely of much higher dimensionality, yet DynaMix is still able to reconstruct and forecast these systems well. Second, we have now included another, 6d DSR benchmark (Lorenz-96), see results in Table R5 below.
>
> Table R5: Reconstruction & forecast results on 6d Lorenz-96 across all foundation models
> | Model | D_stsp | D_H | MASE |
> |-------|--------|-----|-----|
> | DynaMix | 3.67 | 0.10 | 0.98 |
> | Chronos-t5-base | 12.08 | 0.34 | 1.59 |
> | Chronos-bolt-base | 13.55 | 0.69 | 2.07 |
> | Chronos-bolt-small | 13.93 | 0.63 | 2.48 |
> | Chronos-t5-tiny | 14.52 | 0.38 | 2.62 |
> | Chronos-t5-mini | 14.51 | 0.42 | 2.67 |
> | Chronos-bolt-mini | 14.07 | 0.61 | 2.70 |
> | Chronos-bolt-tiny | 13.11 | 0.62 | 2.71 |
> | Chronos-t5-small | 12.12 | 0.38 | 2.91 |
> | TTM | 14.65 | 0.37 | 3.52 |
> | Mamba4Cast | 13.59 | 0.70 | 3.55 |
>
> **Q2)** As it currently stands, when observations arrive at non-equal temporal intervals, this would need proper preprocessing (like interpolation) for *all* the models reported in the paper. So this is currently a limitation that we will add to our Limitations sect..
>
> **References**
>
> [1] Brenner, M., et al. (2024). Almost-linear RNNs yield highly interpretable symbolic codes in dynamical systems reconstruction. Advances in Neural Information Processing Systems, 37, 36829–36868.
>
> [2] Patel, D., & Ott, E. (2023). Using machine learning to anticipate tipping points and extrapolate to post-tipping dynamics of non-stationary dynamical systems. Chaos: An Interdisciplinary Journal of Nonlinear Science, 33(2).
>
> [3] Chen, R. T., et al. (2018). Neural ordinary differential equations. Advances in Neural Information Processing Systems, 31.
>
> [4] Hess, F., et al. (2023). Generalized Teacher Forcing for Learning Chaotic Dynamics. Proceedings of the 40th International Conference on Machine Learning, 13017–13049. PMLR.

---

> > ### Comment · Reviewer_gq5P · 2025-08-01
> >
> > Many thanks for these clarifications and additional experiments. The additional experiments for ablations and non-foundation models are appreciated and could serve to further strengthen the paper.
> >
> > I think this is a solid paper and I'd like to maintain my initial positive assessment.

---

> > > ### Author Response · Authors · 2025-08-01
> > >
> > > We once again thank the referee for their suggestions which prompted us to examine several aspects of our architecture in more detail, and for the encouraging support of our work!

---

### Official Review · Reviewer_qUES · 2025-07-01

**Clarity:** 3
**Significance:** 3
**Originality:** 3
**Rating:** 5
**Confidence:** 4

**Summary:**

The paper proposes DynaMix, a lightweight AL-RNN–based mixture-of-experts architecture. The proposed DynaMix architecture leverages a mixture-of-experts (MoE) framework employing almost-linear RNNs (AL-RNNs) as experts. In this framework, the next hidden state is predicted as a weighted sum of the outputs from multiple experts, with the weights determined by a gating network which computes these weights using the context and the current hidden state as inputs. By using lightweight experts and gating network, DynaMix is computationally efficient compared to recent transformer-based time series (TS) foundation models. Experimental results demonstrate that the proposed method performs well on dynamical system reconstruction (DSR) tasks and even on short-term forecasting without any fine-tuning.

**Questions:**

Q1. Could the authors provide quantitative comparisons with the state-of-the-art DSR models in term of both DSR task and time series forecasting?

Q2. Could the authors conduct ablation studies to clarify which specific architectural choices, beyond the general MoE framework, are responsible for the observed performance gains?

Q3. How does the model scale and generalize as the number of experts increases, and are there strategies in place to control model size or improve robustness to highly novel dynamical systems?

Q4. The rationale for selecting AL-RNNs as experts within the MoE framework is not sufficiently justified. Could you provide a more detailed explanation to explain why this choice is effective or necessary?

Q5. Can you compare the test performances of the proposed scheme, benchmark SOTA DSR models and time series foundation models, for data set with more complicated dynamics (e.g., 6D) in terms of term of both DSR task and time series forecasting?

**Ethical Concerns:**

["NO or VERY MINOR ethics concerns only"]

**Final Justification:**

The authors have satisfactorily addressed most of the concerns and questions raised in the previous review. Therefore, I recommend accepting this paper.

**Limitations:**

yes

**Paper Formatting Concerns:**

No formatting concerns

**Quality:**

3

**Strengths And Weaknesses:**

Strengths:

S1. DynaMix can reconstruct and forecast previously unseen dynamical systems from only a short context without need for retraining or fine-tuning

S2. The model achieves high performance with a lightweight architecture—requiring orders of magnitude fewer parameters and delivering much faster inference compared to existing foundation models, which makes it suitable for practical deployment at scale

S3. The effectiveness of the proposed method is validated through various experiments on both DSR datasets and general time series datasets.

Weaknesses:

W1. The paper lacks quantitative comparisons with the state-of-the-art DSR models in term of both DSR tasks and time series forecasting, which limits the fairness and practical context of its benchmarking.

W2..The Mixture-of-Experts framework is not itself a novel concept, and given that strong performance is reported in a broad sense, it remains unclear to what extent the authors’ specific architecture choices contribute to the observed gains, making the source of novelty somewhat ambiguous

W3. The scalability and generalization of the model remain uncertain, as increasing the number of experts to cover more diverse or fundamentally different dynamical systems could lead to substantial growth in model size and potential degradation in out-of-distribution performance

---

> ### Author Rebuttal · Authors · 2025-07-30
>
> We thank the referee for taking the time to thoroughly scrutinize our work, and for pointing out various omissions in terms of model evaluation, which we have now all addressed and provided in the form of result tables below (as figures are not allowed for the rebuttal this time).
>
> **Weaknesses**
>
> **W1 (further comparisons):** Good point. Our comparison group here were other foundation models (to which we now have further added TimesFM [5]: Dstsp: 12.50 ± 0.93, DH: 0.36 ± 0.11, MASE: 1.11 ± 0.63), as ‘standard’ DSR models cannot perform zero-shot reconstruction and forecasting but need training on at least segments of the new data. However, we fully agree that such a comparison would provide practical context, and have now compared DynaMix’ zero-shot performance with that of three SOTA DSR models explicitly trained on the data, namely the AL-RNN trained with STF as in [1], reservoir computers (RC) for DSR [2], and neural ODEs (NODE) for DSR as in [3,4]. As shown in Table R2 below, the *zero-shot generalization* performance of DynaMix in terms of both long-term statistics and forecasting is completely within the ballpark of custom-trained SOTA DSR models, *even without any fine-tuning or retraining* (statistically largely indistinguishable across the 54 DS test set according to t-tests, and on average even slightly better on the empirical TS).
>
> Table R2: Performance for *custom trained* DSR models (ALRNN, RC, Neural ODE) across all 54 test set DS (as in Fig.4; median+/-MAD) and for the empirical time series (as in Fig. 8). Best in bold.
> | | DynaMix | | | ALRNN | | | RC | | | NODE | | |
> |---|---|---|---|---|---|---|---|---|---|---|---|---|
> | Dataset | D_stsp | D_H | MASE | D_stsp | D_H | MASE | D_stsp | D_H | MASE | D_stsp | D_H | MASE |
> | DS testsets | 3.92 ± 1.47 | 0.19 ± 0.09 | 0.36 ± 0.17 | 3.65 ± 2.54 | **0.08 ± 0.06** | 0.27 ± 0.16 | 5.65 ± 4.77 | 0.23 ± 0.19 | 0.96 ± 0.50 | **3.51 ± 2.32** | 0.12 ± 0.07 | **0.23 ± 0.13** |
> | ETTh1 | 0.8 | **0.08** | 2.13 | **0.33** | 0.13 | **1.63** | 1.01 | 0.27 | 2.99 | 12.76 | 0.31 | 2.81 |
> | Traffic | 0.68 | 0.21 | **1.58** | **0.33** | **0.16** | 2.43 | 1.5 | 0.43 | 2.96 | NaN | 0.5 | 50.65 |
> | Cloud Requests | **0.17** | 0.14 | **1.05** | 0.52 | **0.13** | 4.62 | 3.99 | 0.31 | 2.4 | 11.98 | 0.3 | 6.51 |
> | Weather (Temp.) | **0.59** | **0.09** | **2.74** | 0.91 | 0.16 | 3.63 | 11.26 | 0.66 | 8.8 | 12.4 | 0.51 | 8.82 |
> | Weather (Press.) | 0.41 | 0.19 | 5.25 | **0.23** | 0.28 | 5.6 | 0.26 | 0.29 | 5.95 | 11.42 | **0.16** | **4.9** |
> | Human fMRI | 0.63 | 0.09 | **1.91** | 0.61 | **0.08** | 2.47 | **0.15** | 0.17 | 3.22 | 7.81 | 0.35 | 2.77 |
> | Human EEG | **0.78** | 0.23 | 8.2 | 2.33 | 0.54 | **3.29** | 6.55 | 0.2 | 4.8 | 6.26 | **0.16** | 5.04 |
> | Avg. real-world TS | **0.58 ± 0.22** | **0.14 ± 0.06** | **3.26 ± 2.57** | 0.75 ± 0.73 | 0.21 ± 0.15 | 3.38 ± 1.37 | 3.53 ± 4.11 | 0.33 ± 0.16 | 4.45 ± 2.29 | 10.44 ± 2.72 | 0.33 ± 0.14 | 11.64 ± 17.32 |
>
> **W2 (ablation studies):** Yes, the MoE concept itself is not novel of course, but what is very novel in our minds is the whole model design and training for DSR, making it to our knowledge the *first* foundation model with zero-shot DSR capabilities, incl. long-term properties.
> We, however, fully agree with the referee that some more insight into which model components are crucial for its performance is useful, and have now performed a series of ablation experiments as summarized in Table R3 below. While the STF-based training itself (see also Fig. 24 in Appx.), the attention mechanism, a sufficient number of experts, and a sufficiently diverse training set with different chaos typologies, are all crucial components, the precise form of the context mapping (whether CNN or linear layer) and the exact expert model *within the class of PLRNNs* [4] has less of an influence on the overall architecture’s success (using as experts LSTMs, vanilla RNNs or reservoir computers instead of any of the PLRNN-variants, however, led to decay in performance). Hence, it is really the overall design and training procedure which make the framework so strong. On the side, we discovered that we can improve DynaMix’ performance even further by including noise into the ALRNN experts during training (see line “Expert: probabilistic ALRNN”).
>
> Table R3: Ablation experiments \& different expert models: Mean across 5 different training runs of the median performance across the 54 test sets.
> | Component  | D_stsp | D_H | MASE |
> | --- | --- | --- | --- |
> | full model | 3.78 ± 0.25 | 0.17 ± 0.01 | 0.36 ± 0.04 |
> | no MLP | 4.02 ± 0.54 | 0.17 ± 0.02 | 0.37 ± 0.04 |
> | no CNN | 10.41 ± 0.60 | 0.47 ± 0.02 | 3.34 ± 0.24 |
> | no Attention | 11.63 ± 1.18 | 0.52 ± 0.04 | 3.37 ± 0.22 |
> | CNN to linear layer | 5.65 ± 0.22 | 0.22 ± 0.01 | 0.57 ± 0.06 |
> | no STF | 13.66 ± 0.14 | 0.49 ± 0.03 | 3.02 ± 0.41 |
> | #experts<5| 13.82 ± 0.14 | 0.49 ± 0.06 | 3.50 ± 0.09 |
> | Restricted training set: Lorenz-like DS only | 11.82 ± 0.75 | 0.55 ± 0.02 | 3.34 ± 0.25 |
> | Expert: clipped shallow PLRNN | 3.78 ± 0.26 | 0.16 ± 0.01 | 0.34 ± 0.03 |
> | Expert: probabilistic ALRNN | 3.07 ± 0.10 | 0.16 ± 0.01 | 0.40 ± 0.05 |
> | Expert: vanilla RNN | 5.72 ± 0.42 | 0.22 ± 0.01 | 0.61 ± 0.03 |
> | Expert: LSTM | 8.03 ± 4.31 | 0.32 ± 0.16 | 1.61 ± 1.64 |
> | Expert: Reservoir Computer | 13.93 ± 0.10 | 0.54 ± 0.01 | 3.86 ± 0.18 |
>
> **W3 (scalability):** We now explicitly tested model scaling with the number of AL-RNN experts. As can be appreciated from Table R4 below, a minimum number of experts ($\geq5$) is necessary to achieve optimal performance, while after that performance rather plateaus. At the same time, the computational load (training times) scales only *linearly* with the number of experts $J$ (tested for $J=10:10:80, R^2=0.97$; note we are not allowed to provide figures this time, from which this may have been clearer; regarding inference times, these are all in the split-second range anyway).
>
> Regarding diversity, we would like to point out that in sect. 4.4 we did apply DynaMix to a very diverse range of empirical settings with dynamics fundamentally different from the training set. This, in our minds, demonstrates its generalization capabilities with regards to fundamentally different dynamical systems.
>
> Table R4: Dependence of model performance on \# of experts ($J$): Mean across 5 different training runs of the median performance across the 54 DS test sets.
> | J | D_stsp | D_H | MASE |
> | --- | --- | --- | --- |
> | 1 | 13.82 ± 0.14 | 0.49 ± 0.06 | 3.50 ± 0.09 |
> | 3 | 12.81 ± 0.18 | 0.50 ± 0.02 | 2.19 ± 0.17 |
> | 5 | 3.91 ± 0.48 | 0.16 ± 0.02 | 0.48 ± 0.07 |
> | 10 | 4.17 ± 0.36 | 0.19 ± 0.01 | 0.38 ± 0.03 |
> | 20 | 4.86 ± 0.70 | 0.22 ± 0.02 | 0.44 ± 0.06 |
> | 30 | 5.09 ± 1.26 | 0.23 ± 0.06 | 0.48 ± 0.10 |
>
> **Questions**
>
> **Q1)** We have now done so, see response to W1 and Table R2 above.
>
> **Q2)** Yes, now done, please see our response to W2 and Table R3 above.
>
> **Q3)** We think that the applications of our model to the diverse set of empirical data in Fig. 8 and Table 1 of the paper already constitute test cases with highly novel and diverse dynamics, since no such data were part of the training set! We now further included a high-d hyper-chaotic example, see response to Q5 below. Finally, as our response to W3 and Table R4 above indicates, our system is very well behaved and highly robust with regards to scaling up model size.
>
> **Q4)** Yes, fair question. The original motivation was that the AL-RNN is a recent SOTA model proposed specifically for DSR, which combines a couple of favorable features from a dynamical systems perspective, like direct translation into symbolic dynamics and providing topologically minimally forms [1] (on which we will comment in our revision). As furthermore shown in Table R3 (see reply to W2 above), however, AL-RNNs also perform substantially better in this context than other typical RNN choices like LSTMs or reservoir computers often used for DSR.
>
> **Q5)** In our minds the empirical test case examples (sect. 4.4) already cover systems with very complicated and much higher dimensional underlying dynamics, human EEG for instance, or the air pressure and cloud request data. We now have further included a 6d Lorenz-96 system in its hyper-chaotic regime and compared the DSR and forecasting performance of all models in Table R5 below, verifying that the strong performance of DynaMix also holds for complex hyper-chaotic systems.
>
> Table R5: Reconstruction & forecast results on 6d Lorenz-96 across all foundation models
> | Model | D_stsp | D_H | MASE |
> |-------|--------|-----|-----|
> | DynaMix | 3.67 | 0.10 | 0.98 |
> | Chronos-t5-base | 12.08 | 0.34 | 1.59 |
> | Chronos-bolt-base | 13.55 | 0.69 | 2.07 |
> | Chronos-bolt-small | 13.93 | 0.63 | 2.48 |
> | Chronos-t5-tiny | 14.52 | 0.38 | 2.62 |
> | Chronos-t5-mini | 14.51 | 0.42 | 2.67 |
> | Chronos-bolt-mini | 14.07 | 0.61 | 2.70 |
> | Chronos-bolt-tiny | 13.11 | 0.62 | 2.71 |
> | Chronos-t5-small | 12.12 | 0.38 | 2.91 |
> | TTM | 14.65 | 0.37 | 3.52 |
> | Mamba4Cast | 13.59 | 0.70 | 3.55 |
>
> **References**
>
> [1] Brenner, M., et al. (2024). Almost-linear RNNs yield highly interpretable symbolic codes in dynamical systems reconstruction. Advances in Neural Information Processing Systems, 37, 36829–36868.
>
> [2] Patel, D., & Ott, E. (2023). Using machine learning to anticipate tipping points and extrapolate to post-tipping dynamics of non-stationary dynamical systems. Chaos: An Interdisciplinary Journal of Nonlinear Science, 33(2).
>
> [3] Chen, R. T., et al. (2018). Neural ordinary differential equations. Advances in Neural Information Processing Systems, 31.
>
> [4] Hess, F., et al. (2023). Generalized Teacher Forcing for Learning Chaotic Dynamics. Proceedings of the 40th International Conference on Machine Learning, 13017–13049. PMLR.
>
> [5] Das, A., et al. (2024, July). A decoder-only foundation model for time-series forecasting. In Proceedings of the 41st International Conference on Machine Learning.

---

> > ### Comment · Reviewer_qUES · 2025-08-02
> > **Following-up questions**
> >
> > Thank you very much for the thorough clarifications and the additional comparisons. Based on the results provided in the rebuttal, the reviewer has the following follow-up questions for further clarification:
> >
> > 1. As shown in Table R3, when the clipped shallow PLRNN [4] was used as the expert in the MoE ablation study, all performance indicators were comparable to those of the full model. Could the authors elaborate on the advantages of using ALRNN as an expert, as opposed to the clipped shallow PLRNN?
> >
> > 2. The results also suggest that the probabilistic ALRNN, trained with added noise, achieves strong performance. Have the authors considered applying a similar noise-augmentation strategy during training to the clipped shallow PLRNN expert? This could offer a meaningful point of comparison.
> >
> > 3. In Table R4, performance appears to deteriorate when the number of experts exceeds 10. It would be helpful if the authors could clarify the underlying reasons for this degradation. To support claims of model scalability, it would be important to determine whether this drop is due to overfitting, optimization challenges, or training inefficiencies associated with having a large number of experts.
> >
> > 4. Regarding Table R5, could the authors clarify whether the reported results reflect a model originally trained on the 3D dataset, or one that underwent additional training on the 6D dataset? As the purpose of this experiment is to test generalization from simpler to more complex settings, a clear explanation would be helpful to interpret the results properly.
> >
> > 5 Finally, in response to Q5, the reviewer requested a comparative evaluation of state-of-the-art DSR methods on the 6D Lorenz-96 dataset. This was not addressed in the rebuttal. The reviewer think including such a comparison is essential for positioning the proposed method relative to existing approaches. Could the authors please elaborate on this regard?

---

> > > ### Author Response · Authors · 2025-08-03
> > >
> > > We thank the referee for giving us the chance to provide further clarification on our new results.
> > >
> > > **1)** The ALRNN is simpler (more parsimonious) in its structure than the clipped shallow PLRNN (e.g., uses as few nonlinearities as necessary), and therefore also faster to train and easier to interpret in its behavior (see [1]). However, otherwise we are quite indifferent on whether practitioners would use the ALRNN or some other version of PLRNN (they both belong to the same general class of PL models).
> > >
> > > **2)** Interesting question. We tried this now and found that the cshPLRNN could be similarly improved by adding noise ($D_{stsp}=3.17±0.17, D_H=0.16±0.01, MASE=0.35±0.03$). Please also see our response above.
> > >
> > > **3)** Statistically this apparent decline is not significant (p>0.15, t-test), so one should probably be cautious not to over-interpret this observation. We could imagine that while training time per epoch increases only linearly with the number of experts J, in general other hyper-parameter settings may be required, but since the difference is modest and statistically insignificant, and the performance still good with J=30, we would not want to draw any strong conclusions from this.
> > >
> > > More importantly, given the strong generalization performance of DynaMix on challenging real-world data that we had demonstrated in sect. 4.4, we are also not sure whether DynaMix really needs to be scaled up that much further. Using J=20 as we had done in the paper seems to work just fine, and in any case we take the results in Table R4 more as evidence that one may be able to achieve superb performance *with even fewer experts*.
> > >
> > > **4)** No, for these comparisons DynaMix did not undergo additional training on 6d DS, but we used *exactly the same 3d training set* as before and described in the paper. Hence this is true out-of-domain generalization to a 6d hyperchaotic dynamical system not experienced in any form in training (3d DS do not even exhibit hyper-chaos! [2]).
> > >
> > > **5)** We have now compared DynaMix to custom-trained DSR models on a total of 63 (!!) simulated and real-world dynamical systems (Table R2), from *none* of which there were *any* data in DynaMix’ training set. In particular the 8 diverse *real-world* datasets *do* in fact test generalization from simpler to way more complicated settings. EEG, fMRI, or climate data, for instance, are all well known to be highly challenging and even many SOTA DSR models fail on them, see e.g. [3]. DynaMix actually performs *best* on these data, which at least to us is quite surprising, as there were *no* real-world data in training at all. Yet, at the same time, the SOTA DSR models are *custom-trained on exactly these data*, hence do *not* achieve the type of generalization that DynaMix comes with.
> > >
> > > This type of out-of-domain generalization is precisely the strength of foundation models, and it is the reason why we focused in our paper on comparison with other major foundation models. Thus, while we certainly agree with the referee that the results in Table R2 are useful as a reference, we think one needs to bear in mind that none of the custom-trained models, unlike DynaMix, performs out-of-domain generalization here (such that even on-par performance is actually surprising)!
> > >
> > > In any case, we have now run the additional comparisons on the 6d Lorenz-96 as requested, presented in Table RR1 below, confirming once again that DynaMix can well keep up with custom-trained DSR models with comparable performance.
> > >
> > > Table RR1: Comparison on 6d Lorenz-96
> > > | Model | D_stsp | D_H | MASE |
> > > |-------|--------|-----|-----|
> > > | DynaMix | 3.67 | 0.10 | 0.98 |
> > > | ALRNN | 3.41 | 0.22 | 0.82 |
> > > | RC | 3.63 | 0.27 | 2.52 |
> > > | NODE | 3.09 | 0.20 | 0.92 |
> > >
> > > [1] Brenner, M., et al. (2024). Almost-linear RNNs yield highly interpretable symbolic codes in dynamical systems reconstruction. Advances in Neural Information Processing Systems, 37, 36829–36868.
> > >
> > > [2] Letellier & Roessler (2007), Hyperchaos. Scholarpedia, 2(8):1936.
> > >
> > > [3] Hess, F., et al. (2023). Generalized Teacher Forcing for Learning Chaotic Dynamics. Proceedings of the 40th International Conference on Machine Learning, 13017–13049. PMLR.

---

> ### Comment · Reviewer_qUES · 2025-08-03
>
> Thank you so much for further clarification and experiments!  Based on your responses, I have updated my recommendation to accept.

---

> > ### Author Response · Authors · 2025-08-03
> >
> > We thank the referee very much for their great engagement, and for the many suggestions that helped to further back up and strengthen our results!

---

### Official Review · Reviewer_beYK · 2025-07-08

**Clarity:** 4
**Significance:** 3
**Originality:** 4
**Rating:** 6
**Confidence:** 4

**Summary:**

The paper introduces DynaMix a new type of zero-shot forecasting / dynamical system reconstruction approach.
DynaMix uses a mixture of expert approach, where each expert is a multi-dimensional almost linear RNN model.
Given a contex window of a time series signal, an attention mechanism is used to gate the experts. The model is trained across a wide set of dynamical systems to reproduce dynamical system behavior.

The paper demonstrate in simulations that the model can forecast and re-construct the statistical properties of many dynamical systems and in particular for chaotic dynamics -- a setting where other foundational time series forecasting models fail. The paper also shows that DynaMix can give reasonable forecast accuracy on real world time series forecasting benchmarks, despite never being trained on such datasets.

**Questions:**

- Is there any kind of pre-processing of the time series done? For example scaling, normalization, or time scaling?
- The model seems to be able to match the overall period of a signal well, but it essentially only does a local re-construction of the phase space dynamics. Do you have any insights on how the model is able to match the overall period? It would be interesting to see how the model predicts a simple limit cycle as you change the period.

**Ethical Concerns:**

["NO or VERY MINOR ethics concerns only"]

**Final Justification:**

The authors have addressed my open questions.

**Limitations:**

yes

**Paper Formatting Concerns:**

no concerns

**Quality:**

4

**Strengths And Weaknesses:**

Strengths:
- Very novel and innovative approach
- Strong numerical results for dynamical system and real world time series
- Well written and easy to follow.
- Comprehensive details about model setup, training procedure, parameter selection etc.

Weaknesses:
- On the forecasting task, MAE is not the best metric when evaluating accuracy across time series with different magnitude, because the metric can be dominated by some few time series in the dataset with larger magnitude. It would be better to use MASE or normalize MAE per metric and report geometric averages.
- Somewhat complex training setup relying on not so standard methods (STF). But everything is well documented and the code is available, which mitigates this to some extend.

---

> ### Author Rebuttal · Authors · 2025-07-30
>
> We thank the referee for the thorough reading, and the strong and very encouraging support of our work!
>
> **Weaknesses**
>
> **W1 (MASE instead of MAE):** Thanks for raising this point. We had standardized the time series for these comparisons, but in any case also computed the MASE now as suggested, with results reported in Table R1 below. As can be seen, using MASE, DynaMix even comes out top of the list, surpassing the best Chronos model even w.r.t. forecasting.
>
> Table R1: Forecast performance, same as Fig. 4C in paper, but for MASE (sorted from best to worst)
> | Rank | Model | MASE |
> | ------ | ------- | -------------- |
> | 1 | DynaMix | 0.36 ± 0.17 |
> | 2 | Chronos-t5-large | 0.37 ± 0.20 |
> | 3 | Chronos-t5-small | 0.42 ± 0.16 |
> | 4 | Chronos-t5-base | 0.42 ± 0.20 |
> | 5 | Chronos-t5-mini | 0.61 ± 0.26 |
> | 6 | Chronos-t5-tiny | 0.90 ± 0.47 |
> | 7 | Chronos-bolt-base | 1.46 ± 0.81 |
> | 8 | Chronos-bolt-small | 1.78 ± 0.84 |
> | 9 | Chronos-bolt-mini | 1.79 ± 0.85 |
> | 10 | Mamba4Cast | 2.00 ± 0.90 |
> | 11 | Chronos-bolt-tiny | 2.18 ± 0.80 |
> | 12 | TTM | 3.04 ± 1.25 |
>
> **W2 (complexity of training):** Yes the training is non-standard, but on the one hand, training techniques like STF are indeed important for achieving good dynamical systems reconstruction results, see [1,2] and Fig. 24. And on the other, our overall architecture is instead somewhat simpler we would argue than typical time series FMs.
>
> **Questions**
>
> **Q1)** Yes, all time series were standardized dimension-wise, see sect. 3.2 and A.2, for all models. We will make this more explicit in the revision!
>
> **Q2)** We are not 100% sure we understood the question correctly, but please note that Fig. 3b already demonstrates the capability of the model to forecast into regions of state space not even covered by the context information (in this sense non-local). We confirmed this now with two further examples (Van-der-Pol oscillator and Rayleigh oscillator) which we will include in the revision. We also confirmed on one example (cyclic Rössler DS) that DynaMix can, for instance, correctly forecast a limit cycle with half or twice the period of that experienced in training (Dstsp={0.48, 1.72}, DH={0.008, 0.026}, MASE={0.15, 0.36}).
>
> [1] Hess, F., et al. (2023). Generalized Teacher Forcing for Learning Chaotic Dynamics. Proceedings of the 40th International Conference on Machine Learning, 13017–13049. PMLR.
>
> [2] Mikhaeil, J., et al. (2022). On the difficulty of learning chaotic dynamics with RNNs. Advances in Neural Information Processing Systems, 35, 11297–11312.

---

> > ### Comment · Reviewer_beYK · 2025-08-05
> >
> > Thank you for the detailed response and the updated MASE metric.
> > I stand by my original assessment.

---

> > > ### Author Response · Authors · 2025-08-05
> > >
> > > Thank you so much for your encouraging feedback, and for your appreciation and strong support of our work!

---

### Decision · Program_Chairs · 2025-09-17

**Decision:**

Accept (poster)

**Comment:**

We thank the authors for their submission.

This work present a mixture of experts-based architecture for forecasting dynamical systems.  The model is trained on a wide range of dynamical systems, and is shown to be effective for zero-shot reconstruction of unseen dynamical systems.  The reviewers agreed that the proposed approach is novel and well-suited for the stated task.  They also agreed that the paper is clearly written, well-motivated, with thorough empirical evaluations.

Empirical results show that the architecture is efficient compared to baselines, while still achieving good performance.  Multiple reviewers initially called out the lack of ablation study as a weakness.  But a thorough back and forth with reviewer qUES and gq5P yielded more performance results as well as an informative ablation study.